# Ferritin heavy chain supports stability and function of the regulatory T cell lineage

Qian Wu [1,2,15], Ana Rita Carlos [1,3,15], Faouzi Braza[1,15], Marie-Louise Bergman [1], Jamil Z Kitoko [1], Patricia Bastos-Amador [1], Eloy Cuadrado[4], Rui Martins[1], Bruna Sabino Oliveira [1], Vera C Martins [1], Brendon P Scicluna [5], Jonathan JM Landry [6], Ferris E Jung [6], Temitope W Ademolue[1], Mirko Peitzsch [7], Jose Almeida-Santos [1], Jessica Thompson[1], Silvia Cardoso[1], Pedro Ventura[1], Manon Slot[4], Stamatia Rontogianni[4], Vanessa Ribeiro [3], Vital Da Silva Domingues [1], Inês A Cabral[1], Sebastian Weis [8,9,10], Marco Groth [11], Cristina Ameneiro [12], Miguel Fidalgo [12], Fudi Wang [13], Jocelyne Demengeot [1], Derk Amsen[4,14] & Miguel P Soares [1]✉

## Abstract

Regulatory T (TREG) cells develop via a program orchestrated by the transcription factor forkhead box protein P3 (FOXP3). Maintenance of the TREG cell lineage relies on sustained FOXP3 transcription via a mechanism involving demethylation of cytosine-phosphate-guanine (CpG)-rich elements at conserved non-coding sequences (CNS) in the FOXP3 locus. This cytosine demethylation is catalyzed by the ten–eleven translocation (TET) family of dioxygenases, and it involves a redox reaction that uses iron (Fe) as an essential cofactor. Here, we establish that human and mouse TREG cells express Fe-regulatory genes, including that encoding ferritin heavy chain (FTH), at relatively high levels compared to conventional T helper cells. We show that FTH expression in TREG cells is essential for immune homeostasis. Mechanistically, FTH supports TET-catalyzed demethylation of CpG-rich sequences CNS1 and 2 in the FOXP3 locus, thereby promoting FOXP3 transcription and TREG cell stability. This process, which is essential for TREG lineage stability and function, limits the severity of autoimmune neuroinflammation and infectious diseases, and favors tumor progression. These findings suggest that the regulation of intracellular iron by FTH is a stable property of TREG cells that supports immune homeostasis and limits the pathological outcomes of immune-mediated inflammation.

**Keywords** Regulatory T Cells; FOXP3; Iron Metabolism; Ferritin Heavy Chain; Ten–eleven Translocation Enzymes
**Subject Categories** Cancer; Chromatin, Transcription & Genomics; Immunology

## Introduction

Identified and characterized (Powrie and Mason, 1990; Sakaguchi et al, 1982) originally on the basis of their critical involvement in maintaining peripheral immune tolerance (Coutinho et al, 1993), regulatory T ($T_{REG}$) cells partake in different aspects of immune homeostasis (Campbell and Rudensky, 2020; Dikiy and Rudensky, 2023; Josefowicz et al, 2012; Panduro et al, 2016). One of the main functions of $T_{REG}$ cells, however, is most likely to restrain the breath of innate and adaptive immune responses against commensal microbes to prevent immunopathology (Belkaid, 2007; Demengeot et al, 2006). This evolutionarily conserved trait was probably co-opted through evolution to prevent peripheral self-reactive T and B cells from eliciting autoimmune diseases (Lafaille et al, 1994; Sakaguchi et al, 1995). As an evolutionary trade-off (Stearns and Medzhitov, 2015), $T_{REG}$ cells are pathogenic, for example, when limiting immune-mediated inflammatory responses to pathogens to promote chronic infections (Belkaid, 2007; Demengeot et al, 2006) or when restraining anti-tumor immunity, to promote cancer progression (Curiel et al, 2004; Liu et al, 2016).

$T_{REG}$ cell development and function are controlled by the X-chromosome-encoded transcription factor FOXP3 (Fontenot et al, 2003; Hori et al, 2003), together with auxiliary transcriptional

[1]Instituto Gulbenkian de Ciência, Oeiras, Portugal. [2]International Institutes of Medicine, the Fourth Affiliated Hospital of Zhejiang University, School of Medicine, Yiwu, Zhejiang, China. [3]Departamento de Biologia Animal, Centro de Ecologia, Evolução e Alterações Ambientais, Faculdade de Ciências, Universidade de Lisboa, Lisboa, Portugal. [4]Department of Hematopoiesis and Department of Immunopathology, Sanquin Research and Landsteiner Laboratory, Amsterdam, The Netherlands. [5]Department of Applied Biomedical Science, Faculty of Health Sciences, Mater Dei Hospital, and Centre for Molecular Medicine and Biobanking, University of Malta, Msida, Malta. [6]Genomic Core Facility, European Molecular Biology Laboratory, Heidelberg, Germany. [7]Institute for Clinical Chemistry and Laboratory Medicine, University Clinic Carl Gustav Carus, TU Dresden, Dresden, Germany. [8]Department for Anesthesiology and Intensive Care Medicine, Jena University Hospital, Friedrich-Schiller University, Jena, Germany. [9]Institute for Infectious Disease and Infection Control, Jena University Hospital, Friedrich-Schiller University, Jena, Germany. [10]Leibniz Institute for Natural Product Research and Infection Biology, Hans-Knöll Institute-HKI, Jena, Germany. [11]Leibniz Institute on Aging-Fritz Lipmann Institute, Jena, Germany. [12]Center for Research in Molecular Medicine and Chronic Diseases (CiMUS), Universidade de Santiago de Compostela-Health Research Institute (IDIS), Santiago de Compostela, Spain. [13]The Second Affiliated Hospital, School of Public Health, Zhejiang University School of Medicine, Hangzhou 310058, China. [14]Department of Experimental Immunology, Amsterdam UMC, University of Amsterdam, Amsterdam, The Netherlands. [15]These authors contributed equally: Qian Wu, Ana Rita Carlos, Faouzi Braza. ✉E-mail: mpsoares@igc.gulbenkian.pt

regulators (Kanamori et al, 2016). The transcriptional program enforced by FOXP3 specifies $T_{REG}$ cell lineage commitment in the thymus and in the periphery (Fontenot et al, 2003; Hori et al, 2003; Lee et al, 2012), generating thymic $T_{REG}$ ($tT_{REG}$) cells and peripherally derived $T_{REG}$ ($pT_{REG}$) cells, respectively (Chen et al, 2003). Sustained *FOXP3* transcription maintains $T_{REG}$ cell lineage stability (Williams and Rudensky, 2007), avoiding transdifferentiation towards pro-inflammatory T helper ($T_H$) cells (Gavin et al, 2007; Morikawa et al, 2014).

*FOXP3* transcription is regulated by different signal transduction pathways, emanating from the T-cell receptor (TCR), interleukin (IL-2) receptor, and TGF-β receptor (Bennett et al, 2001; Brunkow et al, 2001; Hori and Sakaguchi, 2004), among others. Sustained *FOXP3* transcription is enforced epigenetically (Gavin et al, 2007; Morikawa et al, 2014), in response to environmental cues (Chapman et al, 2020; Shi and Chi, 2019) that regulate different aspects of $T_{REG}$ cell metabolism (Etchegaray and Mostoslavsky, 2016). These epigenetic modifications include the relative methylation status of cytosine-phosphate-guanine (CpG)-rich sequences in the *FOXP3* conserved non-coding sequences (CNS) 1, 2, and 3 (Ohkura et al, 2012; Zheng et al, 2010), whereby cytosine methylation represses while demethylation sustains *FOXP3* transcription (Ohkura et al, 2012; Zheng et al, 2010).

Cytosine methylation is catalyzed by DNA methyltransferase (DNMT) (Ohkura et al, 2012), while demethylation is catalyzed by the ten–eleven translocation (TET) family of dioxygenases (Wu and Zhang, 2017; Yue et al, 2016). Cytosine demethylation consists on redox-based reactions that oxidize 5-methylcytosine (5-mC) into 5-hydroxymethylcytosine (5-hmC), 5-formylcytosine (5-fC) and 5-carboxylcytosine (5caC) (Kohli and Zhang, 2013). TET dioxygenases catalyze cytosine demethylation at *FOXP3* CNS1 and 2 (Ohkura et al, 2012), supporting $T_{REG}$ cell lineage stability (Nakatsukasa et al, 2019; Wu and Zhang, 2017; Yue et al, 2019; Yue et al, 2016), and preventing $T_{REG}$ cell transdifferentiating into inflammatory effector $T_H$ cells, also referred as ex-$T_{REG}$ cells (Duarte et al, 2009; Komatsu et al, 2009; Zhou et al, 2009).

TET dioxygenases use Fe as an essential cofactor and the intermediate metabolite α-ketoglutarate as an obligatory substrate (Huang and Rao, 2014; Pastor et al, 2013). This TET reliance on Fe availability entertained the hypothesis that regulation of cellular Fe metabolism acts upstream of TET dioxygenases to modulate $T_{REG}$ cell lineage stability.

Several studies have shown that Fe metabolism impacts on immunity. For example, intracellular Fe availability and redox activity is essential to support B and T-cell development (Vanoaica et al, 2014), via a cytoprotective mechanism exerted by the ferritin H chain (FTH) (Berberat et al, 2003; Pham et al, 2004), likely involving the mitochondria (Blankenhaus et al, 2019; Vanoaica et al, 2014). Regulation of cellular Fe content and redox activity also modulate cytokine production by effector $T_H$ cells, via a mechanism involving the PolyC-RNA-Binding Protein 1 (PCBP1) (Wang et al, 2018). Cellular Fe import, via the transferrin receptor 1 (*TFR1/CD71*), supports $T_H$ type 1 ($T_H1$) cell immunity and its regulation by induced $T_{REG}$ ($iT_{REG}$) cells (Voss et al, 2023) as well as antibody responses to vaccination (Frost et al, 2021; Jiang et al, 2019) and immunity against infection by pathogens such as *Plasmodium*, the causative agent of malaria (Wideman et al, 2023).

Here, we demonstrate that regulation of Fe metabolism by FTH operates upstream of TET dioxygenases to enforce cytosine demethylation at CpG-rich sequences in the CNS1 and 2 of the *FOXP3* locus, sustaining *FOXP3* transcription, expression and $T_{REG}$ cell lineage identity. This cell-intrinsic property of $T_{REG}$ cells is essential to maintain immune homeostasis while exerting a major impact on the outcome of immune-driven inflammation.

# Results

## $T_{REG}$ cells express relatively high levels of FTH

In a previously unbiased proteomics analysis, we found that freshly isolated human naive CD45RA$^+$CD25$^{hi}$ $T_{REG}$ ($nT_{REG}$) and memory CD45RA$^-$CD25$^{hi}$ $T_{REG}$ ($mT_{REG}$) cells expressed relatively higher levels of Fe-regulatory proteins, including FTH and ferritin L chain (FTL), when compared to CD45RA$^+$CD25$^-$ naive conventional (nTconv) cells or to CD45RA$^-$CD25$^-$ activated/memory (mTconv) cells (Cuadrado et al, 2018). The relatively higher expression of the FTH and FTL components of the ferritin complex was maintained upon expansion of human CD4$^+$CD127$^-$CD25$^+$ $T_{REG}$ cells in vitro, in comparison to $T_{CONV}$ cells CD4$^+$CD127$^+$CD25$^-$ cells (Fig. 1A,B). Similarly, mouse CD4$^+$Foxp3$^+$ $T_{REG}$ cells also expressed relatively higher levels of FTH protein, compared to CD4$^+$Foxp3$^-$CD44$^{low}$CD62L$^{high}$ naive $T_H$ cells ($T_N$) or CD4$^+$Foxp3$^-$CD44$^{high}$CD62L$^{low}$ memory $T_H$ cells ($T_M$), as determined by western blot (Fig. 1C,D). This suggests that sustained and elevated levels of ferritin expression are a stable property of human and mouse $T_{REG}$ cells. Of note, mouse $T_{REG}$ cells express similar levels of *Fth* mRNA, compared to $T_N$ cells, while (CD4$^+$Foxp3$^-$CD44$^{high}$CD62L$^-$) $T_M$ cells express relatively higher levels of *Fth* mRNA, compared to $T_{REG}$ cells (Appendix Fig. S1). This suggests that the relatively higher level of FTH protein expression in $T_{REG}$ cells is enforced post-transcriptionally, similar to other cell types (Hentze et al, 1987; Meyron-Holtz et al, 2004; Muckenthaler et al, 2017; Rouault et al, 1988).

We monitored FTH expression in induced $T_{REG}$ ($iT_{REG}$) cells, generated from mouse $T_N$ cells activated in vitro with anti-CD3/CD28 mAb plus IL-2 and TGFβ (Chen et al, 2003) (Fig. 1E). To this aim we used *Foxp3$^{GFP}$* $T_{REG}$ cell reporter mice, in which a green fluorescent protein (GFP) humanized Cre-recombinase (GFP-hCre) coding sequence is inserted downstream of the *Foxp3* ATG translational start codon in a bacterial artificial chromosome (BAC) transgene carrying the intact *Foxp3* promoter (Foxp3-GFP-hCre; referred herein as *Foxp3$^{GFP}$*) (Chen et al, 2003; Zhou et al, 2008). FTH protein expression was higher under culture conditions containing TGFβ and enriched in CD4$^+$GFP$^+$ $iT_{REG}$ cells, compared to culture conditions lacking TGFβ and enriched in CD4$^+$GFP$^-$ $T_{CONV}$ cells (Fig. 1F). A similar trend was observed for *Fth* mRNA, which was upregulated in $iT_{REG}$ cells (Fig. 1G). The relative level of *Fth* mRNA expression was similar in thymic, peripheral and $iT_{REG}$ cells (Appendix Fig. S2).

## FTH expression in $T_{REG}$ cells is essential to maintain immune homeostasis

To determine the effect of regulation of intracellular Fe metabolism by FTH on $T_{REG}$ cells, we introduced an additional *loxP*-flanked *Fth* allele (*Fth$^{fl/fl}$*) (Darshan et al, 2009) into *Foxp3$^{GFP}$* mice (Zhou et al, 2008), deleting *Fth* specifically in $T_{REG}$ cells from *Foxp3$^{GFP-FthΔ/Δ}$* vs. control *Foxp3$^{GFP}$* mice (Fig. EV1A). *Fth* deletion was associated with

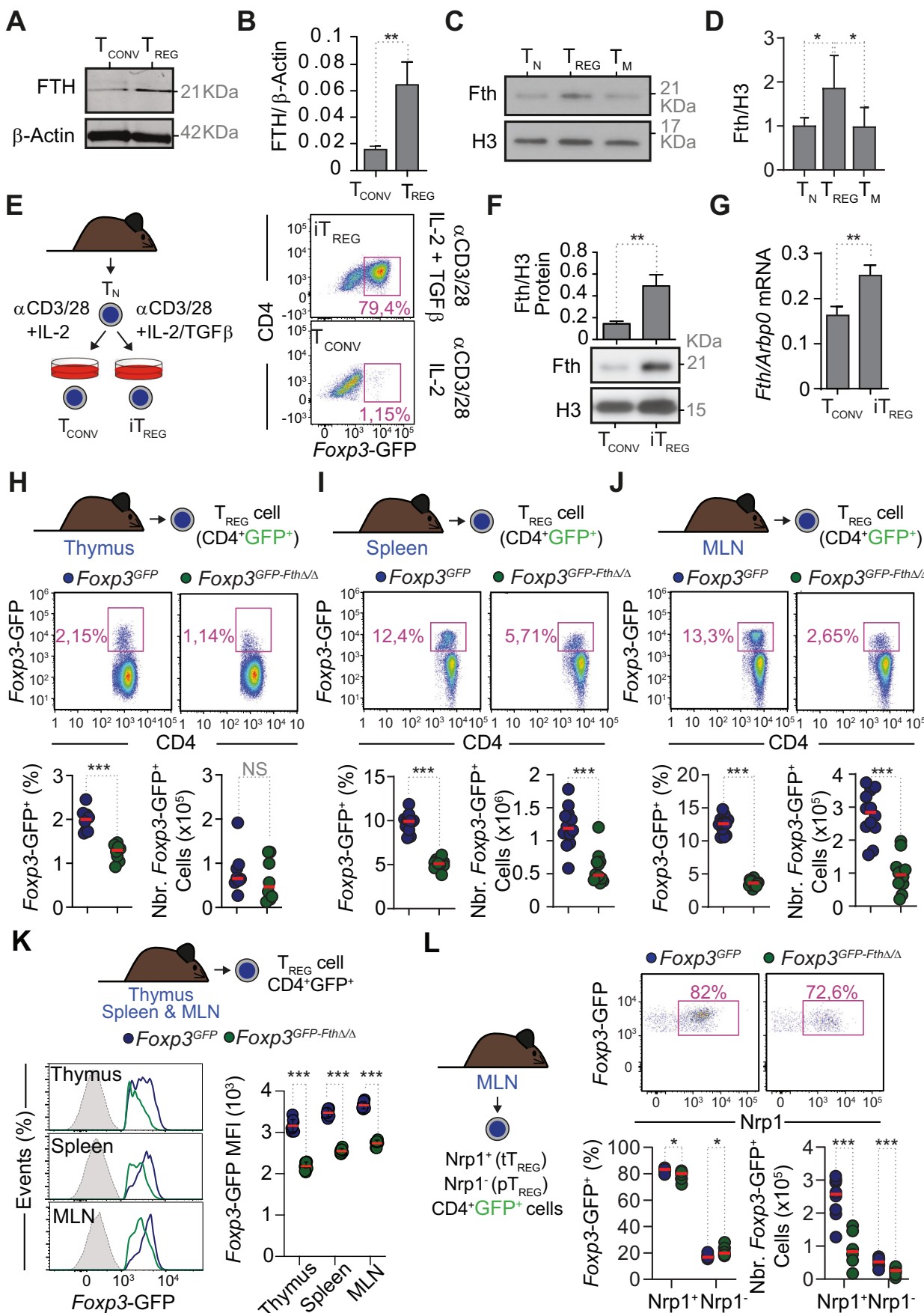

### Figure 1. T$_{REG}$ cell ferritin expression is a stable property of human and mouse T$_{REG}$ cells.

(A) FTH and β-Actin protein expression, detected by western blot in whole-cell extracts from human T conventional (CD4$^+$CD45RA$^+$CD127$^+$CD25$^-$; T$_{CONV}$) and T$_{REG}$ (CD4$^+$CD127$^-$CD45RA$^+$CD25$^{hi}$) cells after two weeks of expansion with anti-CD3, anti-CD28 mAb and IL-2. (B) Relative quantification of FTH protein expression, normalized to β-Actin, detected by western blot as in (A). $n = 3$ independent experiments. (C) FTH and histone H3 protein expression detected by western blot in whole-cell extracts from sorted mouse naive T cells (CD4$^+$Foxp3$^-$CD44$^{low}$CD62L$^{high}$; T$_N$), (CD4$^+$Foxp3$^+$GFP$^+$) T$_{REG}$ cells and memory T cells (CD4$^+$Foxp3$^-$CD44$^{high}$CD62L$^{low}$; T$_M$), by western blot. (D) Relative quantification of FTH, normalized to histone H3, protein expression, detected by western blot as in (C). Data were normalized to FTH expression in T$_N$ cells, pooled from four independent experiments. (E) Schematic representation of the protocol used for the generation of iT$_{REG}$ and representative flow cytometry dot plots of mouse iT$_{REG}$ generated from sorted naive T cells (T$_N$), stimulated with anti-CD3 and anti-CD28 mAb plus IL-2 and TGFβ for 5 days. Control T$_{CONV}$ cells were subjected to the same experimental conditions, without TGFβ. (F) FTH and histone H3 protein expression, detected by western blot in whole-cell extracts from iT$_{REG}$ and T$_{CONV}$ generated as depicted in (E). Data pooled from three independent experiments with similar trend. (G) The relative level of *Fth* mRNA expression, quantified by qRT-PCR, using *Arbp0* as housekeeping gene. Data pooled from three independent experiments, with similar trend. (H–J) Schematic representation of the protocol used (top panels), representative flow cytometry dot plots (middle panels) and corresponding quantification of percentage (%) in CD4$^+$ cells and cell number (Nbr.) (bottom panels) of live (TCRβ$^+$CD4$^+$ Foxp3$^+$) GFP$^+$ T$_{REG}$ cells in (H) thymus, (I) spleen and (J) mesenteric LN (MLN). (H) Data from $N = 8$ mice per genotype, per organ, from two independent experiments, with similar trend. (I, J) Data from $N = 12$ mice per genotype, per organ, from three independent experiments, with similar trend. (K) Schematic representation of the experimental approach (top panel) used to monitor the expression of the GFP-hCre transgene in the thymus, spleen, and MLN of (CD4$^+$GFP$^+$) T$_{REG}$ cells. Representative flow cytometry histograms of GFP-hCre (bottom left panel). Relative quantification of GFP-hCre expression (bottom right panel), represented as mean fluorescence intensity (MFI). Data from $N = 8$ mice per genotype, pooled from two independent experiments with similar trend. (L) Schematic representation of the experimental approach (left panel) used, representative flow cytometry dot plots (top panel) and corresponding quantification of percentage (%) (bottom left panel) and cell number (Nbr.) (bottom right panel) of live (TCRβ$^+$CD4$^+$ GFP$^+$) Nrp1$^+$ and Nrp1$^-$T$_{REG}$ cells in MLN. Data from $N = 8$ mice per genotype, pooled from two independent experiments with similar trend. Data information: Data in (B, F, G) are presented as mean ± SD. Data in (D) are presented as mean ± SEM. Circles in (H–L) correspond to individual mice. P values in (B, F–J) determined using unpaired t test with Welch's correction, in (D, H–J) using ordinary one-way ANOVA, and in (K, L) using Two-way ANOVA with Sidak's multiple comparisons test. NS not significant ($P > 0.05$); *$P < 0.05$; **$P < 0.01$; ***$P < 0.001$. Source data are available online for this figure.

the accumulation of intracellular labile Fe$^{2+}$ in T$_{REG}$ cells, isolated from the mesenteric lymph nodes (MLN) of *Foxp3$^{GFP-FthΔ/Δ}$* vs. control *Foxp3$^{GFP}$* mice (Fig. EV1B).

The frequency of thymic CD4$^+$GFP$^+$ T$_{REG}$ cells was lower in *Foxp3$^{GFP-FthΔ/Δ}$* vs. control *Foxp3$^{GFP}$* mice (Fig. 1H). This was not associated, however, with a concomitant reduction in the number of CD4$^+$GFP$^+$ T$_{REG}$ cells (Fig. 1H). In contrast, there was a marked reduction of both the frequency and numbers of T$_{REG}$ cells in the spleen (Fig. 1I) and in the MLN (Fig. 1J) of *Foxp3$^{GFP-FthΔ/Δ}$* vs. control *Foxp3$^{GFP}$* mice. The frequency and number of CD4$^+$GFP$^+$CXCR5$^+$PD1$^+$ follicular T$_{REG}$ (FT$_{REG}$) cells was also decreased in the spleen of *Foxp3$^{GFP-FthΔ/Δ}$* vs. control *Foxp3$^{GFP}$* mice (Fig. EV1C). These observations suggest that regulation of intracellular Fe by FTH is required to sustain the number of circulating T$_{REG}$ and FT$_{REG}$ cells in the periphery, without interfering with thymic T$_{REG}$ cell output.

We noticed that the relative level of GFP expression, reporting on *Foxp3* transcription under the control of an intact *Foxp3* locus (Chen et al, 2003; Zhou et al, 2008), was reduced in thymic, splenic and MLN T$_{REG}$ cells from *Foxp3$^{GFP-FthΔ/Δ}$* vs. control *Foxp3$^{GFP}$* mice (Fig. 1K). These observations suggest that FTH regulates *Foxp3* transcription in the thymus as well as in circulating T$_{REG}$ cells, which is not sufficient however, to interfere with thymic T$_{REG}$ cell output.

Thymic and peripheral T$_{REG}$ cell development give rise to tT$_{REG}$ and iT$_{REG}$ cells, expressing neuropilin1 (Nrp1) or not, respectively (Weiss et al, 2012; Yadav et al, 2012). The frequency and number of Nrp1$^+$ tT$_{REG}$ cells and Nrp1$^-$ iT$_{REG}$ cells was reduced, to the same extent, as assessed in the MLN (Fig. 1L) of *Foxp3$^{GFP-FthΔ/Δ}$* vs. control *Foxp3$^{GFP}$* mice. This suggests that regulation of Fe metabolism by FTH is required for the maintenance of thymic-derived and peripherally induced T$_{REG}$ cells.

### FTH restrains T$_{REG}$ cell transdifferentiation into inflammatory ex-T$_{REG}$ cells

The reduction in T$_{REG}$ cells imposed by *Fth* deletion was associated with an accumulation of activated CD4$^+$CD44$^{high}$CD62L$^{low}$ T$_{CONV}$ cells and CD8$^+$CD44$^{high}$CD62L$^{low}$ cytotoxic T (T$_C$) cells, in the

spleen (Fig. EV1D) and in MLN of *Foxp3$^{GFP-FthΔ/Δ}$* vs. control *Fth$^{fl/fl}$* mice (Fig. EV1E). Concomitantly, there was an increase in the frequency of interferon-γ (IFN γ)-expressing activated CD4$^+$ T$_H$ cells and CD8$^+$ T$_C$ cells in the spleen (Fig. EV1F) and MLN (Fig. EV1G). These observations suggest that regulation of Fe metabolism in T$_{REG}$ cells is essential to maintain immune homeostasis, preventing the activation and accumulation of inflammatory CD4$^+$ T$_H$ cells and CD8$^+$ T cells.

Secreted ferritin complexes can restrain human T-cell proliferation in vitro (Gray et al, 2001), entertaining the hypothesis that ferritin secretion supports the antiproliferative function of T$_{REG}$ cells. However, T$_{REG}$ cells from *Foxp3$^{GFP-FthΔ/Δ}$* mice inhibited T-cell proliferation in vitro, to a similar extent as T$_{REG}$ cells from *Foxp3$^{GFP}$* (Fig. EV2A). This suggests that FTH is not essential to support the antiproliferative function of T$_{REG}$ cells, consistent with *Foxp3$^{GFP-FthΔ/Δ}$* mice not developing overt autoimmune pathologic lesions, compared to control *Fth$^{fl/fl}$* mice (Fig. EV2B).

To gain further insight into the mechanism via which FTH modulates T$_{REG}$ cell function in vivo, we performed RNA sequencing (RNAseq), to compare the gene expression profiles of CD4$^+$ CD25$^+$GFP$^+$ T$_{REG}$ cells sorted from the lymph nodes of *Foxp3$^{GFP-FthΔ/Δ}$* vs. control *Foxp3$^{GFP}$* mice (Fig. 2A). T$_{REG}$ cells from *Foxp3$^{GFP-FthΔ/Δ}$* mice upregulated 1832 genes and downregulated 1340 genes, compared to T$_{REG}$ cells from control *Foxp3$^{GFP}$* mice (Fig. 2B). *Fth* deletion was associated with dysregulation of Foxp3-dependent and -independent "T$_{REG}$ transcriptional signature" (Hill et al, 2007), affecting at least 194 genes involved in T$_{REG}$ cell function and lineage maintenance (Fig. 2C). Pathway enrichment analysis (Fig. 2D,E) showed that T$_{REG}$ cells from *Foxp3$^{GFP-FthΔ/Δ}$* mice presented T$_H$1 and type 2 (T$_H$2) transcriptional signatures (Fig. 2D), as illustrated by the induction of the transcriptional master regulators of T$_H$1 and T$_H$2 effector functions, *Tbx21* (T-box transcription factor; T-bet) and *Gata3* (GATA Binding Protein 3) as well as *Runx3* (RUNX Family Transcription Factor 3), respectively (Fig. EV2C).

Consistently, the percentage of activated CD4$^+$GFP$^+$ CD44$^{high}$CD62L$^{low}$ T$_{REG}$ cells was higher in the spleen and MLN

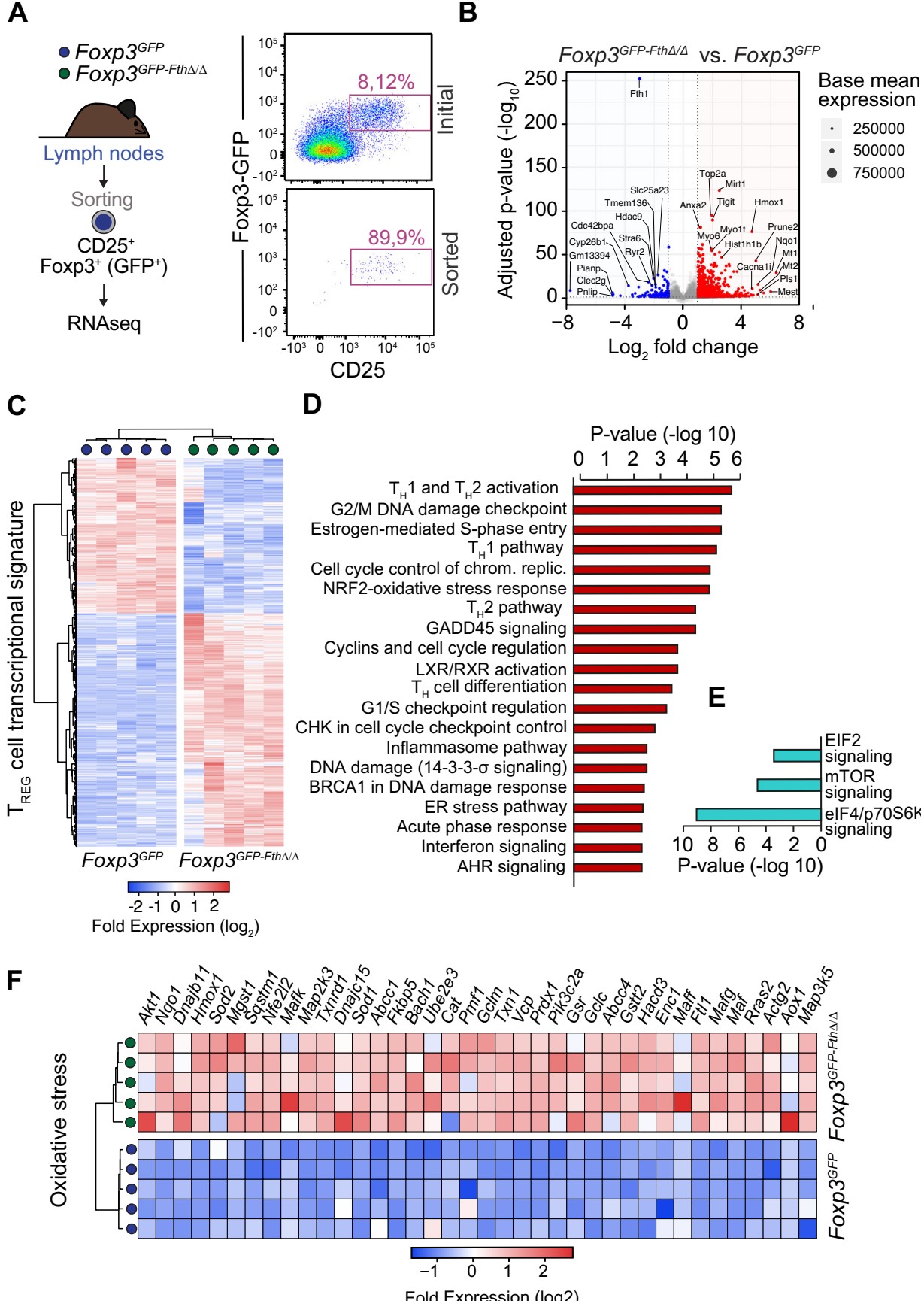

**Figure 2.  FTH expression in T_REG cells prevents transdifferentiation into inflammatory ex-T_REG cells.**

(A) Schematic representation of experimental approach (left panel) and representative flow cytometry dot plots of (CD4$^+$CD25$^+$GFP$^+$) T_REG cells from the lymph nodes before and after sorting. (B) Volcano plot representation of RNA sequencing data of genes overexpressed (red) or under-expressed (blue) in (CD4$^+$CD25$^+$GFP$^+$) T_REG cells sorted from *Foxp3$^{GFP-FthΔ/Δ}$* and control *Foxp3$^{GFP}$* (N = 5 per genotype) mice. P values determined by Benjamini and Hochberg adjusted probabilities. (C) Heatmap representation of T_REG transcriptional signature genes differentially expressed in T_REG cells sorted from *Foxp3$^{GFP-FthΔ/Δ}$* vs. *Foxp3$^{GFP}$* mice, as illustrated in (A, B). (D, E) Pathway enrichment analysis of genetic programs overexpressed (D) or under-expressed (E) in T_REG cells sorted from *Foxp3$^{GFP-FthΔ/Δ}$* vs. *Foxp3$^{GFP}$* mice, as illustrated in (A, B). Data were analyzed using g:SCS multiple testing correction method with a significance threshold of 0.05. (F) Heatmap representation of individual genes associated with oxidative stress-responsive programs, differentially expressed in T_REG cells sorted from *Foxp3$^{GFP-FthΔ/Δ}$* vs. *Foxp3$^{GFP}$* mice, as illustrated in (A, B). Source data are available online for this figure.

from *Foxp3$^{GFP-FthΔ/Δ}$* vs. control *Foxp3$^{GFP}$* mice (Fig. EV2D). This was associated with an increase in the frequency of CD4$^+$GFP$^+$ T_REG cells expressing IFNγ in the spleen (Fig. EV2E). This suggests that FTH is essential to prevent the transdifferentiation of T_REG cells into inflammatory T_H cells.

RNAseq analysis also revealed that *Fth* deletion in T_REG cells led to the induction of the canonical oxidative stress response controlled by the transcription factor nuclear factor erythroid-derived 2-like 2 (NRF2) (Fig. 2D). This was associated with the activation of other canonical stress responses, including the cell cycle and DNA damage, unfolded protein, and hypoxic response (Fig. 2D) as well as an overall shutdown of eIF2, mTOR and eIF4/p70s6K signaling transduction pathways (Fig. 2E). Activation of the oxidative stress response regulated by NRF2 was characterized by the induction of *Nqo1* and *Hmox1*, among several other NRF2-regulated genes (Fig. 2F). These observations suggest that FTH is essential to support a transcriptional profile that maintains T_REG cell redox homeostasis, similar to described in other cell types (Blankenhaus et al, 2019; Vanoaica et al, 2014).

## FTH supports T_REG cell lineage maintenance

To establish whether FTH enforces T_REG cell lineage maintenance and prevents the transdifferentiation of T_REG cells into inflammatory T_H cells, an additional Rosa26-tandem dimer (td) Tomato-Flox-stop-Flox allele was introduced into *Foxp3$^{GFP-FthΔ/Δ}$* mice, driving the expression of tdTomato (tdT) by Cre-driven excision of a Flox-stop-Flox cassette, under the control of Foxp3 regulatory regions (Fig. 3A). The resulting *Foxp3$^{GFP-FthΔ/Δ-tdT}$* mice allow distinguishing CD4$^+$GFP$^+$tdT$^+$ T_REG from CD4$^+$GFP$^-$tdT$^+$ ex-T_REG cells that at some point in their developmental history downregulated *Foxp3* (i.e., GFP$^-$), while retaining TdT expression (Fig. 3A). *Fth* deletion in T_REG cells was associated with a progressive reduction in the frequency of circulating T_REG cells from *Foxp3$^{GFP-FthΔ/Δ-tdT}$* vs. control *Foxp3$^{GFP-tdT}$* mice, as assessed from 2 to 24 weeks after birth (Figs. 3B and EV3A). Concomitantly, there was an increase in the frequency of circulating ex-T_REG cells (Figs. 3C and EV3A), with 60% of circulating T_REG cells becoming ex-T_REG cells in *Foxp3$^{GFP-FthΔ/Δ-tdT}$*, 24 weeks after birth (Figs. 3D and EV3A). This relative enrichment in the proportion of ex-T_REG cells suggests that *Fth* deletion promotes the conversion of T_REG cells into ex-T_REG cells.

We reasoned that if FTH prevents T_REG cells from transdifferentiating into ex-T_REG cells, then *Fth* deletion in T_REG cells should be associated with an accumulation of ex-T_REG cells in the spleen and/or lymph nodes. As expected, the percentage and number of T_REG cells were reduced in the spleen from *Foxp3$^{GFP-FthΔ/Δ-tdT}$* vs. control *Foxp3$^{GFP-tdT}$* mice, as assessed at 19–30 weeks after birth (Fig. 3E). The percentage and number of splenic ex-T_REG cells remained relatively stable (Fig. 3F), but the ratio of ex-T_REG over tdT$^+$ cells was higher in the spleen from

*Foxp3$^{GFP-FthΔ/Δ-tdT}$* vs. control *Foxp3$^{GFP-tdT}$* mice (Fig. 3G), with over 80% of T_REG cells becoming ex-T_REG cells (Fig. 3G). *Fth* was deleted in ex-T_REG cells from *Foxp3$^{GFP-FthΔ/Δ-tdT}$* mice, as determined by qRT-PCR (Fig. EV3B), confirming that the ex-T_REG cells in *Foxp3$^{GFP-FthΔ/Δ-tdT}$* mice do originate from T_REG cells, in which the *Fth* allele was deleted under the control of the *Foxp3* promoter.

We then asked whether the transdifferentiation of T_REG cells into ex-T_REG cells was associated with the expression of pro-inflammatory cytokine, which is a feature of T_H cells activation. In strong support of this hypothesis, a large percentage of T_REG and ex-T_REG cells in the LN and to a lesser extent in the spleen from *Foxp3$^{GFP-FthΔ/Δ-tdT}$* mice expressed IFNγ, as compared to the lack of IFNγ expression in T_REG and ex-T_REG cells from control *Foxp3$^{GFP-tdT}$* mice (Figs. 3H and EV3C). Moreover, a significant proportion of T_REG and ex-T_REG cells in the LN (Fig. 3I) and spleen (Fig. EV3D) from *Foxp3$^{GFP-FthΔ/Δ-tdT}$* mice co-expressed the proliferation markers Ki67 and CD71 (i.e., transferrin receptor), as compared to T_REG and ex-T_REG cells from control *Foxp3$^{GFP-tdT}$* mice (Figs. 3I and EV3D). This was not associated, however, with changes in the relative levels of CD71 expression (Fig. EV3E). Taken together these observations suggest that FTH is essential to restrain T_REG cells from transdifferentiating into inflammatory ex-T_REG cells.

## FTH supports T_REG lineage maintenance in a cell-autonomous manner

To disentangle cell-autonomous from systemic effects associated with *Fth* deletion in T_REG cells we used mixed bone marrow (BM) chimeric mice. Briefly, sub lethally irradiated lymphogenic *Rag2*-deficient (*Rag2$^{-/-}$*) mice were reconstituted with BM cells from CD45.2$^+$ *Foxp3$^{GFP-FthΔ/Δ-tdT}$* vs. *Foxp3$^{GFP-tdT}$* mice (50%), plus congenic BM cells (50%) from CD45.1$^+$ C57BL/6 mice (Fig. 4A). The proportion of CD45.2$^+$ vs. CD45.1$^+$ T_REG cells in LN (Fig. 4A–C) was markedly reduced in BM chimeric mice, when reconstituted with BM cells from *Foxp3$^{GFP-FthΔ/Δ-tdT}$* vs. *Foxp3$^{GFP-tdT}$* mice. In contrast, there were no differences in the relative proportion of CD45.2$^+$ vs. CD45.1$^+$ T_REG cells in the thymus of these BM chimeric mice (Fig. EV3F,G). This suggests that FTH sustains the number of circulating T_REG cells via a cell-autonomous mechanism that does not affect T_REG cell development in the thymus.

The proportion of CD45.2$^+$ vs. CD45.1$^+$ Foxp3$^-$CD3$^+$CD4$^+$ T_H cells and CD3$^+$CD8$^+$ T_C cells was indistinguishable in the LN (Fig. 4A–C) of mixed BM chimeras reconstituted with CD45.2$^+$ BM cells from *Foxp3$^{GFP-FthΔ/Δ-tdT}$* vs. control *Foxp3$^{GFP-tdT}$* mice. Moreover, the percentage and number of activated CD45.2$^+$CD4$^+$ CD44$^{high}$CD62L$^{low}$ T_CONV cells and CD45.2$^+$CD8$^+$CD44$^{high}$CD62L$^{low}$ T_C cells in the LN (Fig. EV4A–C) were also similar in these BM chimeric mice. This confirms that FTH sustains the number of

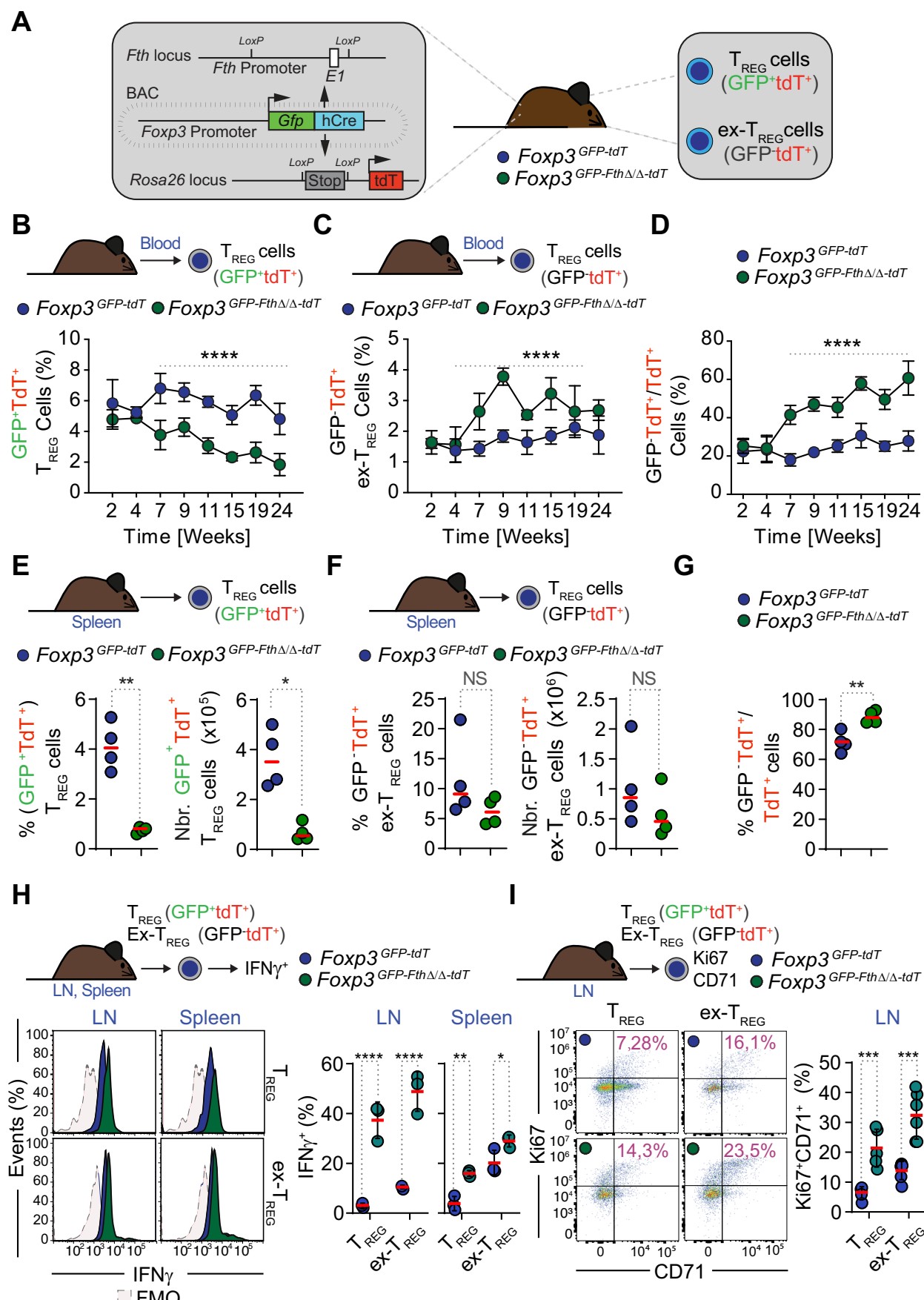

◄

**Figure 3. FTH enforces $T_{REG}$ cell lineage stability.**

(A) Schematic representation of $Foxp3^{GFP\text{-}Fth\Delta/\Delta\text{-}tdT}$ mice used to monitor the transition of (CD4$^+$GFP$^+$tdT$^+$) $T_{REG}$ cells into (CD4$^+$GFP$^-$tdT$^+$) ex-$T_{REG}$ cells that repressed GFP expression while retaining the expression of a tdT transgene. (B–D) Percentage of circulating: (B) $T_{REG}$ and (C) ex-$T_{REG}$ cells in $Foxp3^{GFP\text{-}Fth\Delta/\Delta\text{-}tdT}$ and control $Foxp3^{GFP\text{-}tdT}$ mice, and (D) Percentage of ex-$T_{REG}$ cells among CD4$^+$tdT$^+$ cells, calculated as the ratio of CD4$^+$GFP$^-$tdT$^+$/CD4$^+$tdT$^+$ cells in the same mice as (B, C). Data from $N = 4$–6 mice per genotype was pooled from three independent experiments with a similar trend. (E–G) Percentage and number of splenic (E) $T_{REG}$ cells, (F) ex-$T_{REG}$ cells and (G) relative percentage of ex-$T_{REG}$ cells over total CD4$^+$tdT$^+$ cells. Data from $N = 4$ mice per genotype, pooled from two independent experiments with similar trends. (H) Representative flow cytometry histograms of IFNγ expression by live activated (CD4$^+$GFP$^+$tdT$^+$) $T_{REG}$ and (CD4$^+$GFP$^-$tdT$^+$) ex-$T_{REG}$ cells in lymph nodes and spleen from $Foxp3^{GFP\text{-}Fth\Delta/\Delta\text{-}tdT}$ and control $Foxp3^{GFP\text{-}tdT}$ mice (left panels) and corresponding quantification of the percentage of IFNγ expressing (CD4$^+$GFP$^+$tdT$^+$) $T_{REG}$ and (CD4$^+$GFP$^-$tdT$^+$) ex-$T_{REG}$ cells (right panels). Expression of IFNγ was induced upon Phorbol-12-myristate-13-acetate (PMA) and Ionomycin re-activation in vitro. Data from $N = 3$ wells per genotype, in one experiment, representative of two independent experiments with similar trend. (I) Representative flow cytometry dot plots (left panels) and corresponding quantification (right panel) of the percentage of (CD4$^+$GFP$^+$tdT$^+$) $T_{REG}$ and (CD4$^+$GFP$^-$tdT$^+$) ex-$T_{REG}$ cells expressing Ki67 and CD71 in the lymph nodes $Foxp3^{GFP\text{-}Fth\Delta/\Delta\text{-}tdT}$ and control $Foxp3^{GFP\text{-}tdT}$ mice. Data from $N = 6$ mice per genotype, pooled from two independent experiments, with similar trend. Data information: Data in (B–D) represented as mean ± SD. Circles correspond to mean values. Data in (E–I) circles correspond to individual mice and red bars to mean values. (H, I) represented as mean ± SD. $P$ values in (B–D, H, I) calculated using Two-way ANOVA analysis with Sidak's multiple comparisons test and in (E–G) with unpaired $t$ test with Welch's correction. NS not significant ($P > 0.05$); *$P < 0.05$; **$P < 0.01$; ***$P < 0.001$; ****$P < 0.0001$. Source data are available online for this figure.

circulating $T_{REG}$ cells, via a cell-autonomous mechanism that acts irrespectively of the systemic inflammatory response associated with *Fth* deletion in $T_{REG}$ cells from $Foxp3^{GFP\text{-}Fth\Delta/\Delta}$ mice (Fig. EV1D–G).

We then asked whether FTH restrains the transdifferentiation of $T_{REG}$ cells towards ex-$T_{REG}$ cells via a cell-autonomous mechanism. In support of this notion, there was a marked reduction in the percentage and number of CD45.2$^+$CD3$^+$CD4$^+$GFP$^+$tdT$^+$ $T_{REG}$ cells (Fig. 4D,E) and CD45.2$^+$CD3$^+$CD4$^+$GFP$^-$tdT$^+$ ex-$T_{REG}$ cells (Fig. 4D,F) in the LN of mixed BM chimeric mice reconstituted with BM cells from $Foxp3^{GFP\text{-}Fth\Delta/\Delta\text{-}tdT}$ vs. $Foxp3^{GFP\text{-}tdT}$ mice. The ratio of CD45.2$^+$ ex-$T_{REG}$ cells over tdT$^+$ cells was increased in the LN from chimeric mice reconstituted with BM cells from $Foxp3^{GFP\text{-}Fth\Delta/\Delta\text{-}tdT}$ vs. $Foxp3^{GFP\text{-}tdT}$ mice (Fig. 4G). This suggests that FTH maintains peripheral $T_{REG}$ cell lineage stability, via a cell-autonomous mechanism, irrespectively of the systemic inflammatory response associated with *Fth* deletion in $T_{REG}$ cells from $Foxp3^{GFP\text{-}Fth\Delta/\Delta}$ mice (Fig. EV1D–G).

## FTH maintains $T_{REG}$ cell redox homeostasis via a cell-autonomous mechanism

We then asked whether FTH regulates gene expression in $T_{REG}$ cells, irrespective of the systemic inflammatory response associated with *Fth* deletion in $T_{REG}$ cells from $Foxp3^{GFP\text{-}Fth\Delta/\Delta}$ mice (Fig. EV1D–G). To test this hypothesis, the gene expression profile of $T_{REG}$ cells sorted from the LN of mixed BM chimeras (Fig. 4H) was compared to that of $T_{REG}$ cells sorted from the LN of non-chimeric mice (Fig. 2A). Analysis of RNAseq data from BM chimeric mice showed that *Fth*-deleted $T_{REG}$ cells (CD45.2$^+$GFP$^+$tdT$^+$) developing from the BM of $Foxp3^{GFP\text{-}Fth\Delta/\Delta\text{-}tdT}$ mice, upregulated 149 genes and down-regulated 96 genes, compared to *Fth*-competent $T_{REG}$ cells from control $Foxp3^{GFP\text{-}tdT}$ mice (Fig. 4I). In the same BM chimeric mice, *Fth*-deleted ex-$T_{REG}$ cells (CD45.2$^+$GFP$^-$tdT$^+$) developing from the BM of $Foxp3^{GFP\text{-}Fth\Delta/\Delta\text{-}tdT}$ mice upregulated 90 genes and down-regulated 36 genes, compared to *Fth*-competent ex-$T_{REG}$ cells from control $Foxp3^{GFP\text{-}tdT}$ mice (Fig. 4J). The genes regulated in $T_{REG}$ and ex-$T_{REG}$ cells originating from the BM of $Foxp3^{GFP\text{-}Fth\Delta/\Delta\text{-}tdT}$ vs. $Foxp3^{GFP\text{-}tdT}$ mice in BM chimeric mice, were associated with the oxidative stress response regulated by NRF2 (Fig. 4I,J). This suggests that FTH exerts cell-autonomous control of $T_{REG}$ cell redox homeostasis, irrespective of the systemic inflammatory response associated with *Fth* deletion in $T_{REG}$ cells from $Foxp3^{GFP\text{-}Fth\Delta/\Delta}$ mice (Fig. EV1D–G).

The inflammatory transcriptional signature of $T_{REG}$ cells from $Foxp3^{GFP\text{-}Fth\Delta/\Delta}$ vs. $Foxp3^{GFP}$ mice (Fig. 2B–D) was not observed in $T_{REG}$ cells from BM chimeric mice, originating from $Foxp3^{GFP\text{-}Fth\Delta/\Delta\text{-}tdT}$ vs. $Foxp3^{GFP\text{-}tdT}$ (Fig. 4I,J). Among the 245 differentially expressed genes in *Fth*-deficient (CD4$^+$GFP$^+$) $T_{REG}$ cells from BM chimeric mice (Fig. 4I,J), 134 matched those differentially expressed in *Fth*-deficient $T_{REG}$ cells from non-chimeric mice (Fig. 4K,L). Gene ontology analyzes of the overlapping genes, showed an enrichment for pathways related to oxidative stress response, comprising several NRF2-regulated genes (Fig. 4K–M). In contrast, *Fth*-deficient (CD4$^+$GFP$^+$) $T_{REG}$ cells from BM chimeric mice did not show an enrichment for pathways associated with $T_H$ cell activation (Fig. 4K–M). This suggests that FTH acts in a cell-autonomous manner to support $T_{REG}$ cell redox homeostasis and restrain $T_{REG}$ cell transdifferentiation into ex-$T_{REG}$ cells (Fig. 4). In contrast, the inflammatory profile associated with the transition of *Fth*-deleted $T_{REG}$ cells towards inflammatory ex-$T_{REG}$ cells, observed in $Foxp3^{GFP\text{-}Fth\Delta/\Delta}$ mice (Fig. 2) and $Foxp3^{GFP\text{-}Fth\Delta/\Delta\text{-}tdT}$ mice (Fig. 2) requires, in addition, the development of systemic inflammation.

## FTH acts in a cell-autonomous manner to support $T_{REG}$ cell homeostatic expansion

We took advantage of the homeostatic expansion of $T_{REG}$ cells, upon adoptive transfer into lymphopenic $Rag2^{-/-}$ mice (Duarte et al, 2009), to compare the proliferative capacity of CD4$^+$GFP$^+$tdT$^+$ $T_{REG}$ cells from $Foxp3^{GFP\text{-}Fth\Delta/\Delta\text{-}tdT}$ vs. control $Foxp3^{GFP\text{-}tdT}$ mice. The number of CD4$^+$GFP$^+$tdT$^+$ $T_{REG}$ cells recovered from the LN was markedly reduced when $Rag2^{-/-}$ mice received $T_{REG}$ cells from $Foxp3^{GFP\text{-}Fth\Delta/\Delta\text{-}tdT}$ vs. $Foxp3^{GFP\text{-}tdT}$ mice (Fig. 5A–C). While there were no differences in the number of CD4$^+$GFP$^-$tdT$^+$ ex-$T_{REG}$, the ratio of $T_{REG}$ over tdT$^+$ cells (GFP$^-$tdT$^+$/TdT +) were higher in $Rag2^{-/-}$ mice receiving $T_{REG}$ cells from $Foxp3^{GFP\text{-}Fth\Delta/\Delta\text{-}tdT}$ vs. $Foxp3^{GFP\text{-}tdT}$ mice, albeit without statistical significance (Fig. 5A–C). *Fth* expression in CD4$^+$GFP$^+$tdT$^+$ $T_{REG}$ cells used in the adoptive transfer was confirmed by qRT-PCR (Fig. EV4D,E).

We considered the possibility of an increase in the ratio of ex-$T_{REG}$ to $T_{REG}$ cells associated with *Fth* deletion in $T_{REG}$ cells reflecting, to some extent, a $T_{REG}$ cell survival defect. Therefore, we asked whether FTH acts in a cell-autonomous manner to support $T_{REG}$ viability and proliferation in vitro. The frequency and number of induced $T_{REG}$ (i$T_{REG}$) cells generated from $T_N$ cells activated in vitro with anti-CD3/CD28 mAb plus IL-2 and TGFβ, were indistinguishable regardless of

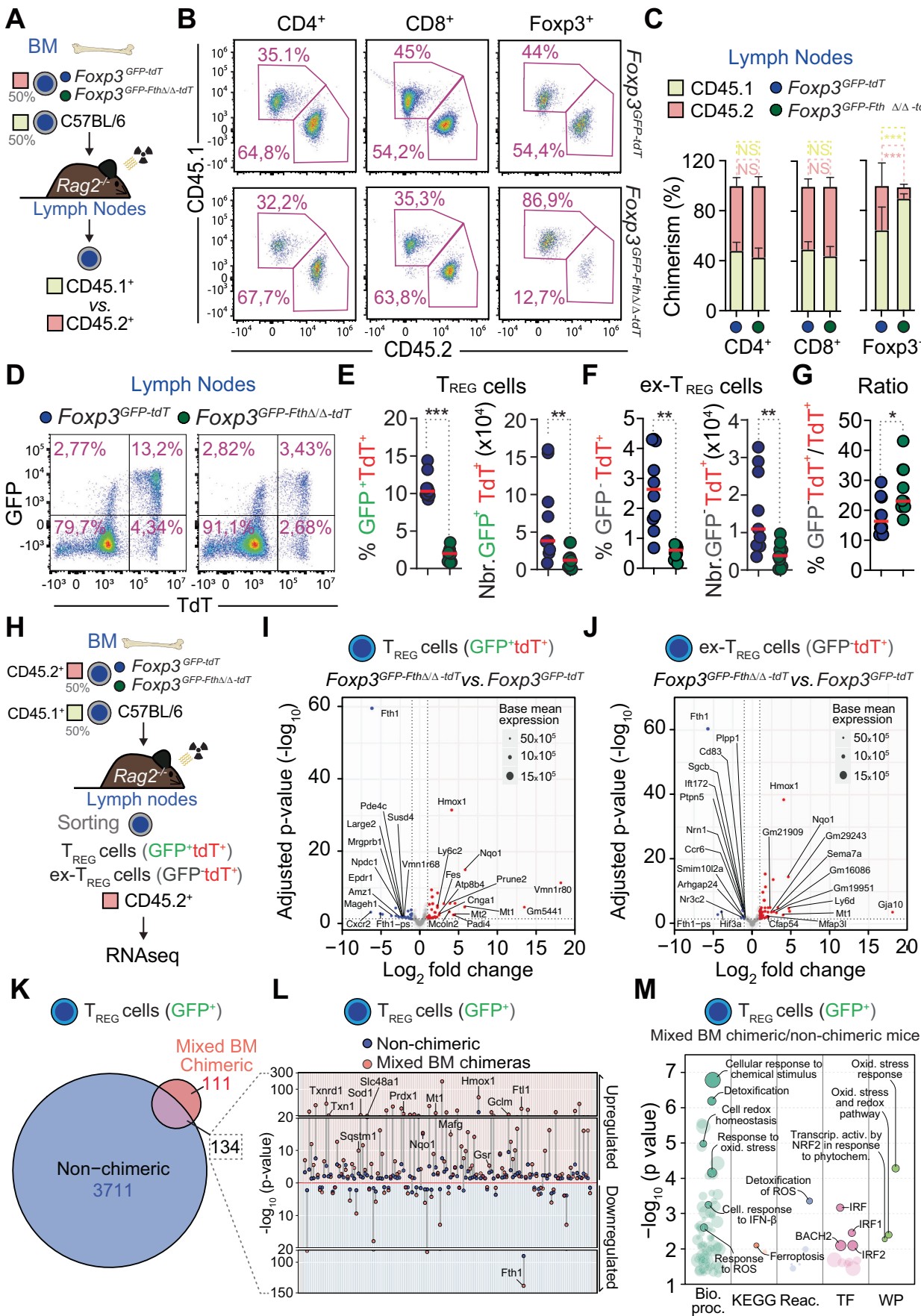

**Figure 4.  FTH enforces T$_{REG}$ cell lineage stability via a cell-autonomous mechanism.**

(A) Schematic representation of bone marrow (BM) chimeric mice used for flow cytometry analyzed 6 weeks after reconstitution. For reconstitution BM cells from C57BL/6 CD45.1$^+$ mice were mixed with BM cells from congenic C57BL/6 CD45.2$^+$ *Foxp3*$^{GFP-Fth\Delta/\Delta-tdT}$ or control *Foxp3*$^{GFP-tdT}$ mice at a 1:1 ratio and injected (i.v.; tail vein, 200 μL) into congenic C57BL/6-recipient *Rag2*-deficient (*Rag2*$^{-/-}$) female mice, 2 h after irradiation (600 Gys). (B) Representative flow cytometry dot plots and (C) Percentage of CD45.1$^+$ vs. CD45.2$^+$ CD4$^+$ T cells, CD8$^+$ T cells and CD4$^+$Foxp3$^+$ T$_{REG}$ cells in the lymph nodes of mixed BM chimeric mice, from (A). Data from $N = 10$ mice per genotype. (D–G) Representative flow cytometry dot plots (D) and corresponding quantification (E, F) of the percentage and number of (CD45.2$^+$CD4$^+$GFP$^+$tdT$^+$) T$_{REG}$ cells (E), (CD45.2$^+$CD4$^+$GFP$^-$tdT$^+$) ex-T$_{REG}$ cells (F) and relative proportion of ex-T$_{REG}$ cells over total CD45.2$^+$CD4$^+$tdT$^+$ cells in lymph nodes (LN) (G) of mixed BM chimeric mice (as in A). Data from $N = 9$–10 mice per genotype. (H) Schematic representation of (CD45.2$^+$CD4$^+$GFP$^+$tdT$^+$) T$_{REG}$ cells and (CD45.2$^+$CD4$^+$GFP$^-$tdT$^+$) ex-T$_{REG}$ cells FACS-sorting from the lymph nodes of mixed BM chimeric mice, used for RNA sequencing analysis. (I, J) Volcano plot representations of RNA sequencing data of genes overexpressed (red) or under-expressed (blue) in T$_{REG}$ cells (I) and ex-T$_{REG}$ cells (J) from lymph nodes of BM chimeric mice (as in H). Data from $N = 3$–4 mice per genotype. *P* values determined by Benjamini and Hochberg adjusted probabilities. (K) Euler plot of differentially regulated genes in (CD45.2$^+$CD4$^+$GFP$^+$) T$_{REG}$ cells sorted from the *Foxp3*$^{GFP-Fth\Delta/\Delta}$ or control *Foxp3*$^{GFP}$ T$_{REG}$ cells (non-chimeric; blue; from analysis described in Fig. 2A,B) or from the mixed BM chimeric mice (chimeric; red; from analysis illustrated in H–J). (L) Dumbbell plot, showing the adjusted *P* value of the 134 overlapping differentially regulated genes (from analysis described in K). Gray bars connecting dots represent the difference in *P* values for the differentially regulated genes (from analysis described in K). (M) Functional enrichment analysis of the overlapping genes ($N = 134$) (from analysis described in K), considering five different functional categories: biological processes (Bio. Proc.); KEGG (*Kyoto Encyclopedia of Genes and Genomes*) database; reactome (Reac.); transcription factors (TF); and WikiPathways (WP). Data were analyzed using g:SCS multiple testing correction method with a significance threshold of 0.05. Data information: Data in (C) are represented as mean ± SD. Data in (E–G) are represented as mean, circles correspond to individual mice and red bars are mean values. *P* value in (C) was determined by two-way ANOVA with Sidak's multiple comparisons test and in (E–G) by unpaired *t* test with Welch's correction. *$P < 0.05$; **$P < 0.01$; ***$P < 0.001$. Source data are available online for this figure.

whether T$_N$ cells were sorted from *Foxp3*$^{GFP-Fth\Delta/\Delta}$ or *Foxp3*$^{GFP}$ mice (Fig. EV4F–I). There were also no changes in the iT$_{REG}$ cell proliferation (Fig. EV4J), suggesting that FTH controls T$_{REG}$ cells in vivo via a mechanism that acts beyond its cytoprotective effects (Berberat et al, 2003; Pham et al, 2004).

## FTH regulates T$_{REG}$ cell mitochondrial function and bioenergetics

T$_{REG}$ cells rely on a core metabolic program whereby mitochondrial oxidative phosphorylation is the major source of ATP (Angelin et al, 2017; Weinberg et al, 2019). The observation that FTH regulates mitochondrial integrity in parenchyma cells (Blankenhaus et al, 2019) suggested that FTH also regulates the mitochondrial integrity of T$_{REG}$ cells. However, the number of mitochondria per CD4$^+$GFP$^+$ T$_{REG}$ cells in the LN of *Foxp3*$^{GFP-Fth\Delta/\Delta}$ was similar to that of T$_{REG}$ cells from *Foxp3*$^{GFP}$ mice, corresponding to ~55 mitochondria per T$_{REG}$ cell (Fig. 5D). This suggests that FTH is not essential to maintain the mitochondrial integrity of T$_{REG}$ cells.

The mitochondrial membrane potential of *Fth*-deficient T$_{REG}$ cells from *Foxp3*$^{GFP-Fth\Delta/\Delta}$ mice was markedly reduced, compared to control T$_{REG}$ cells from *Foxp3*$^{GFP}$ mice (Fig. 5E). This suggests that FTH regulates the mitochondrial function of T$_{REG}$ cells.

We then tested whether FTH regulates the mitochondrial energetic capacity of T$_{REG}$ cells, by quantifying the relative increase in basal O$_2$ consumption rate (OCR), upon uncoupling of the mitochondrial respiratory chain by carbonyl cyanide 4-(trifluoromethoxy)phenylhydrazone (FCCP). Spare mitochondrial respiratory capacity was higher in (CD4$^+$GFP$^+$) T$_{REG}$ cells from *Foxp3*$^{GFP-Fth\Delta/\Delta}$ vs. *Foxp3*$^{GFP}$ mice (Fig. 5F,G). Basal OCR and mitochondrial ATP production were similar in T$_{REG}$ cells from *Foxp3*$^{GFP-Fth\Delta/\Delta}$ vs. *Foxp3*$^{GFP}$ mice, as assessed by the relative decrease of OCR upon ATP synthase inhibition by Oligomycin (Fig. 5F,G). Non-mitochondrial OCR was also similar in T$_{REG}$ cells from *Foxp3*$^{GFP-Fth\Delta/\Delta}$ vs. *Foxp3*$^{GFP}$ mice, as assessed by the inhibition of the electron transport chain complex-I and -III by Rotenone and Antimycin A, respectively (Fig. 5F,G).

The spare respiratory capacity of T$_{REG}$ cells from control *Foxp3*$^{GFP}$ mice was lower, compared to T$_{CONV}$ cells (Fig. EV5A,B). In contrast, the spare respiratory capacity of T$_{REG}$ cells from

*Foxp3*$^{GFP-Fth\Delta/\Delta}$ mice was similar to that of T$_{CONV}$ cells (Fig. EV5C,D), while the extracellular acidification rate (ECAR), reflecting the rate of glycolysis, was higher in T$_{REG}$ cells from *Foxp3*$^{GFP-Fth\Delta/\Delta}$ vs. *Foxp3*$^{GFP}$ mice (Fig. 5H). This suggests that FTH plays an essential role in supporting the metabolic and bioenergetic profile of T$_{REG}$ cells, favoring mitochondrial oxidative OXPHOS over glycolysis (Angelin et al, 2017).

To gain further insight regarding how FTH regulates T$_{REG}$ cell bioenergetics, we performed targeted metabolomics analysis. The intracellular concentration of different TCA cycle metabolites in splenic *Fth*-deficient T$_{REG}$ cells from *Foxp3*$^{GFP-Fth\Delta/\Delta}$ mice was similar to that of control T$_{REG}$ cells from *Foxp3*$^{GFP}$ mice (Fig. EV5E). However, the ratio of α-ketoglutarate and isocitrate intracellular concentrations was lower in splenic *Fth*-deficient T$_{REG}$ cells from *Foxp3*$^{GFP-Fth\Delta/\Delta}$ mice vs. control T$_{REG}$ cells from *Foxp3*$^{GFP}$ mice (Fig. 5I). In contrast, the ratio of intracellular α-ketoglutarate and glutamate concentration was similar (Fig. 5I). This suggests that FTH modulates the rate of isocitrate to α-ketoglutarate conversion (Fendt et al, 2013), catalyzed by isocitrate dehydrogenase (IDH) in the mitochondrial TCA cycle. This occurs, most likely, without interfering with glutaminolysis, whereby α-ketoglutarate is generated via the conversion of glutamine into glutamate, catalyzed by glutamate dehydrogenases and glutaminase, respectively.

In contrast to other intermediate TCA cycle metabolites, the intracellular concentration of lactate was lower in splenic *Fth*-deficient T$_{REG}$ cells from *Foxp3*$^{GFP-Fth\Delta/\Delta}$ mice vs. control T$_{REG}$ cells from *Foxp3*$^{GFP}$ mice (Fig. EV5E). This is consistent with FTH modulating the metabolic and bioenergetics profile of T$_{REG}$ cells, presumably favoring lactate production via glycolysis, similar to observed in hepatocytes (Weis et al, 2017).

## FTH regulates cytosine methylation in T$_{REG}$ cells

Maintenance of T$_{REG}$ cell lineage relies on sustained demethylation of cytosines at CpG-rich sequences in the *FOXP3* CNS1 and 2 (Ohkura et al, 2012; Zheng et al, 2010). Cytosine demethylation is catalyzed by the TET family of methylcytosine dioxygenases (Yue et al, 2019; Yue et al, 2016), via redox-based reactions that use Fe and α-ketoglutarate as an essential cofactor and obligate substrate, respectively (Kohli and Zhang, 2013; Pastor et al, 2013). Having

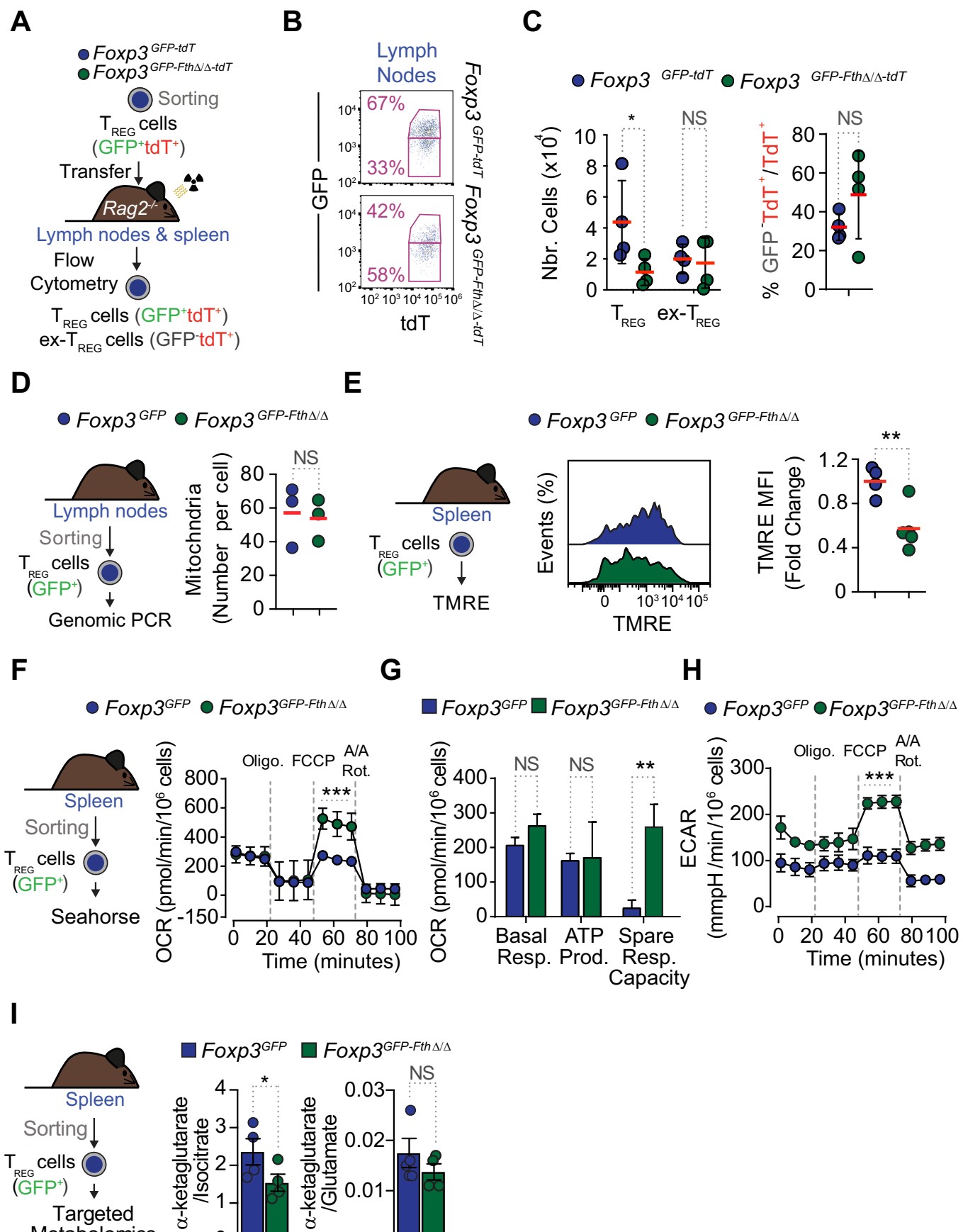

**Figure 5. FTH is a $T_{REG}$ cell-autonomous cytoprotectant.**

(A) Schematic representation of flow cytometry analyses of (CD4$^+$GFP$^+$tdT$^+$) $T_{REG}$ cells, isolated from the lymph nodes of lymphopenic $Rag2^{-/-}$ mice, 6 weeks after adoptive transfer of $T_{REG}$ cells sorted from $Foxp3^{GFP-FthΔ/Δ-tdT}$ vs. control $Foxp3^{GFP-tdT}$ mice. (B) Representative flow cytometry dot plots of (CD4$^+$GFP$^+$tdT$^+$) $T_{REG}$ cells and (CD4$^+$GFP$^-$tdT$^+$) ex-$T_{REG}$ cells, analyzed, as illustrated in (A). (C) Number (CD4$^+$GFP$^+$tdT$^+$) $T_{REG}$ cells and (CD4$^+$GFP$^-$tdT$^+$) ex-$T_{REG}$ (left panel), as well as the relative proportion of ex-$T_{REG}$ cells over total CD4$^+$tdT$^+$ cells (right panel), 6 weeks after adoptive transfer into $Rag2^{-/-}$ mice, as illustrated in (A). Data from $N = 4$ per genotype, in one out of two independent experiments with similar trend. (D) Schematic representation of the experimental approach (left panel) and number (Nbr.) of mitochondria in (CD4$^+$GFP$^+$) $T_{REG}$ cells sorted from lymph nodes of the $Foxp3^{GFP-FthΔ/Δ}$ or control $Foxp3^{GFP}$ mice (right panel), quantified by genomic quantitative PCR. Data from $N = 3$ mice per genotype in one experiment. (E) Schematic representation of the experimental approach used to quantify mitochondrial membrane potential in splenic mouse (CD4$^+$GFP$^+$) $T_{REG}$ cells (left panel). Representative flow cytometry histograms of tetramethylrhodamine ethyl ester (TMRE) staining (middle panels). Mean fluorescence intensity (MFI) of TMRE (right panel). Data from $N = 4$–5 mice per genotype, pooled from two independent experiments, with similar trend. (F) Schematic representation of experimental approach (left panel) used to quantify oxygen consumption rate (OCR) in live splenic (CD4$^+$GFP$^+$) $T_{REG}$ cells (right panel). Data pooled from $N = 3$ mice per genotype, represented as mean ± SD ($N = 3$–5 technical replicates) in one out of three independent experiments, with similar trend. Oligomycin (Oligo.), carbonilcyanide p-triflouromethoxyphenylhydrazone (FCCP), Antimycin A/Rotenone (A/A+Rot.). (G) Quantification of basal respiration, ATP production and spare respiratory capacity, from data represented in (F). (H) Extracellular acidification rate (ECAR) in live splenic (CD4$^+$GFP$^+$) $T_{REG}$ cells represented as mean ± SD ($N = 3$–5 technical replicates) in one out of three independent experiments, with similar trend. (I) Schematic representation of the experimental approach (left panel), used to quantify the ratio of α-ketoglutarate to isocitrate and α-ketoglutarate to glutamate (right panels) in live splenic (CD4$^+$GFP$^+$) $T_{REG}$ cells. Data from $N = 4$ mice per genotype in one out of two independent experiments with similar trend. Data information: Data in (C–E) circles correspond to individual value and red bars to mean values. Data are presented as mean ± SD. Data in (F, H) are presented as mean ± SD, circles correspond to technical replicates. Data in (G) are presented as mean ± SD of technical replicates. Data in (I) are presented as mean ± SD, circles correspond to individual values. $P$ values in (C (left panel), G) were calculated using two-way ANOVA with Sidak's multiple comparison test, $P$ values in (F, H) were calculated using two-way ANOVA with Bonferroni's multiple comparisons test. $P$ values in (C, right panel), D, E, I) were calculated using unpaired $t$ test with Welch's correction. NS not significant ($P > 0.05$), *$P < 0.05$; **$P < 0.01$; ***$P < 0.001$. Source data are available online for this figure.

established that FTH regulates intracellular catalytic Fe$^{2+}$, cellular redox homeostasis and possibly the rate of α-ketoglutarate to isocitrate conversion, we asked whether FTH modulates cytosine demethylation in $T_{REG}$ cells. To test this hypothesis, we performed a genome-wide methyl-sequencing (EM-seq) profiling of CD45.2$^+$CD4$^+$tdT$^+$ cells, which include (GFP$^+$) $T_{REG}$ and (GFP$^-$) ex-$T_{REG}$ cells sorted from the LN of mixed BM chimeras, reconstituted with CD45.1$^+$ (50%) BM cells plus CD45.2$^+$ (50%) BM cells from $Foxp3^{GFP-FthΔ/Δ-tdT}$ or control $Foxp3^{GFP-tdT}$ mice (Fig. 6A). The Methylome of CD45.2$^+$CD4$^+$tdT$^+$ cells, originating from the BM of $Foxp3^{GFP-FthΔ/Δ-tdT}$ and control $Foxp3^{GFP-tdT}$ mice clustered independently, as assessed by principal component analysis (Fig. 6B), revealing that FTH regulates the Methylome of $T_{REG}$ and ex-$T_{REG}$ cells.

*Fth*-deficient CD45.2$^+$CD4$^+$tdT$^+$ cells, originating from the BM of $Foxp3^{GFP-FthΔ/Δ-tdT}$ mice, presented a discrete number of hyper-methylated and hypomethylated CpG-rich sequences, compared to CD45.2$^+$CD4$^+$tdT$^+$ cells originating from the BM of control $Foxp3^{GFP-tdT}$ mice (Fig. 6C). These hyper and hypomethylated regions were located primarily in the promoter and intergenic regions and only to a lesser extent in introns and exons (Fig. 6D), from different chromosomes (Fig. 6E). This suggests that FTH regulates cytosine methylation in a mixed population of $T_{REG}$ and ex-$T_{REG}$ cells, via a cell-autonomous mechanism.

## FTH regulates *FOXP3* CNS1 and CNS2 methylation

Having established that FTH modulates the methylome of $T_{REG}$ and ex-$T_{REG}$ cells, we asked whether this was associated with changes in the methylation of CpG-rich sequences at the *FOXP3* CNS1 and CNS2 (Fig. 6F), controlling $T_{REG}$ cell lineage stability (Yue et al, 2019; Yue et al, 2016). Analyzes of the EM-seq data from mixed BM chimeric mice showed sustained hypermethylation of CpG-rich sequences in the *FOXP3* CNS1 and CNS2 from splenic $T_{REG}$ and ex-$T_{REG}$ cells originating from $Foxp3^{GFP-FthΔ/Δ-tdT}$ vs. control $Foxp3^{GFP-tdT}$ mice (Fig. 6G). This suggests that FTH acts in a cell-autonomous manner to sustain cytosine demethylation at *FOXP3*

CNS1 and CNS2 and presumably therefore regulate $T_{REG}$ cell lineage maintenance.

## The ferroxidase activity of FTH controls TET dioxygenase activity

We then questioned whether FTH modulates TET activity and tested this hypothesis in human HEK293 cells, transiently co-transfected with human TET3 plus FTH or an FTH mutant (FTH$^{mut}$) lacking ferroxidase activity (Broxmeyer et al, 1991). TET3 enzymatic activity was reduced in cells co-transfected with the FTH$^{mut}$, compared to control cells co-transfected with an empty vector (Fig. 6H). In contrast, FTH overexpression did not alter TET3 enzymatic activity, compared to controls. This suggests that FTH$^{mut}$, possibly acts as a dominant negative mutant failing to regulate catalytic Fe and likely therefore impairing TET activity. Expression of FTH or FTH$^{mut}$ did not alter the level of TET3 protein expression, as assessed by western blot (Fig. 6I). Taken together, these observations establish, as a proof of principle, that FTH ferroxidase activity regulates TET methylcytosine dioxygenase activity.

## FTH is required to sustain Foxp3 transcription and expression

Having established a functional link between FTH ferroxidase activity and TET methylcytosine dioxygenase activity, we questioned whether FTH regulates FOXP3 transcription and expression. Suppression of FTH expression in human $T_{REG}$ cells transduced with shRNAs targeting FTH, was associated with a reduction of FOXP3 protein expression, compared to control $T_{REG}$ cells transduced with non-targeting shRNA (Fig. 7A,B). This suggests that FTH acts in a cell-intrinsic manner to regulate the expression of FOXP3 in human $T_{REG}$ cells.

Having established that regulation of Fe metabolism by FTH regulates *Foxp3* transcription in the thymus as well as in circulating $T_{REG}$ cells (Fig. 1K) we compared the relative level of *Foxp3*-GFP expression in Nrp1$^+$ t$T_{REG}$ and Nrp1$^-$ p$T_{REG}$ cells. GFP expression

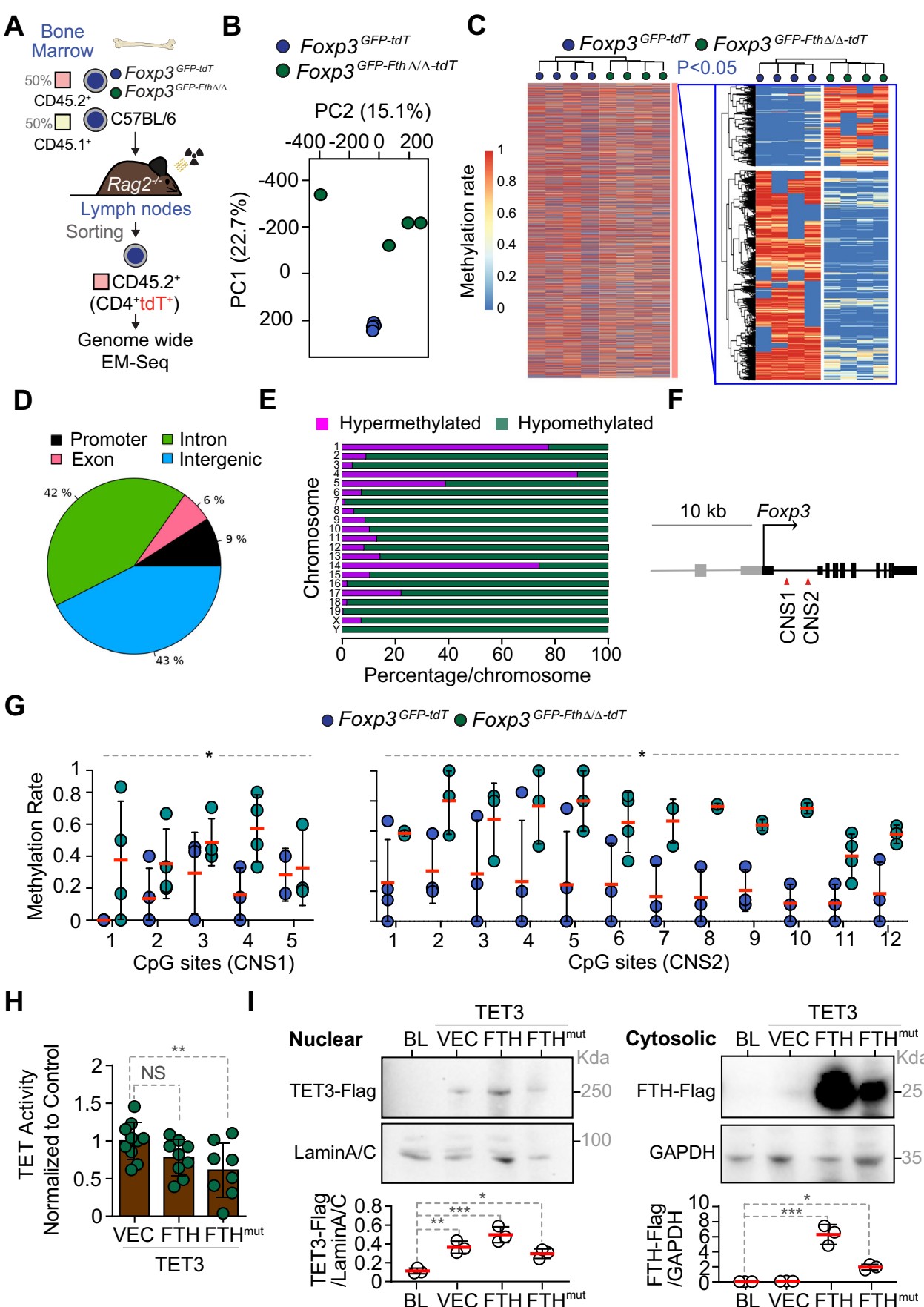

**Figure 6.  FTH regulates mitochondrial function and cytosine methylation in T_{REG} cells.**

(A) Schematic representation of mixed bone marrow (BM) chimeric mice from which lymph node CD45.2+CD4+tdT+ cells (i.e., GFP+ T_{REG} cells and GFP− ex-T_{REG} cells) were sorted for genome-wide EM-seq analyses. (B) Principal component analysis (PCA) of the methylome of lymph node T_{REG} and ex-T_{REG} cells in BM chimeric mice generated, as illustrated in (A). Data from $N = 4$ mice per genotype is represented as individual circles in one experiment. (C) Unsupervised heatmap representation of genome-wide EM-seq analyzes (i.e., 5-hmC) in CD45.2+CD4+tdT+ cells sorted from the mixed BM chimeric mice illustrated in (A). Statistical analysis for multiple testing correction was performed with Sliding Linear Model (SLIM). (D) The relative percentage of total methylated regions (i.e., hyper- and hypomethylated regions) ($q < 0.01$ and methylation difference >=10%) according to different genome regions (promoter, exon, intron and intergenic). (E) The number of hyper- and hypomethylation events (10% change in methylation and a $q$ value of 1%) per chromosome, shown as a percent of the differential sites. (F) Schematic representation of the *Foxp3* in enhancer regions CNS1 and CNS2. (G) Relative quantification of the methylation rate of CpG sequences in the *Foxp3* CNS1 (left panel) and CNS2 (right panel) from CD45.2+CD4+tdT+ cells sorted from the lymph nodes of BM chimeric mice, illustrated in (A). Data from $N = 4$ mice per genotype in one experiment. (H) TET activity in nuclear extracts from HEK293T cells transiently transfected with human TET3-flag, FTH-flag, or FTH^{mut}-flag cDNAs. Data shows technical replicates, pooled from four independent experiments. (I) FTH-flag, FTH^{mut}-flag, TET3-flag, Lamin A/C and GAPDH protein expression, detected by western blot in nuclear and cytosol extracts from HEK293T cells transfected as described in (H). Relative quantification of TET3-flag, normalized to Lamin A/C (bottom left panel), and FTH-flag, normalized to GAPDH (bottom right panel). Data from one experiment, representative of three independent experiments with similar trend. Data information: Data in (G) are presented as mean ± SD, circles correspond to individual mice and red bars to mean values. Data in (H, I) are presented as mean ± SD, circles correspond to technical replicates. *P* values in (G) were calculated using Two-way ANOVA with Sidak's multiple comparison test, and in (H, I) using one-way ANOVA using with Sidak's multiple comparison test. NS not significant ($P > 0.05$), *$P < 0.05$; **$P < 0.01$, ***$P < 0.001$. Source data are available online for this figure.

was reduced in both tT_{REG} and pT_{REG} cells, as assessed in the MLN from *Foxp3^{GFP-FthΔ/Δ}* vs. control *Foxp3^{GFP}* mice (Fig. 7C). Considering that GFP is expressed under the control of a bacterial artificial chromosome (BAC) transgene carrying the intact *Foxp3* promoter (Chen et al, 2003; Zhou et al, 2008), these observations suggest that FTH is essential to sustain *Foxp3* transcription in tT_{REG} and pT_{REG} cells.

The relative level of Foxp3 protein expression was also reduced when *Fth* was deleted, as assessed in the spleen and MLN from *Foxp3^{GFP-FthΔ/Δ}* vs. control *Foxp3^{GFP}* mice (Fig. 7D). This was also observed in BM chimeric mice, that is, the relative levels of Foxp3 protein expression were lower in *Fth*-deficient GFP+tdT+ cells (T_{REG} cells) originating from *Foxp3^{GFP-FthΔ/Δ-tdT}* vs. control *Foxp3^{GFP-tdT}* mice (Fig. 7E). Moreover, GFP expression was lower in *Fth*-deficient tdT+ cells (ex-T_{REG}+T_{REG} cells) originating from *Foxp3^{GFP-FthΔ/Δ-tdT}* vs. control *Foxp3^{GFP-tdT}* mice (Fig. 7F). This suggests that FTH acts in a cell-intrinsic manner to regulate Foxp3 transcription and expression in mouse T_{REG} cells.

To establish whether FTH regulates endogenous *Foxp3* transcription, we asked whether *Fth* deletion was associated with a concomitant reduction of *Foxp3* and *Gfp* mRNA expression. The relative level of *Foxp3* (Fig. 7G) and *Gfp* (Fig. 7H) mRNA expression were lower in *Fth*-deficient CD4+tdT+ cells, including T_{REG} (GFP+tdT+) and ex-T_{REG} (GFP−tdT+) cells, from *Foxp3^{GFP-FthΔ/Δ-tdT}* vs. control *Foxp3^{GFP-tdT}* mice. *Fth* deletion in CD4+tdT+ cells from *Foxp3^{GFP-FthΔ/Δ-tdT}* vs. *Foxp3^{GFP-tdT}* mice was confirmed by qRT-PCR (Fig. 7H). While *tdT* mRNA expression was not altered (Fig. 7H). These observations suggest that FTH is essential to sustain *FOXP3* transcription.

### FTH expression in T_{REG} cells limits the pathologic outcome of autoimmune neuroinflammation

Given the central role of T_{REG} cells in preventing the onset of autoimmune diseases (Kohm et al, 2002; Lafaille et al, 1994), we tested whether FTH expression in T_{REG} cells impacts on the pathogenesis of experimental autoimmune encephalomyelitis (EAE) (Fig. 8A). *Foxp3^{GFP-FthΔ/Δ}* mice had an increase in EAE incidence (Fig. 8B) and severity (Fig. 8C), in response to immunization with myelin oligodendrocyte glycoprotein (MOG)-derived peptide 35-55 (MOG_{35-55}) emulsified in complete Freund's adjuvant compared to control immunized *Fth^{fl/fl}* mice. Of note, the immunization protocol

used was "sub-optimal", as suggested by the relatively low disease scores in control immunized *Fth^{fl/fl}* mice, likely favoring the increase in EAE severity observed in *Foxp3^{GFP-FthΔ/Δ}* mice.

The frequency of T_H cells expressing IFNγ, IL-17A and IL17+IFNγ+ T_H cells was higher in the spinal cord of MOG_{35-55}-immunized *Foxp3^{GFP-FthΔ/Δ}* vs. control *Fth^{fl/fl}* mice (Fig. 8D). This suggests that FTH expression in T_{REG} cells limits the activation, proliferation and/or infiltration of self-reactive T_H type 1 (T_H1) and T_H type 17 (T_H17) cells as well as pathogenic IL17+IFNγ+ T_H cells (Duhen et al, 2013) into the central nervous system, in response to MOG_{35-55} immunization.

### FTH expression in T_{REG} cells limits malaria severity

T_{REG} cells constrain the extent of immunopathology associated with infectious diseases (Arpaia et al, 2015), supporting the hypothesis that regulation of T_{REG} cell lineage stability by FTH impacts on the severity of infectious diseases (Soares et al, 2017). We tested this hypothesis for severe malaria, an often-lethal outcome of *Plasmodium spp.* infection. *Foxp3^{GFP-FthΔ/Δ}* mice succumbed to *Plasmodium chabaudi chabaudi* (*Pcc*) AS infection, as compared to *Foxp3^{GFP}* mice that survived and cleared the parasite (Fig. 8E). However, the number of circulating infected red blood cells (i.e., parasite burden) was lower at the peak of infection, in *Foxp3^{GFP-FthΔ/Δ}* vs. control *Foxp3^{GFP}* mice (Fig. 8E). The lethal outcome of *Pcc* infection in *Foxp3^{GFP-FthΔ/Δ}* mice was associated with a reduction in the frequency (but not the number) of splenic T_{REG} cells (Fig. 8F), as well as with an increase in the number (but not the frequency) of IFNγ-expressing T_H cells in the spleen (Fig. 8G). This suggests that FTH expression in T_{REG} cells is essential to restrain immune-driven pathology underlying the development of severe presentation of malaria, emphasizing the critical involvement of T_{REG} cells in the control of the pathogenesis of severe malaria (Walther et al, 2005). Moreover, these findings illustrate how dysregulation of Fe metabolism promotes the pathogenesis of severe malaria (Ferreira et al, 2008; Gozzelino et al, 2012; Ramos et al, 2019; Seixas et al, 2009; Wu et al, 2023).

### FTH expression in T_{REG} cells favors tumor progression

T_{REG} cells are pathogenic, for example, when restraining anti-tumor immunity (Curiel et al, 2004; Liu et al, 2016), suggesting that FTH

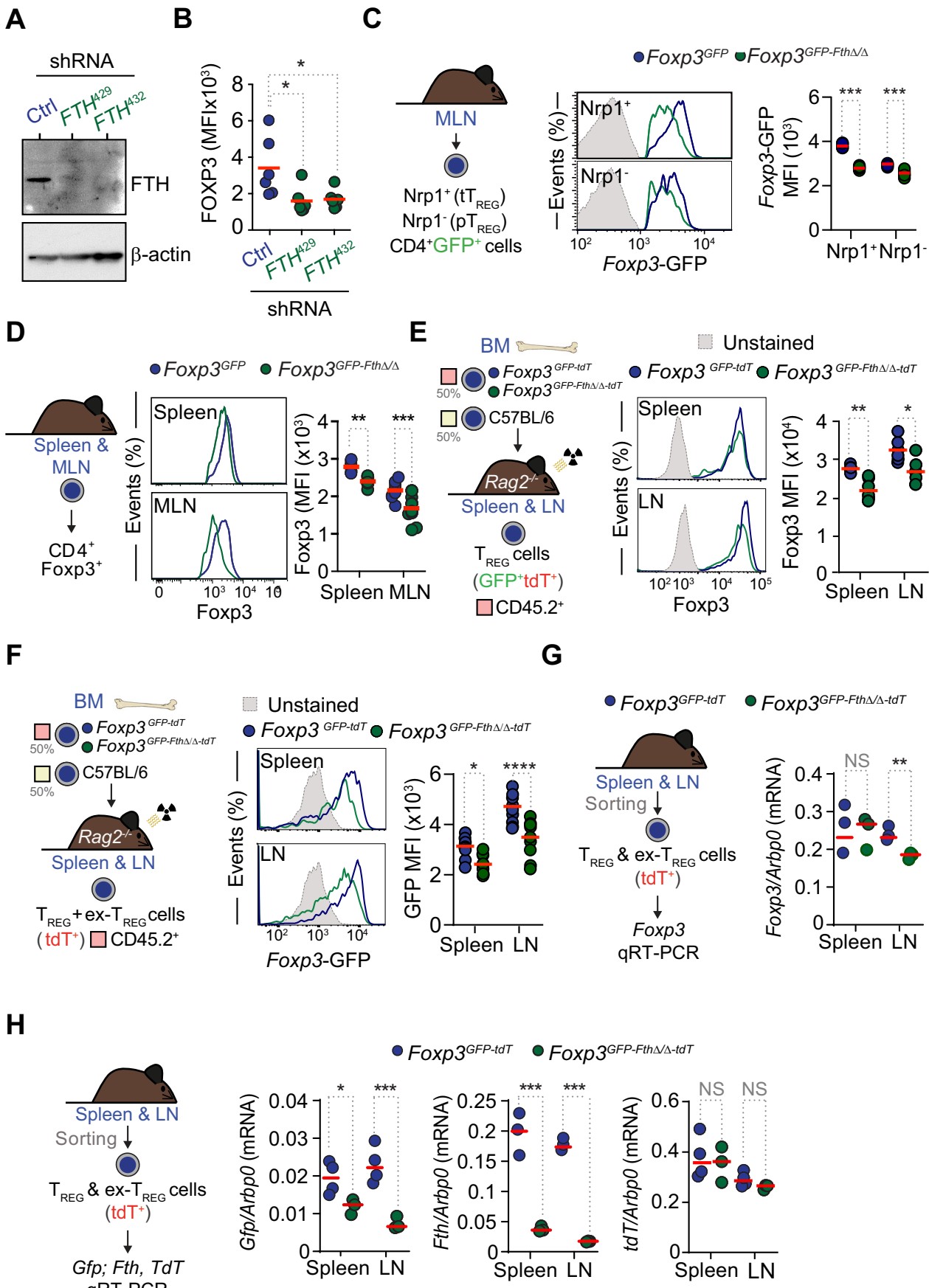

**Figure 7.  FTH is required to sustain Foxp3 transcription and expression.**

(A) FTH protein detected by western blot in whole-cell extracts from HEK293T cells infected with recombinant lentiviruses coding shRNAs targeting *FTH* (*FTH*[429] and *FTH*[432]) or control (Ctrl.) recombinant lentiviruses non-targeting shRNA. (B) Mean fluorescence intensity (MFI) of FOXP3 expression, detected by flow cytometry in human (CD4$^+$CD45RA$^+$CD25$^+$) T$_{REG}$ cells infected with the same recombinant lentiviruses as in (A). Data from $N = 6$ samples per experimental group. (C) Schematic representation of the experimental approach (left panel) used to monitor GFP transgene expression in the mesenteric LN (MLN) of (CD4$^+$GFP$^+$Nrp1$^+$) tT$_{REG}$ cells and (CD4$^+$GFP$^+$Nrp1$^-$) pT$_{REG}$ cells. Representative flow cytometry histogram (middle panel) and quantification of relative GFP expression (right panel), shown as mean fluorescence intensity (MFI). Data from $N = 8$ mice per genotype, pooled from two independent experiments with similar trend. (D) Schematic representation of the experimental approach (left panel) used to monitor Foxp3 expression by flow cytometry in mouse spleen and MLN (CD4$^+$Foxp3$^+$) T$_{REG}$ cells. Representative flow cytometry staining of Foxp3 (middle panel). Relative quantification of Foxp3 expression (right panel), represented as mean of fluorescence intensity (MFI). Data from $N = 9$ mice per genotype, pooled from two to three independent experiments with similar trend. (E) Schematic representation of the experimental approach (left panel) used to monitor Foxp3 expression in (CD45.2$^+$CD4$^+$GFP$^+$tdT$^+$) T$_{REG}$ cells isolated from the spleen and LN of BM chimeras. Representative flow cytometry of Foxp3 staining (middle panel). Relative quantification of Foxp3 expression (right panel), shown as mean fluorescence intensity (MFI). Data in (E) from $N = 5$–6 mice per genotype, representative of two independent experiments with similar trend. (F) Schematic representation of the experimental approach (left panel) used to monitor GFP expression in the spleen and LN of CD45.2$^+$CD4$^+$tdT$^+$ cells (T$_{REG}$+ ex-T$_{REG}$) from BM chimeras. Representative flow cytometry of GFP (F) staining (middle panel). Relative quantification of GFP expression (right panel), represented as mean fluorescence intensity (MFI). Data from $N = 10$ mice per genotype, pooled from two independent experiments with similar trend. (G) Schematic representation of the experimental approach (left panel), where (CD4$^+$tdT$^+$) cells were sorted from the spleen and LN for qRT-PCR (G, left panel). Relative expression of *Foxp3* (right panel). (H) *Gfp, Fth,* and *tdT* mRNA expression normalized to *Arbp0* of cells sorted as in (G). Data in (G, H) from $N = 3$–4 mice per genotype from one experiment. Data information: Data in (B) are presented as mean ± SD, circles correspond to individual wells and red bars to mean values. Data in (C–H) are presented as mean ± SD, circles correspond to individual mice and red bars to mean values. *P* values in (B) were calculated using the Fiedman test with Dunn's multiple comparison test, in (C–H) using two-way ANOVA with Sidak's multiple comparison test. NS not significant ($P > 0.05$), *$P < 0.05$; **$P < 0.01$; ***$P < 0.001$; ****$P < 0.0001$. Source data are available online for this figure.

expression in T$_{REG}$ cells favors tumor progression. In support of this hypothesis, the relative growth of syngeneic B16 melanoma cells was reduced in *Foxp3*$^{GFP\text{-}Fth\Delta/\Delta}$ vs. control *Fth*$^{fl/fl}$ mice (Fig. 8H). This was associated with lower frequency of tumor-infiltrating T$_{REG}$ cells (Fig. 8I) and higher frequency of activated Foxp3$^-$CD4$^+$CD25$^+$ effector T$_H$ cells (Fig. 8J). The frequency and number of activated CD4$^+$IFNγ$^+$ T$_H$ cells isolated from B16 melanomas was similar in *Foxp3*$^{GFP\text{-}Fth\Delta/\Delta}$ vs. control *Foxp3*$^{GFP}$ mice (Fig. EV5F). In contrast, the number of activated CD8$^+$IFNγ$^+$ T$_C$ cells was higher in B16 melanomas from *Foxp3*$^{GFP\text{-}Fth\Delta/\Delta}$ vs. control *Foxp3*$^{GFP}$ mice (Fig. EV5F). This tendency was also observed for CD8$^+$granzymeB$^+$ T$_C$ cells, *albeit* not statistically significant (Fig. EV5G). This suggests that regulation of Fe metabolism by FTH in T$_{REG}$ cells supports tumor progression, via a mechanism that hinders anti-tumor immunity.

# Discussion

T$_{REG}$ cells respond to variations in the relative levels of nutrients, vitamins and metabolites in their environment (Chapman et al, 2020; Shi and Chi, 2019), via dedicated transporter-receptor coupled sensors that modulate T$_{REG}$ cell function and lineage stability (Kempkes et al, 2019; Shi and Chi, 2019). In keeping with Fe-regulatory genes being a core property of T$_{REG}$ cells (Cuadrado et al, 2018), we found that the Fe-regulatory protein FTH is essential to support T$_{REG}$ cell lineage maintenance in vivo (Figs. 1–4) and support immune homeostasis (Figs. 2, EV1D–G, and EV2C).

FTH regulates T$_{REG}$ cell lineage stability (Figs. 3 and 4A–G) without interfering with the antiproliferative function of T$_{REG}$ cells (Fig. EV2A). Instead, FTH targets the intracellular pool of redox-active Fe$^{2+}$ (Fig. EV1B) to regulate T$_{REG}$ cell redox homeostasis (Figs. 2F and 4K–M) in a manner that controls T$_{REG}$ cell: (i) energy metabolism (Figs. 5F–I and EV5A–E), (ii) TET activity (Fig. 6H), (iii) cytosine demethylation at CpG-rich sequences at CNS1 and CNS2 in the *FOXP3* locus (Fig. 6G), and (iv) FOXP3 transcription/ expression (Figs. 1K,L and 7F–H). As the latter is essential to maintain the transcriptional program supporting T$_{REG}$ cell lineage stability (Williams and Rudensky, 2007) (Nakatsukasa et al, 2019;

Ohkura et al, 2012; Yue et al, 2019) *Fth* deletion is associated with a decrease of T$_{REG}$ cells, including tT$_{REG}$ cells and pT$_{REG}$ cells, without interfering with thymic T$_{REG}$ cell output (Fig. 1H,L). These observations are consistent with FTH acting upstream of TET methylcytosine dioxygenases, to control redox-based cytosine demethylation in T$_{REG}$ cells (Fig. 6) (Huang and Rao, 2014; Pastor et al, 2013).

That FTH targets the intracellular pool of redox-active Fe$^{2+}$ to control TET enzymatic activity is suggested by the observation that overexpression of a ferroxidase deficient FTH$^m$ compromised TET methylcytosine dioxygenase activity (Fig. 6H, I). Several non-mutually exclusive mechanisms might explain this observation. First, FTH could act "directly" via sequestration of catalytic Fe$^{2+}$ (Fig. EV1B), controlling the availability of this essential cofactor of TET enzymatic activity (Kohli and Zhang, 2013; Pastor et al, 2013). Second, FTH could act "indirectly" via the regulation of cellular redox homeostasis (Figs. 2D,F and 4K–M), preventing catalytic Fe$^{2+}$ from catalyzing oxidative stress, which compromises TET activity (Niu et al, 2015). Moreover, this should restrain NRF2 activation (Figs. 2D,F and 4K–M) from repressing *Foxp3* expression and impair T$_{REG}$ cell function (Klemm et al, 2020).

Consistent with our findings, intracellular Fe mobilization via lysosome-mediated ferritinophagy, was recently shown to regulate TET-driven (de)methylation of the peroxisome proliferator-activated receptor γ (PPARγ) locus, the master regulators of adipocyte development (Suzuki et al, 2023). This suggest that FTH controls Fe-responsive epigenetic programs defining different cellular developmental programs.

FTH regulates T$_{REG}$ cell mitochondrial TCA cycle and OXPHOS (Figs. 5F–I and EV5A–E), consistent with similar findings in other cell types (Blankenhaus et al, 2019; Oexle et al, 1999). While FTH does not modulate the intracellular concentration of intermediate TCA cycle metabolites (Fig. EV5E), including α-ketoglutarate (Fig. 5I), it does appear to modulate the rate of isocitrate conversion into α-ketoglutarate, likely acting irrespectively of α-ketoglutarate generation via glutaminolysis. It is possible therefore that FTH regulates the production of this obligatory substrate of TET cytosine dioxygenases (Kohli and Zhang, 2013; Pastor et al, 2013) via regulation

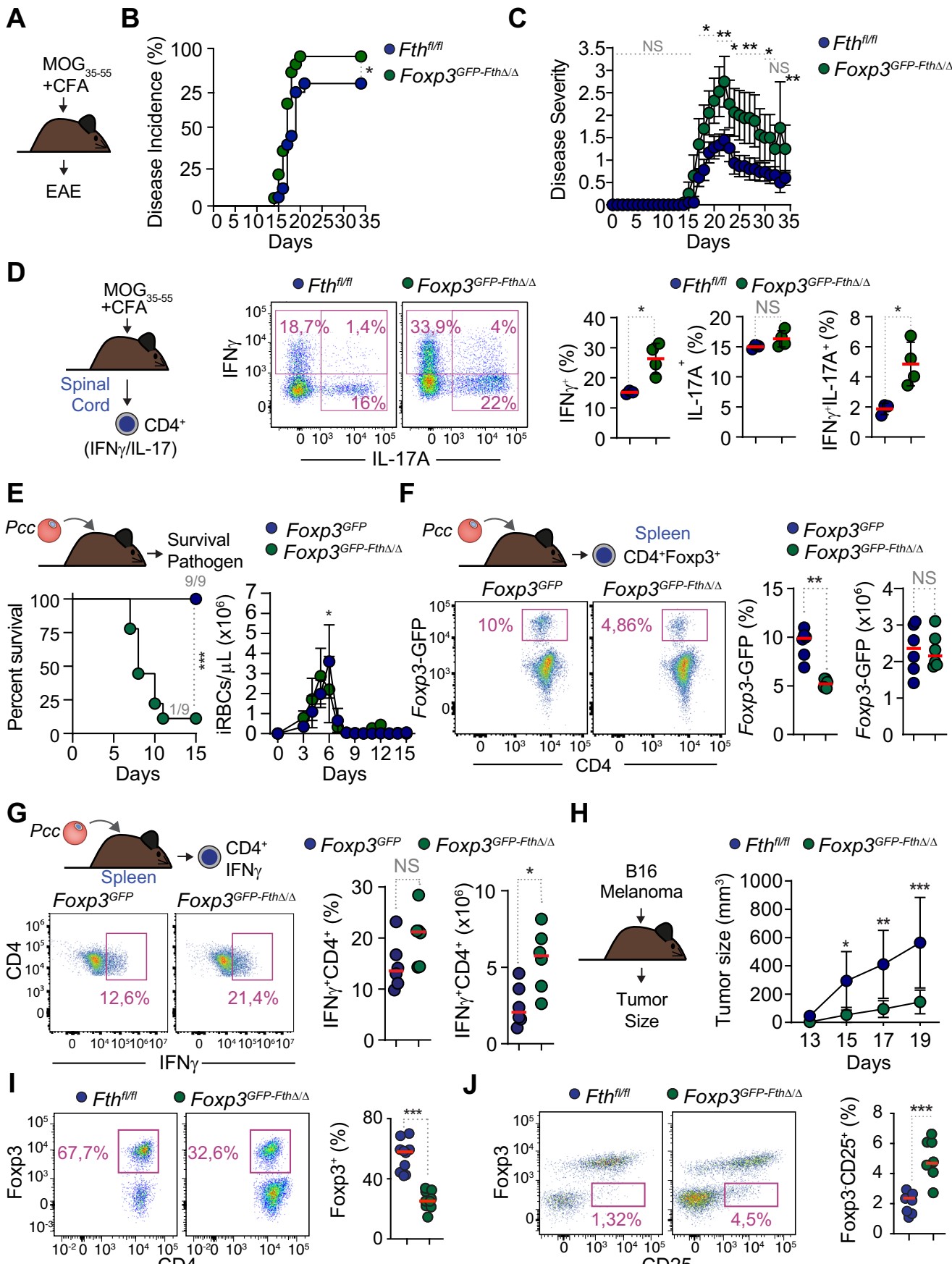

**Figure 8. FTH expression in T$_{REG}$ cells controls the pathologic outcome of experimental immune-driven inflammatory conditions.**

(**A**) Schematic representation of the induction experimental autoimmune encephalomyelitis (EAE) in response to MOG$_{35-55}$ immunization. (**B**) EAE incidence (percentage) and (**C**) EAE severity in MOG$_{35-55}$ immunized mice. Data from $N = 18$–$20$ mice per genotype, pooled from three independent experiments, with similar trend. (**D**) Experimental approach (right panel), representative flow cytometry dot plots (middle panel) and corresponding quantification (left panel) of the relative percentage of activated T$_H$1 (CD3$^+$CD4$^+$IFN-γ$^+$), T$_H$17 (CD3$^+$CD4$^+$IL-17A$^+$); and double positive IFN-γ$^+$IL-17A$^+$ T$_H$ cells in the spinal cord, 22 days after MOG$_{35-55}$ immunization. Data from $N = 3$–$4$ mice per genotype. (**E**) Survival (left panel) and number of circulating *Plasmodium chabaudi chabaudi* (*Pcc*)-infected red blood cells (iRBC) per μL of whole blood (i.e., parasite burden) (right panel). $N = 9$ mice per genotype, pooled from two independent experiments, with similar trend. (**F, G**) Representative flow cytometry dot plot (left panels) and corresponding percentage and cell numbers (right panels) of splenic (CD4$^+$Foxp3-GFP$^+$) T$_{REG}$ cells (**F**) and IFNγ$^+$CD4$^+$ activated T$_H$ cells (**G**), 7 days after *Pcc* infection. Data from $N = 8$–$9$ mice per genotype, pooled from two independent experiments, with similar trend. (**H**) Relative tumor (B16-F10-luc2) size, 13–19 days after inoculation ($2 \times 10^5$ cells). Data from $N = 7$–$11$ mice per genotype, pooled from 3 independent experiments, with similar trend. (**I, J**) Representative flow cytometry dot plots (left panels) and corresponding percentage (right panels) of live tumor-infiltrating (CD4$^+$Foxp3$^+$) T$_{REG}$ cells (**I**) and (CD4$^+$Foxp3$^-$CD25$^+$) effector T$_H$ cells (**J**), 3 weeks after tumor inoculation. $N = 7$ mice per genotype, pooled from three independent experiments, with similar trend). Data information: Circles in (**D, F, G, I, J**) correspond to individual mice and red bars to mean values. Data in (**C, H**) are presented as mean ± SEM. Data in (**E**, right panel) are presented as mean ± SD. *P* values in (**C**), (**E**, right panel), and (**H**) were determined using Holm–Sidak method (multiple *t* tests), with alpha = 0.05 under the assumption that both genotypes have similar SEM, in (**B, E**) by log-rank (Mantel–Cox) test, and in (**D, F, G, I, J**) by unpaired *t* test with Welch's correction. NS not significant (*P* > 0.05); *P < 0.05; **P < 0.01; ***P < 0.001. Source data are available online for this figure.

of the TCA cycle. This interpretation is consistent with other signal transduction pathways regulating the TCA cycle, such as the one triggered by the programmed cell death 1 ligand 2 (PD-L2), regulating cytosine methylation at CpG sequences in the T$_{REG}$-specific demethylation region, compromising Foxp3 stability in T$_{REG}$ cells (Hurrell et al, 2022).

The metabolic program orchestrated by FOXP3, supports T$_{REG}$ cell antiproliferative function and lineage stability, via a mechanism that relies on the suppression of c-Myc, a transcription factor that represses mitochondrial OXPHOS and promotes glycolysis (Angelin et al, 2017). Importantly, c-Myc can repress *FTH* transcription and translation in other cell types, the later occurring via a mechanism involving the Fe-regulatory protein 2 (IRP2) (Wu et al, 1999). This suggests that the FOXP3-driven genetic program encompasses the induction of *FTH* via a mechanism that could involve the repression of c-Myc. Moreover, the *FTH* promoter contains at least one Foxp3 DNA binding site (Appendix Figs. S3 and 4) and as such it is possible that Foxp3 would regulate *FTH* expression directly. This would argue for a positive feedback loop in which FTH enforces *Foxp3* expression, via the regulation of TET dioxygenases, and the later enforces the *FTH* expression transcriptionally. This hypothesis remains however to be tested experimentally.

While FTH prevents the transdifferentiation of T$_{REG}$ cells toward inflammatory ex-T$_{REG}$ cells (Figs. 3 and EV2C–E), via a non-cell-autonomous mechanism that relies on systemic inflammation (Fig. EV1D–G), this is not associated with the accumulation of inflammatory ex-T$_{REG}$ cells in *Foxp3*$^{GFP-FthΔ/Δ-tdT}$ vs. control *Foxp3*$^{GFP-tdT}$ mice (Fig. 3E–G). The same is true in mixed BM chimeric mice reconstituted with BM cells from *Foxp3*$^{GFP-FthΔ/Δ-tdT}$ vs. control *Foxp3*$^{GFP-tdT}$ mice (Fig. 4A–G). One possible interpretation is that FTH is essential not only to restrain T$_{REG}$ cells from transdifferentiating into inflammatory ex-T$_{REG}$ cells but also to support the viability of highly proliferating T$_{REG}$ and ex-T$_{REG}$ cells. This is consistent with the lack of autoimmune lesions in *Foxp3*$^{GFP-FthΔ/Δ}$ mice (Fig. EV2B). We note, however, that *Fth* deletion in T$_{REG}$ cells does not compromise the T$_{REG}$ cell thymic output (Fig. 1H) nor the generation and proliferation of iT$_{REG}$ cells in vitro (Fig. EV4F–J).

One cannot exclude that the highly methylated status of the *Foxp3* locus from ex-T$_{REG}$ cells (Komatsu et al, 2014; Miyao et al, 2012; Zhou et al, 2009) together with the increase frequency of ex-T$_{REG}$ cells among CD45.2$^+$CD4$^+$tdT$^+$ cells (Fig. 4A–G), contributes to the observed increase in the methylation of the *Foxp3* locus in

*Fth*-deleted T$_{REG}$ cells (Fig. 6A–G). This does not invalidate however, that *Fth* deletion acts in a cell-autonomous manner to decrease *Foxp3* transcription/expression (Figs. 1K,L and 7F–H), therefore increasing the frequency at which T$_{REG}$ cells transdifferentiate into ex-T$_{REG}$ cells (Fig. 4A–G).

In contrast with *genetic* deletion of *Tet2* and *Tet3* in T$_{REG}$ cells, which promotes the transdifferentiation of T$_{REG}$ cells into inflammatory ex-T$_{REG}$ cells and the development of autoimmunity (Nakatsukasa et al, 2019; Ohkura et al, 2012; Yue et al, 2019), *Fth* deletion in T$_{REG}$ cells is not associated with overt autoimmunity (Fig. EV2BG). This is consistent with FTH exerting additional effects, beyond the regulation of TET dioxygenases, preventing cellular stress from compromising the viability of the inflammatory ex-T$_{REG}$ cells that would otherwise elicit autoimmunity.

*Fth* deletion in T$_{REG}$ cells led to an increase in EAE susceptibility and severity (Fig. 8A–C), induced by an immunization protocol leading to low-grade disease severity in control mice (Fig. 8C). This was associated with a higher accumulation of activated T$_H$1 and T$_H$17 cells as well as pathogenic IFNγ$^+$IL-17A$^+$ T$_H$ cells (Duhen et al, 2013) in the central nervous system (Fig. 8D), likely originating from auto-reactive T$_N$ cells and/from ex-T$_{REG}$ cells that lost Foxp3 expression to become inflammatory and presumably pathogenic (Bailey-Bucktrout et al, 2013). This is consistent with regulation of Fe metabolism modulating the incidence and severity of autoimmune conditions, as demonstrated for systemic lupus erythematosus (Gao et al, 2022a). Consistent with our findings, this was linked to modulation of T$_{REG}$ (Gao et al, 2022a), T$_H$17 (Teh et al, 2021), and FT$_H$ (Gao et al, 2022b) cells and was associated with regulation of DNA demethylation (Gao et al, 2022b; Teh et al, 2021). However, whether regulation of Fe metabolism in T$_{REG}$ cells affects systemic lupus erythematosus was, to the best of our knowledge, not established (Gao et al, 2022a).

*Fth* deletion in T$_{REG}$ cells increased susceptibility to malaria (Fig. 8E), consistent with dysregulation of Fe metabolism promoting malaria lethality (Ferreira et al, 2008; Ramos et al, 2022; Ramos et al, 2019; Wu et al, 2023). This was associated with an increase in host-parasite burden (Fig. 8E), in keeping with T$_{REG}$ cells being essential to counter the pathogenesis of severe presentations of malaria while limiting immune-driven resistance mechanisms driving parasite clearance (Hisaeda et al, 2004; Kurup et al, 2017; Walther et al, 2005). We infer that the protective effect of T$_{REG}$ cells against malaria acts via a mechanism that is not associated with a reduction of the

host-pathogen burden, a defense strategy known as disease tolerance (Medzhitov et al, 2012; Soares et al, 2017). Presumably, the mechanism(s) via which FTH acts in $T_{REG}$ cells to establish disease tolerance to malaria is multifactorial, restraining unfettered immune activation to prevent the pathogenesis of severe forms of malaria.

Dysregulation of Fe metabolism in *Fth*-deleted $T_{REG}$ cells was associated with better control of tumor progression (Fig. 8H), consistent with a relative reduction of tumor-infiltrating $T_{REG}$ cells (Fig. 8I) and a more pronounced activation and/or infiltration of activated T effector cells (Fig. 8J), including $CD8^+IFN\gamma^+$ $T_C$ cells (Fig. EV5F). While the mechanism via which FTH expression in $T_{REG}$ cells promotes tumor progression is not clear, these observations are consistent with the regulation of Fe metabolism in the tumor microenvironment impacting on tumor progression (Alaluf et al, 2020; Consonni et al, 2021).

In conclusion, regulation of intracellular Fe metabolism by FTH is essential to maintain $T_{REG}$ cell lineage and function in vivo, reflecting how intracellular catalytic Fe controls the activity of TET dioxygenases that sustain FOXP3 transcription and support $T_{REG}$ cell lineage identity. Moreover, FTH might regulate other iron-dependent mechanisms supporting $T_{REG}$ function, for example, by enforcing the expression of c-Maf in $T_{REG}$ cells (Zhu et al, 2023) that control immunological tolerance to the microbiota (Xu et al, 2018). We propose that targeting Fe metabolism pharmacologically maybe considered when manipulating $T_{REG}$ cells for therapeutic purposes, either to enhance $T_{REG}$ cell function in the context of immune-mediated inflammatory diseases or to dampen $T_{REG}$ cell function as in the context of cancer therapies.

# Methods

### Reagents and tools table

| Reagent/resource | Reference or source | Identifier or catalog number |
|---|---|---|
| **Experimental models** | | |
| Human: HEK293T | ATCC | ATCC® CRL-3216™ |
| Mouse: Tumor cells B16-F10-luc2 (B16) | CaliperLS | B16-F10-luc2 |
| Mouse: B6.C57BL/6 Fth^fl/fl^ | Lukas Kuhn, ETH, Switzerland | (Darshan et al, 2009) |
| Mouse: B6129S-Tg(Foxp3 EGFP/icre)1aJbs/J backcrossed into B6.C57BL/6 background | Jackson Laboratory | JAX stock: 023161 |
| Mouse: B6.Cg-Gt(ROSA)26Sortm9(CAG-tdTomato)Hze/J | Jackson Laboratory | JAX stock: 007909 |
| Mouse: RAG2 -/- (B6.129S6-Rag2<tm1Fwa>N12) | Taconic | Taconic # RAGN12 |
| *Plasmodium chabaudi chabaudi* strains: PcAS clone AJ4916 | Reece & Thompson, 2008 | N/A |
| Blood samples from anonymized healthy male donors were obtained in accordance with guidelines established by the Sanquin Medical Ethical Committee. | This paper | NA |

| Reagent/resource | Reference or source | Identifier or catalog number |
|---|---|---|
| **Recombinant DNA** | | |
| psPAX | Addgene | Cat#12260 |
| pMD2.G | Addgene | Cat#12259 |
| pCMV-FTH-3tag3a | This paper | |
| pCMV-FTH^mut^-3tag3a | This paper | |
| pCMV-3×FLAG-TET3(human)-Neo | miaolingBio | P45302 |
| **Antibodies** | | |
| PE anti-human CD25 (Clone 2A3) | BD Biosciences | Cat#341011 (RRID: AB_2783790) |
| PE-Cy7 anti-human CD45RA (Clone HI100) | BD Biosciences | Cat#560675 (RRID: AB_1727498) |
| Brilliant Violet 421 Anti-human CD127 (Clone A019D5) | Biolegend | Cat#351309 (RRID: AB_10898326) |
| PE-Cy7 anti-Human FOXP3 (Clone 236A/E7) | eBioscience | Cat#25-4777-42 (RRID: AB_2573450) |
| Anti-Ferritin Heavy Chain | Abcam | ab65080 (RRID:AB_10564857) |
| CD45 APC-eFluor780 | eBioscience | 30-F11, 47-0451-82 (RRID:AB_1548781) |
| TCR-β BV421 | BioLegend | H57-597, 109229 (RRID:AB_10933263) |
| TCR-β BV711 | BioLegend | H57-597, 109243 (RRID:AB_2629564) |
| CD4 PE-Cy7 | eBioscience | RM4-5, 25-0042-82 (RRID:AB_469578) |
| CD4 APC-eFluor780 | eBioscience | GK1.5, 47-0041-82 (RRID:AB_11218896) |
| CD4 BV421 | BioLegend | GK1.5, 100438 (RRID:AB_10900241) |
| CD8 PercpCy5.5 | eBioscience | 53-6.7, 45-0081-82 (RRID:AB_1107004) |
| CD8 APC/Fire 750 | BioLegend | 53-6.7, 100766 (RRID:AB_2572113) |
| CD44 eFluor450 | eBioscience | IM7, 48-0441-82 (RRID:AB_1272246) |
| CD62L Pe-Cy7 | eBioscience | MEL-14, 25-0621-82 (RRID:AB_469633) |
| CD304 (NRP1) PE | BioLegend | 3E12, 145204 (RRID:AB_2561928) |
| PD1-PE | eBioscience | RMP1-30, 12-9981-82 (RRID:AB_466290) |
| CXCR5-biotin | BD Biosciences | 2G8, 551960 RRID: AB_394301 |
| Alexa Fluor® 647 streptavidin | BioLegend | 405237 |
| CD25 PE-Cy7 | BioLegend | PC61, 102016 (RRID:AB_312865) |
| Foxp3 PE | eBioscience | FJK-16s, 12-5773-82 (RRID:AB_465936) |
| Foxp3 FITC | eBioscience | FJK-16s, 11-5773-82 (RRID:AB_465243) |
| Foxp3 eF450 | eBioscience | FJK-16s, 48-5773-82 (RRID:AB_1518812) |

| Reagent/resource | Reference or source | Identifier or catalog number |
|---|---|---|
| CD71 | BioLegend | RI7217, 113811 (RRID:AB_2203383) |
| Ki67 | eBioscience | SolA15, 50-5698-82 (RRID:AB_2896285) |
| IL-17A | eBioscience | eBio17B7, 50-7177-82 (RRID:AB_11220280) |
| IFN-gamma | eBioscience | XMG1.2,12-7311-81 (RRID:AB_466192) |
| CD45RA | BD Biosciences | HI100 (RRID: AB_1727498) |
| CD45.1 Pacific blue | Produced at IGC | A20 |
| CD45.2 AF647 | Produced at IGC | 104.2 |
| CD4 MicroBeads, human | Miltenyi Biotec | 130-045-101 (RRID:AB_2889919) |
| Thy1.1 | Produced at IGC | 19E12 |
| Thy1.2 | Produced at IGC | 30H12 |
| anti-CD3 mAb (Clone 1XE) | Pelicluster | M1654 (RRID:AB_10553652) |
| anti-CD28 mAb (Clone CD28.2) | eBioscience | 16-0289-85 (RRID:AB_468927) |
| HO-1 | Enzo Life Sciences | ADI-SPA-896 (RRID:AB_10614948) |
| CD16/CD32 | BioLegend | 93, 101331 |
| **Oligonucleotides and other sequence-based reagents** | | |
| Human FTH1 shRNA: FTH[429]; TRCN0000029429; target sequence: GCCTCGGGCTAATTTCCCATA | GPP Web Portal, Broad Institute | RHS3979-9596837 |
| Human FTH1 shRNA: FTH[432]; TRCN0000029432; target sequence: CCTGTCCATGTCTTACTACTT | GPP Web Portal, Broad Institute | RHS3979-9596840 |
| Primers for RT-qPCR, see Table 1 | See Table 1 | N/A |
| **Chemicals, enzymes, and other reagents** | | |
| TMRE-Mitochondrial Membrane Potential Assay Kit | Abcam | ab113852 |
| FerroFarRed™ | Goryo Chemical | GC903-01 |
| LIVE/DEAD™ Fixable Aqua Stain | ThermoFisher Scientific | L34957 |
| eBioscience™ Cell Stimulation Cocktail | ThermoFisher Scientific | 00-4970-93 |
| Protein Transport Inhibitor Cocktail | ThermoFisher Scientific | 00-4980 |
| CFSE | ThermoFisher Scientific | C34554 |
| Solid Phase Reversible Immobilization (SPRI) beads | Beckman Coulter | |
| **Software** *Include version where applicable* | | |
| ImageJ | Schneider et al, 2012 | https://imagej.nih.gov/ij/ |
| Flowjo | BD Sciences | Version 10.8.1 |
| R | R Core Team 2014, Vienna, Austria | Version 3.5.1 |

| Reagent/resource | Reference or source | Identifier or catalog number |
|---|---|---|
| FastQC method | Babraham bioinform. | Version 0.11.5 |
| Python (Linux/UNIX) | Python | Version 2.7.12 |
| Trimmomatic | Bolger et al, 2014 | Version 0.36 |
| HISAT2 | Kim et al, 2015 | Version 2.1.0 |
| HTseq | Anders et al, 2015 | Version 0.6.1p1 |
| DESeq2 | Love et al, 2014 | Version 1.26.0 |
| **Other** | | |
| QIAamp DNA Micro Kit | QIAGEN | 56304 |
| NucleoSpin RNA XS | Macherey-Nagel | 740902 |
| ChIP DNA Clean & Concentrator columns | Zymo Research | D5205 |
| MagniSort™ Mouse CD4 T-cell Enrichment Kit | ThermoFisher Scientific | 8804-6821-74 |
| NEBNext® Enzymatic Methyl-seq Kit | New England Biolabs | E7120 |
| Qubit HS dsDNA kit | ThermoFisher Scientific | Q32851 |
| High Sensitivity DNA Bioanalyzer kit | Agilent | 5067-4626 |
| Seahorse XF Cell Mito Stress Test Kit | Agilent Technologies | 103015-100 |

## Animals

Mice were bred and maintained under specific pathogen-free (SPF) conditions at the Instituto Gulbenkian de Ciência (IGC). All experimental protocols were approved by the Ethics Committee of the IGC, the "*Órgão Responsável pelo Bem-estar dos Animais*" (ORBEA) (license numbers A001-2017, A003-2017) and the Portuguese National Entity (Direcção Geral de Alimentação e Veterinária) (notification numbers 003722, 008830). Experimental procedures were performed according to the Portuguese (Portaria no. 1005/92, Decreto-Lei no. 113/2013 and Decreto-lei no.1/2019) and European (Directive 2010/63/EU) legislations, concerning housing, husbandry, and animal welfare. $Foxp3^{GFP-Fth\Delta/\Delta}$ mice were generated by crossing $Foxp3^{GFP}$ (i.e., B6129S-Tg(Foxp3 EGFP/icre) 1aJbs/J) mice (Chen et al, 2003; Zhou et al, 2008) with $Fth^{fl/fl}$ mice (Darshan et al, 2009). Mouse progeny was genotyped for the presence of the Cre allele. The $Foxp3^{GFP}$ mice express a humanized Cre-recombinase (GFP-hCre) from a *Foxp3* ATG translational start codon, inserted in a bacterial artificial chromosome (BAC) transgene (Chen et al, 2003; Zhou et al, 2008). As Cre expression is not sex-dependent, this allows for *Fth* deletion in $T_{REG}$ cells from male and female $Foxp3^{GFP-Fth\Delta/\Delta}$ mice, using $Foxp3^{GFP}$ and $Fth^{fl/fl}$ mice as controls. $Foxp3^{GFP-Fth\Delta/\Delta-tdT}$ mice were generated by crossing $Foxp3^{GFP-Fth\Delta/\Delta}$ mice with *Ai9 (RCL-tdT)* mice. Control $Foxp3^{GFP-tdT}$ mice were generated by crossing $Foxp3^{GFP}$ mice with C57BL/6 *Ai9 (RCL-tdT)* mice, similar to described above. Progeny was genotyped for the presence of the humanized Cre-recombinase (*Gfp-hCre*) allele. The genetic background of the mouse strains used was C57BL/6J, including $Foxp3^{GFP}$ mice, backcrossed into C57BL/6/J

background for over 10 generations. Rag2$^{-/-}$ mice used as recipients of BM precursor cells were in C57BL/6NTac background.

## Experimental autoimmune encephalomyelitis (EAE)

C57BL/6 mice were immunized with the MOG$_{35-55}$ peptide (s.c.; 200 µg), emulsified in Complete Freund's Adjuvant (CFA) containing *Mycobacterium tuberculosis* (4 mg/mL; BD Diagnostics). Mice received 200 ng of Pertussis toxin (i.v.; Sigma-Aldrich) at the time of immunization and 2 days thereafter. Clinical EAE severity scores were evaluated daily as follows: 0, normal; 1, limp tail; 2, partial paralysis of the hind limbs; 3, complete paralysis of the hind limbs; 4, hind-limb paralysis and forelimb weakness; 5, moribund or deceased, essentially as described (Chora et al, 2007).

## *Plasmodium* infection (malaria)

Mice were infected by the inoculation of blood isolated from mice infected with a *Plasmodium chabaudi chabaudi* (*Pcc*) AS strain [i.p.; $2 \times 10^6$ infected red blood cells (iRBC) per mouse]. Mice were monitored daily for parasitemia, weight, temperature, RBC number, and survival, essentially as described (Seixas et al, 2009).

## Tumor model

Tumor cells B16-F10-luc2 (B16) (CaliperLS) were cultured at 37 °C in RPMI 1640 (Life Technologies) supplemented with 10% Fetal Bovine Serum (Biowest), 1% penicillin–streptomycin (Life Technologies), 50 µg/mL Gentamicin (Life Technologies), and 50 µM 2-Mercaptoethanol (Life Technologies). After trypsin (Life Technologies) treatment, single-cell suspensions were resuspended in ice cold calcium-free and magnesium-free HBSS (Life Technologies). Mice were injected subcutaneously in the right flank with $2 \times 10^5$ B16 cells in 100 µL volume. Tumor size was measured with a caliper every 2 or 3 days, from day 8 post injection, and the tumor diameter (TD) was calculated as TD = (L + W)/2. Mice were sacrificed when TD ≥ 20 mm. By the end of each experiment, tumor clearance (TD ≤ 5 mm) was confirmed upon dissection.

## Human peripheral blood mononuclear cells (PBMC)

Blood samples from anonymized healthy male donors were obtained in accordance with guidelines established by the Sanquin Medical Ethical Committee. Briefly, PBMC was isolated from fresh buffy coats using Ficoll-Paque Plus (GEHealthcare) gradient centrifugation. Next, CD4$^+$ T cells were isolated using magnetic sorting with CD4 microbeads (Miltenyi Biotec) and viable cells were separated using flow cytometric sorting based on the expression of CD25, CD45RA, and CD127 on a FACS Aria III (BD Biosciences).

## Mouse leukocyte isolation

For isolation of leukocytes, spleen and lymph were harvested, disrupted, passed through a cell strainer (70 µm) in PBS (3% FBS, 1 mM EDTA), pelleted (300 × *g*, 4 °C, 10 min), and RBC were lyzed (5 mL RBC lysis buffer; 5 min, RT). Lysis was stopped by adding 5 mL of medium, and cells were passed through a 40-µm cell strainer, centrifuged (300 × *g*, 4 °C, 10 min) and resuspended in PBS containing 3% FBS and 1 mM EDTA.

## Cell sorting

Mice were sacrificed, and LN (i.e., inguinal, brachial, axillary, mandibular, superficial cervical, mesenteric, pancreatic, renal, and lumbar) or spleen were collected, and leukocytes isolated as described above. The negative fraction from CD4$^+$ T cells enrichment (MagniSort™ Mouse CD4 T-cell Enrichment Kit) was recovered, centrifuged, and stained with the following antibody panel: anti-CD11b A647, anti-B220 A647, anti-CD8 A647, anti-CD4 PerCPCy5.5, anti-CD62L Pe-Cy7, and anti-CD44 eF450. Foxp3$^+$ cells were sorted based on endogenous Foxp3$^{EGFP/icre}$ expression (FACS Aria; BD Biosciences). When indicated, naive T cells (CD11b/B220/CD8$^-$CD4$^+$Foxp3$^-$CD44$^{low}$CD62L$^{high}$), memory/activated T cells (CD11b/B220/CD8$^-$CD4$^+$Foxp3$^-$CD44$^{high}$CD62L$^{low}$) and T$_{REG}$ cells (CD11b/B220/CD8$^-$CD4$^+$Foxp3$^+$) were sorted and recovered. A similar procedure was followed for sorting ex-T$_{REG}$ cells, based on endogenous expression of Tomato within GFP$^+$ (CD4$^+$GFP$^+$TdT$^+$) and GFP$^-$ (CD4$^+$GFP$^-$TdT$^+$) cell populations.

## Immunophenotyping

Cells isolated as described in "Leukocyte isolation" were stained for flow cytometry analysis. For surface staining, cells were incubated with Fc block together with LIVE/DEAD™ Fixable Aqua Stain in PBS, followed by incubation with antibodies against the following surface markers: CD8, CD4, CD62L, CD44, CD25, TCRβ, CD11b, and CD304 (Nrp1). Intracellular Foxp3 staining was performed using Foxp3/Transcription Factor Staining Buffer Set. Briefly, upon surface staining, cells were fixed, washed with permeabilization buffer, and incubated with anti-Foxp3 antibody in permeabilization buffer. For T$_{REG}$ cells (CD4$^+$GFP$^+$tdT$^+$) and ex-T$_{REG}$ cells (CD4$^+$GFP$^-$tdT$^+$) staining, cells were fixed and incubated with Fc block, followed by incubation with antibodies directed against the following surface markers: CD4, CD62L, CD44, CD3, TCRβ. T$_{REG}$ cells and ex-T$_{REG}$ cells were distinguished based on endogenous Foxp3$^{EGFP/icre}$ and Tomato expression. Follicular T cells were fixed and incubated with Fc block, followed by incubation with antibodies against surface markers: CD4, CD3, TCRβ, CXCR5, and PD1. Foxp3$^+$ cells were selected based on endogenous *Foxp3$^{EGFP/icre}$* expression. For T$_{REG}$ cells lineage maintenance analysis comparing T$_{REG}$ cells (CD4$^+$GFP$^+$tdT$^+$) and ex-T$_{REG}$ cells (CD4$^+$GFP$^-$tdT$^+$), cells were incubated with Fc block together with LIVE/DEAD™ Fixable Aqua Stain in PBS, followed by incubation with antibodies against surface markers: CD4, CD3, CD71, and fixation with intracellular staining for the proliferation marker Ki67. Cell acquisition was performed using a CYTEK Aurora (Cytek Biosciences) flow cytometer, and data was analyzed using FlowJo software Version 10.8.1.

## Cytokine staining

Cells were isolated as described in "Leukocyte isolation" and were stimulated using Cell Stimulation Cocktail together with Protein Transport Inhibitor Cocktail (4 h; 37 °C) in complete RPMI (10% FBS, 100 U/mL Penicillin and 100 µg/mL Streptomycin). For surface staining, cells were incubated with Fc block together with LIVE/DEAD™ Fixable Aqua Stain, followed by incubation with antibodies directed against the following surface markers: CD8, CD4, and TCRβ. Intracellular Foxp3, IFNγ,

and IL-17 staining were performed using Foxp3/Transcription Factor Staining Buffer Set. Briefly, were fixed upon surface staining cells, washed with permeabilization buffer, and incubated with anti-Foxp3, anti-IFNγ, and anti-IL-17 antibodies in permeabilization buffer. Cell acquisition was performed using BD LSRFortessa X-20 (BD Biosciences) flow cytometer. Alternatively, cells were incubated with Fc block together with LIVE/DEAD™ Fixable Yellow Stain, followed by incubation with antibodies against the following surface markers: CD8, CD4 and TCRβ. Intracellular Foxp3, IFNγ, IL-17 and IL10 staining were performed using Foxp3/Transcription Factor Staining Buffer Set. Briefly, upon surface staining cells were fixed, washed with permeabilization buffer, and incubated with anti-Foxp3, anti-IFNγ, anti-IL-17 and anti-IL-10 antibodies in permeabilization buffer. Cell acquisition was performed using a CYTEK Aurora (Cytek Biosciences) flow cytometer. Data were analyzed with FlowJo software Version 10.8.1.

## Leukocyte staining with fluorescent probes

Cells were isolated as described in "Leukocytes isolation". To evaluate mitochondrial membrane potential, cells were incubated with (tetramethylrhodamine, ethyl ester) TMRE-Mitochondrial Membrane Potential probe (20 nM) in RPMI without serum (20 min 37 °C). Control samples were pre-incubated with the ionophore uncoupler of oxidative phosphorylation FCCP (carbonyl cyanide 4-(trifluoromethoxy) phenylhydrazone; 20 μM, 10 min, 37 °C), to eliminate mitochondrial membrane potential and positive TMRE staining. To detect intracellular labile $Fe^{2+}$, cells were incubated with FerroFarRed (5 μM; 1 h 37 °C) in RPMI without serum. After incubation with the fluorescent probes, cells were stained with Fc block together with LIVE/DEAD™ Fixable Aqua Stain, followed by incubation antibodies against the following surface markers: CD4, CD44, CD62L, and TCRβ. $Foxp3^+$ cells were selected based on endogenous $Foxp3^{GFP}$ expression. Cell acquisition was performed using BD LSRFortessa X-20 (BD Biosciences) flow cytometer and data were analyzed using FlowJo software Version 10.8.1.

## Bone marrow transplants

Bone marrow (BM) cells were harvested by flushing the femurs and tibias of donor mice, and T cells were depleted by antibody-mediated complement killing. The rabbit complement was prepared fresh, via incubation on ice (30 min), centrifugation (300 × *g*, 10 min, 4 °C) and filtering (0.22 μm). BM cell suspensions (1 × 10⁷/mL in PBS) were incubated with an anti-Thy1.2 mAb (0.5 μg/mL) and mixed gently (every 15 min) with rabbit complement (LowTox-M, CL3051, CEDARLANE) at a ratio of 100 μL per mL of BM cell suspension (37 °C, 1 h). Complement activity was neutralized (FBS, 200 μL/mL), cell suspensions were filtered (70-μm mesh, cell strainer), washed (2x in PBS, 2% FBS and 1× in PBS without serum), and cell numbers adjusted (10⁸/mL) in PBS. BM cells from C57BL/6 CD45.1⁺ mice were mixed with BM cells from congenic C57BL/6 CD45.2⁺ *Foxp3^{GFP-FthΔ/Δ-tdT}* or control *Foxp3^{GFP-tdT}* mice at a 1:1 ratio and injected (i.v.; tail vein, 200 μL) into congenic C57BL/6-recipient *Rag2*-deficient (*Rag2^{−/−}*) female mice, 2 h after irradiation (600 Gys). Hematopoietic chimerism was monitored by immunophenotyping, 6 weeks after bone marrow reconstitution and thereafter.

## T_REG cell in vivo homeostatic expansion

LN (i.e., superficial, cervical, axillary, brachial, mesenteric, inguinal, lumbar, renal, caudal, and popliteal) and spleen were collected from *Foxp3^{GFP-tdT}* and *Foxp3^{GFP-FthΔ/Δ-tdT}* mice and gently disrupted in 70-μm mesh tissue to isolate leukocytes. Cell suspensions were washed in cold PBS, red blood cells were lyzed (i.e., ammonium chloride), passed through 40-μm cell strainer and CD4⁺GFP⁺TdT⁺ T_REG cells were sorted upon surface marker staining, as described in "Cell sorting". To test T_REG cell stability in vivo, CD4⁺GFP⁺TdT⁺ T_REG cells from *Foxp3^{GFP-tdT}* and *Foxp3^{GFP-FthΔ/Δ-tdT}* mice were adoptively transferred (i.v., 1 × 10⁵ cells) to *Rag2^{-/-}* mice. LN and spleen were collected six weeks later, disrupted, and RBC was lysed. The number of CD4⁺GFP⁺TdT⁺ T_REG cells and CD4⁺GFP⁻TdT⁺ ex-T_REG cells was quantified by flow cytometry, as described in "Immunophenotyping". An additional aliquot of spleen cells was used to sort CD4⁺GFP⁺TdT⁺ T_REG cells, as described in "Cell sorting". These were used to extract mRNA and monitor *Fth* mRNA expression in the CD4⁺GFP⁺TdT⁺ T_REG cells used for the adoptive transfer.

## T_REG cell proliferation suppression assay

Naive T cells were sorted, as described in "Cell sorting", washed with PBS, and incubated with Cell Tracer Violet (CTV) (Thermofisher; 1/1000 in PBS without serum) at RT in the dark for 15 min. Staining was stopped by adding five volumes of complete media (containing 10% FBS). Sort-purified T_REG cells were plated and serially twofold diluted, starting at 2.5 × 10⁴ cells/well in round-bottom 96-well plates with 1 μg/mL soluble anti-CD3 mAb and 5 × 10⁴ irradiated splenocytes. CTV-labeled T_N cells were plated (2.5 × 10⁴ cells/well), resulting in T_REG:T_NAIVE ratio ranging from 1:1 to 1:64. Cultures were set in triplicates in a final volume of 200 μL. On day 3 of culture, CTV intensity was measured in responder T cells defined as Thy1.1⁺ Thy1.2⁻TCRb⁺CD4⁺, live lymphocytes.

## In vitro induction of T_REG cells and flow cytometry analysis

Naive T cells sorted, as described in "Cell sorting", were cultured for 5 days on a maxisorb 96-well plate pre-coated with anti-CD3 mAb (1 μg/mL; 100 μL/well in PBS) in RPMI complete media (10% FBS, 100 U/mL Penicillin and 100 μg/mL Streptomycin), supplemented with anti-CD28 mAb (1 μg/mL), mouse recombinant IL-2 (20 ng/mL) and TGFβ (5 ng/mL) (iT_REG cell differentiation medium). Alternatively, naive T cells were cultured in the same media, without TGFβ (conventional T-cell differentiation medium). For cell surface staining, cells were incubated with Fc block together with LIVE/DEAD™ Fixable Aqua Stain, followed by incubation with anti-CD4 antibody. For intracellular Foxp3 staining cells were fixed after surface staining, washed with permeabilization buffer, and incubated with anti-Foxp3 antibody in permeabilization buffer, according to the Foxp3/Transcription Factor Staining Buffer Set. Cells were acquired in a BD LSRFortessa X-20 (BD Biosciences) flow cytometer and analyzed using FlowJo software Version 10.8.1. For analysis at day 12 after T_REG induction, induced iT_REG cells were re-plated at day 5 in RPMI media supplemented with IL-2 (100 ng/mL) with or without Fe sulfate (20 μM) and cultured for 7 days. To determine cell proliferation, naive T cells (5 × 10⁶/mL) were stained with CFSE (5 μM, 20 min RT), washed with complete medium to stop the reaction and re-cultured in T_REG or conventional T cells medium.

## Lentiviral transduction

Lentivirus was produced by transfecting confluent human HEK293T cells with packaging (psPAX2) and envelope plasmids (pMD2.G) with pLKO.1. HEK293T cells were cultured in DMEM with HEPES (Life Technologies) supplemented with 10% fetal calf serum and 1% penicillin/streptomycin. Polyethylenimine (Polysciences, Hirschberg an der Bergstrasse, Germany) was used as a transfection reagent. After 24 h, the cultures were refreshed with medium with 2% FCS and 24 h later, lentiviral particles were concentrated and purified by ultracentrifugation at $50{,}000 \times g$, 2.5 h, 8 °C. Naive $T_{CONV}$ (CD4$^+$, CD127$^+$, CD25$^-$, CD45RA$^+$) and naive $T_{REG}$ (CD4$^+$, CD127$^-$, CD25$^+$, CD45RA$^+$) cells were isolated by FACS sorting (FACS Aria III, BD Biosciences) as described above. The cells were then cultured in presence of 0.1 µg/mL of anti-CD3 mAb (M1654, clone 1XE, PeliCluster) and anti-CD28 mAb (16-0289-85, clone CD28.2, eBioscience) for 5 days in IMDM containing 10% FCS and 300 U/mL IL-2 and restimulated one day prior to transduction. Cells were then infected in RetroNectin® (Clontech) coated plates for 24 h. After that, the cells were transferred into tissue culture-treated plates with medium containing 100 U/mL IL-2 and puromycin. After 5 days, the cells were directly used for FACS analysis or lysed for western blot assays.

## Western blot

Human T conventional (CD4$^+$CD45RA$^+$CD127$^+$CD25$^-$), $T_{REG}$ (CD4$^+$CD127$^-$CD45RA$^+$CD25$^{hi}$) cells or HEK293T were washed (2× in PBS) and directly lysed in RIPA buffer. Cell lysates containing equal amounts of protein were boiled in a sample buffer prior to gel electrophoresis. SDS-PAGE gel electrophoresis was performed using the NuPAGE electrophoresis system (Novex, Life Technologies). Proteins were transferred using the iBlot system (Thermo Scientific) and analyzed using the corresponding antibodies. ECL signals on Western blots were developed using the Pierce ECL substrate kit (Pierce) followed by autoradiographic detection on film (Fuji Medical). Sorted cells mouse $T_{REG}$, $T_M$, and $T_N$ cells were directly lysed in 2× SDS-page sample buffer (20% glycerol, 4% SDS, 100 mM Tris pH 6.8, 0.002% bromophenol blue, 100 mM dithiothreitol). Samples were then sonicated or treated with Benzonase to degrade DNA, heated (10 min; 70 °C), and centrifuged. The supernatant was collected, and the protein was quantified using NanoDrop™ 1000. Anti-FTH1 (1:1000), anti-Histone H3 (1:1000) were detected using peroxidase-conjugated secondary antibodies (1 h; RT) and developed with SuperSignal™ West Pico PLUS Chemiluminescent Substrate (ThermoFisher Scientific). ECL signal was developed using Pierce ECL substrate kit followed by autoradiographic detection on film (Fuji Medical). Alternatively, western blots were developed using Amersham Imager 680 (GEHealthcare), equipped with a Peltier-cooled Fujifilm Super CCD. Densitometry analysis was performed with ImageJ using images without saturated pixels.

## qRT-PCR

RNA was isolated from cells using NucleoSpin RNA XS kit (Macherey-Nagel) according to the manufacturer's instructions. cDNA was transcribed from total RNA with transcriptor first strand cDNA synthesis kit (Roche) or Xpert cDNA Synthesis Kit (GRiSP). Quantitative real-time PCR (qRT-PCR) was performed using 1 µg cDNA and SYBR Green Master Mix (Applied Biosystems, Foster City, CA, USA) in duplicate on a 7500 Fast Real-Time PCR System (Applied Biosystems) under the following conditions: 95 °C/10 min, 40 cycles/95 °C/15 s, annealing at 60 °C/ 30 s, and elongation 72 °C/30 s. Primers listed in Table 1 were designed using Primer Blast (Ye et al, 2012).

## Serology

Mice were euthanized using $CO_2$ inhalation. Whole blood was collected by cardiac puncture and transferred into an EDTA for hemogram analyses or heparin tubes for serology (Iron, Transferrin, and Transferrin saturation). Analysis was performed by DNAtech (Lisbon).

## Histology

Organs were harvested, fixed (10% formalin), embedded in paraffin, sectioned (3-µm-thick sections), and stained with Hematoxylin and Eosin (H&E). Whole sections were analyzed, and images acquired with a Leica DMLB2 microscope (Leica) and NanoZoomer-SQ Digital slide scanner (Hamamatsu).

## mtDNA/nDNA qPCR

Total isolated DNA was used to perform the quantification of mitochondrial DNA (mtDNA) in comparison to nuclear DNA (nDNA) using a qRT-PCR-based method (Quiros et al, 2017). Briefly, qRT-PCR was performed form using 20 ng of DNA and SYBR Green Master Mix (Bio-Rad), in duplicate on a 7500 Fast Real-Time PCR System (Applied Biosystems), under the following conditions: 50 °C/2 min and 95 °C/5 min (Hold stage), 45 cycles/ 95 °C/10 s, annealing at 60 °C/30 s, and elongation 72 °C/20 s, followed by melting curve: 95 °C/15 s, 60 °C/1 min, and gradual increase in temperature up to 95 °C. Primers for NADH-ubiquinone oxidoreductase chain 1 encoded by the mitochondrial gene *MT-Nd1* (*Nd1*) and for the nuclear-encoded *hexokinase 2* gene (*Hk2*) (Quiros et al, 2017) are listed in Table 1. Mitochondria number per cell was calculated according to the ratio of mRNA expression of the single copy mitochondrial gene Nd1 and the single copy nuclear gene Hk2.

## Seahorse assays

Oxygen consumption rate (OCR) and extracellular acidification rate (ECAR) were measured using a Seahorse XFe96 analyzer (Agilent Tech.) and the Seahorse XF Cell Mito Stress Test Kit according to instructions from the manufacturer. Specifically sorted TREG and TCONV cells were plated on poly-L lysine-coated Seahorse XF96 plates ($15 \times 10^3$ cells/well) XF medium (10 mM glucose, 1 mM sodium pyruvate, 2 mM L-glutamine, pH 7.4), and centrifuged ($400 \times g$, for 5 min) to promote cell adhesion. Cells were incubated in a non-$CO_2$ incubator (37 °C; 1 h) prior to the assay. The analyzer was programmed to calibrate and equalize samples, followed by three baseline measurements (3 min each) and mixing (2 min) between measurements prior to inhibitor injection. The inhibitors were injected in the following order: Oligomycin (1 µM); FCCP (2 µM); Antimycin A/Rotenone (1 µM); and three

**Table 1.** RT-qPCR primers.

| Oligonucleotides | Sequences | Reference |
| --- | --- | --- |
| *Arbp0* Fwd | 5'-CTTTGGGCATCACCACGAA-3' | Blankenhaus et al, 2019 |
| *Arbp0* Rev | 5'-GCTGGCTCCCACCTTGTCT-3' | Blankenhaus et al, 2019 |
| *Fth* Fwd | 5'-CCATCAACCGCCAGATCAAC-3' | Blankenhaus et al, 2019 |
| *Fth* Rev | 5'-GCCACATCATCTCGGTCAAA-3' | Blankenhaus et al, 2019 |
| *Ftl* Fwd | 5'-AAGATGGGCAACCATCTGAC-3' | This work |
| *Ftl* Rev | 5'-GCCTCCTAGTCGTGCTTGAG-3' | This work |
| *Hk2* Fwd | 5'-GCCAGCCTCTCCTGATTTTAGTGT-3' | Quiros et al, 2017 |
| *Hk2* Rev | 5'-GGGAACACAAAAGACCTCTTCTGG-3' | Quiros et al, 2017 |
| *Nd1* Fwd | 5'-CTAGCAGAAACAAACCGGGC-3' | Quiros et al, 2017 |
| *Nd1* Rev | 5'-CCGGCTGCGTATTCTACGTT-3' | Quiros et al, 2017 |
| *GFP* Fwd | 5'-CGACGTAAACGGCCACAAGTTCAG-3' | Liu et al, 2012 |
| *GFP* Rev | 5'-CCGTAGGTCAGGGTGGTCACGAG-3' | Liu et al, 2012 |
| Tdt Fwd | 5'-GCCGACATCCCCGATTACAAGA-3' | Wienert et al, 2015 |
| Tdt Rev | 5'-CGATGGTGTAGTCCTCGTTGTGG-3' | Wienert et al, 2015 |
| *Foxp3* Fwd | 5'-GGCCCTTCTCCAGGACAGA-3' | Fontenot et al, 2003 |
| *Foxp3* Rev | 5'-GCTGATCATGGCTGGGTTGT-3' | Fontenot et al, 2003 |

measurements (3 min each) were made following each injection with 2 min of mixing between measurements.

## RNA sequencing

RNA was extracted from (CD4 + GFP + ) $T_{REG}$ cells sorted from the LN of Foxp3$^{GFP\text{-}Fth\Delta/\Delta}$ vs. control Foxp3$^{GFP}$ mice (see Fig. 2) or from (CD45.2 + CD3 + CD4 + GFP+tdT + ) $T_{REG}$ and (CD45.2 + CD3 + CD4 + GFP-tdT + ) ex-$T_{REG}$ cells sorted from LN of BM chimeric mice (see Fig. 4). RNA sequencing data from non-chimeric mice was analyzed as follows: RNA sequencing was performed using Illumina's next-generation sequencing (Bentley et al, 2008). Quality check and quantification of total RNA was done using the Agilent Bioanalyzer 2100 in combination with the RNA 6000 pico kit (Agilent Technologies). Library preparation was done using SMARTer Stranded Total RNA-Seq Kit v2-Pico Input Mammalian (Takara) following the manufacturer's description. Library quantification and quality check was done using the Agilent Bioanalyzer 2100 in combination with the DNA 7500 kit. Libraries were sequenced on a HiSeq2500 running in 51cycle/single-end/rapid mode. All libraries were pooled and sequenced in two lanes. Sequence information was extracted in FastQ format using Illumina's bcl2fastq v.2.19.1.403. Sequencing resulted in around 29mio reads per sample. RNA sequencing data from cells extracted from mixed BM chimeric mice was analyzed as follows: RNA was extracted and quality assessed using Agilent Bioanalyzer 2100 using the RNA 6000 pico kit (Agilent Technologies). Full-length cDNAs and sequencing libraries were generated following the SMART-Seq2 protocol[62]. Library preparation, including cDNA "tagmentation", PCR-mediated adaptor addition and library amplification was performed using the Nextera library preparation protocol (Nextera XT DNA Library Preparation kit, Illumina). Libraries were sequenced (NextSeq 500, Illumina) using High Output kit v2.5 (75 cycles). Sequencing data was extracted in FastQ format, using Illumina's bcl2fastq v.2.19.1.403, producing $30.14 \times 10^6$ reads per sample on average. Library preparation and next-generation sequencing were performed at the IGC Genomics Unit. Fastq reads were aligned against the mouse reference genome GRCm39 using the GENCODE vM27 annotation to extract splice junction information (STAR; v.2.5.2a)[64]. Read summarization was performed by assigning uniquely mapped reads to genomic features using *FeatureCounts* (subread package v.1.5.0-p1). Gene expression tables were imported into the R environment (v.4.1.0) to perform differential gene expression, functional enrichment analyses, and data visualization (R Core-Team, 2021). Differential gene expression analysis was performed using the DESeq2 R package (v.1.32). Gene expression was modeled by genotype. Genes not expressed or with fewer than 10 counts across the samples were removed. We subsequently ran the function *DESeq* to estimate the size factors (by *estimateSizeFactors*), dispersion (by *estimateDispersions*) and fit a binomial GLM fitting for βi coefficient and Wald statistics (by *nbinomWaldTest*). Pairwise comparisons were performed with the function *results* (alpha = 0.05), and the log$_2$ fold change for each pairwise comparison was shrunken with the function *lfcShrink* using the algorithm *ashr* (v.2.2-47)[65]. Differentially expressed genes were considered as genes with an adjusted *P* value < 0.05 and an absolute log$_2$ fold change >0. Normalized gene expression counts were obtained with the function *counts* using the option normalized = TRUE. Regularized log-transformed gene expression counts were obtained with *rlog*, using the option blind = TRUE. Ensembl gene ids were converted into gene symbols from Ensembl (v.104 - May 2021—https://may2021.archive.ensembl.org) by using the mouse reference (GRCm39) database with biomaRt R package (v.2.48.2). All plots were created using the ggplot2 R package (v.3.3.5). Heatmaps were created with pHeatmap (v.1.0.12), using Euclidean distance and Ward.D2 methods for clustering estimation. For hierarchical clustering, gene expression counts were scaled (Z-score) with the function *scale*. Functional enrichment analysis was performed using the gprofiler2 R package (v.0.2.1). Enrichment was performed with the function *gost* based on the list of up- or downregulated genes, between each pairwise comparison, against annotated genes (domain_scope = "annotated")

of the organism Mus musculus (organism = "mmusculus"). Gene lists were sorted based on adjusted p value (ordered_query = TRUE) to generate GSEA (Gene Set Enrichment Analysis) style *p* values. Only statistically significant (user_threshold=0.05) enriched functions are returned (significant=TRUE) after multiple testing corrections with the default method g:SCS (correction_method = "analytical"). Gprofiler2 queries were run against the default functional databases for mouse which include Gene Ontology (GO:MF, GO:BP, GO:CC), KEGG (KEGG), Reactome (REAC), TRANSFAC (TF), miRTarBase (MIRNA), Human phenotype ontology (HP), WikiPathways (WP), and CORUM (CORUM). Gprofiler2 was performed using database versions Ensembl 104, and Ensembl gene 51 (database updated on 07/05/2021).

## RNA-sequencing data analysis

Sequence read quality was assessed by means of the FastQC method (v0.11.5; http://www.bioinformatics.babraham.ac.uk/projects/fastqc/). Trimmomatic version 0.36 was used to trim Illumina adapters and poor-quality bases (trimmomatic parameters: leading=3, trailing=3, sliding window=4:15, minimum length=40). The remaining high-quality reads were used to align against the Genome Reference Consortium mouse genome build 38 (GRCm38). Mapping was performed by HISAT2 version 2.1.0 with parameters as default. Count data were generated by means of the HTSeq method and analyzed using the DESeq2 method in the R statistical computing environment (R Core Team 2014. R: A language and environment for statistical computing. R Foundation for Statistical Computing, Vienna, Austria). Statistically significant differences were defined by Benjamini & Hochberg adjusted probabilities <0.05. Canonical signaling pathways and biofunctions were generated by Ingenuity Pathway Analysis (IPA; QIAGEN) specifying mouse species and ingenuity database as reference. Benjamini & Hochberg adjusted probabilities <0.05 demarcated significance. Functional enrichment analysis was performed with gProfiler (Kolberg et al, 2023). Data were analyzed using g:SCS multiple testing correction method with a significance threshold of 0.05.

## Targeted metabolomics

Targeted metabolomics analyses of cell extracts for intermediates of the tricarboxylic acid cycle (TCA cycle; citrate, isocitrate, α-ketoglutarate, succinate, fumarate, malate, cis-aconitate) and additional metabolites closely linked to the TCA cycle (pyruvate, lactate, DL-2-hydroxyglutarate, itaconate, and the amino acids aspartate, glutamate, glutamine) were performed by liquid chromatography-tandem mass spectrometry (LC-MS/MS), as described elsewhere (Richter et al, 2019). For metabolite quantification, ratios of analyte peak areas to peak areas of respective stable isotope labeled internal standards determined in cell extracts were compared to those in calibrators.

## Genome-wide methyl-sequencing (EM-seq)

The initial genomic DNA (gDNA) was quantified using the Invitrogen Qubit 4 as per the manufacturer's protocol (1 μL in 199 μL of Qubit working solution). For library preparation, NEB enzymatic Methy-Seq kit was used, following the manufacturer's protocol for large insert libraries. Based on the quality assessment, samples were standardized to 10 ng of DNA in a volume of 50 μL, including a spike-in of pUC and Lambda DNA provided with the kit, as a control for methylation

efficiency as per the manufacturer's protocol. Samples were fragmented using the Covaris S2 system, with settings to achieve an average fragment size of 350-400 bp and individual barcoded during the PCR, using eight PCR cycles as per the manufacturer's protocol. Libraries were quantified using the Qubit HS DNA assay as per the manufacturer's protocol (1 μL of sample in 199 μL of Qubit working solution). The quality and molarity of the libraries were assessed using Agilent Bioanalyzer with the DNA HS Assay kit as per the manufacturer's protocol. Molarity was used to equimolarly combine individual libraries into one pool, loaded and sequenced on an Illumina NextSeq 2000 platform (Illumina, San Diego, CA, USA) using a P3 300 cycle kit and a read-length of 155 bp paired-end reads. Raw sequencing reads were deposited to ENA under the accession number: Sequencing reads were processed by the methylseq (v1.6.1) (https://zenodo.org/record/2555454/export/xd) nf-core (Ewels et al, 2020). Default parameters were used with BWA-meth as the aligner, GRCm38 as the reference genome. The --em-seq and --methyl_kit options were also used for downstream analysis. R methylkit package (v1.22.0) (Akalin et al, 2012) was used to load the methylation calls, calculate the methylation rate, and generate plots presented in this paper. Sliding Linear Model (SLIM) multiple testing correction (Wang et al, 2011) was used to extract the significant methylated sites.

## TET activity assay

TET enzymatic activity was monitored in human HEK293T cells, transiently transfected with human TET3 (pCMV-3×FLAG-TET3(human)-Neo; MiaolingBio P45302) plus human FTH (pCMV-FTH-3Tag3a) or FTH mutant (FTH^mut; pCMV-FTH^mut-3Tag3a; Glu 62 and His 65 and Lys86 were mutated as Lys, Gly, Gln, respectively, to achieve fully ablation of ferroxidase activity) lacking ferroxidase mutant FTH (FTH^mut), similar to previously described (Broxmeyer et al, 1991), followed by nuclear protein extraction and ELISA-based TET activity measurement. Briefly, HEK293T cells, at 60% confluency in six-well plates, were transiently transfected with the human TET3 (800 ng/well) together with FTH (200 ng/well), FTH^mut (200 ng/well) cDNA expressing vectors or empty vector (pCMV-3Tag3a; 200 ng/well) using transfection reagent (YEASEN, 40802ES03). Nuclear proteins were extracted 48 h after transfection, using a commercial kit (EpiQuik, OP-0002-1) and quantified (BCA assay; Meilunbio MA0082-2). TET activity was measured by ELISA-based the manufacturer's instructions (EpiQuik, P-3087). Expression of flag-tagged TET3, FTH and FTH^mut was monitored by western blot. Briefly, cytosol and nuclear extracts were loaded on a 4–20% gradient SDS-PAGE precast-Gel (ACE Biotechnology, ET15420LGel) and proteins were transferred on PVDF membranes. These were blocked (5% skimmed milk in TBST; 1 h) and incubated (4 °C; overnight) with an HRP-labeled anti-FLAG-HRP antibody (Sigma, A8592). Membranes were washed (3 × 10 min; TBST) and HRP activity was detected by ECL (Thermo Scientific, 32106). Images were capture by Molecular Imager® ChemiDoc™ XRS+ with Image Lab™ Software (BIO-RAD). Membranes were further incubated (1.5 h, RT) with anti-Lamin A/C (Proteintech, 10298-1-AP) and anti-GAPDH (Abclonal, AC033) antibodies, used as reference cytosolic and nuclear proteins, respectively. Membranes were washed (3 × 10 min; TBST), incubated with goat anti-rabbit (Abclonal, AS014) and goat anti-mouse (Abclonal, AS003), secondary antibodies, respectively, washed (3 × 10min in TBST), and HRP signal was detected by ECL to blot references proteins for cytosol and nuclear fractions, respectively, as above.

## Quantification and statistical analysis

Statistical analysis was conducted using GraphPad Prism 9 software. All distributed data are displayed as means ± standard deviation of the mean (SD) unless otherwise noted. Measurements between two groups were performed with unpaired *t* test with Welch's correction, paired *t* test, or Mann–Whitney test. Groups of three or more were analyzed by one-way or two-way analysis of variance (ANOVA). Survival was assessed using a log-rank (Mantel–Cox) test. Statistical parameters for each experiment can be found within the corresponding figure legends.

## Data availability

Data from RNA sequencing studies are available at GEO database GSE173181 (Fig. 3B) and GSE226032 (Fig. 5I,J). The methylation data for this study have been deposited in the European Nucleotide Archive (ENA) at EMBL-EBI under accession number PRJEB59884 (Fig. 6).

## Peer review information

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

## Acknowledgements

The authors are indebted to the excellent core facilities of the Instituto Gulbenkian de Ciência. QW was supported by the Marie Skłodowska-Curie Research Fellowship (RIGM 892773), the International Postdoctoral Exchange Fellowship Program from the People´s Republic of China (20190090), and the National Natural Science Foundation of China (32171166, 82030003). FB was supported by Marie Skłodowska-Curie Research Fellowship (REGDAM 707998), ARC by Fundação para a Ciência e Tecnologia, Portugal (FCT; SFRH/BPD/101608/2014 and CEECIND/01589/2017), JZK by FCT (2022.08590.PTDC) and Human Frontier Science Program fellowship (LT0043/2022-L), RM by EMBO long-term fellowship (ALTF290-2017) and Marie Skłodowska-Curie Research Fellowship (BILITOLERANCE 753236), BSPO by Calouste Gulbenkian Foundation, VCM by Calouste Gulbenkian Foundation, FCT (PTDC/MED-IMU/3649/2021 and CEECIND/03106/2018), and La Caixa Foundation (LCF/PR/HR22/52420023). MLB by FCT (283/BI/15; UID/Multi/04555/2013), JAT by ESCMID Research Grant and FCT (SFRH/BPD/112135/2015), SW by DFG Excellence Cluster EXS 2051 and CSCC, Jena University Hospital (BMBF 01EO1502), MF by Spanish Agencia Estatal de Investigación (PID2019-105739GB-I00), JD by Calouste Gulbenkian foundation, DA, MS, and SR by the Landsteiner Foundation for Blood Cell Research (LSBR 1818), MPS by Calouste Gulbenkian Foundation, "La Caixa"/FCT (HR18-00502), Bill & Melinda Gates Foundation (OPP1148170) and FCT (5723/2014 and FEDER029411). Methylome analyzes was supported by SymbNET-Genomics and Metabolomics in a Host-Microbe Symbiosis Network Project (Horizon 2020 Framework Programme (H2020) - 952537). Mouse experiments were supported by FCT/Lisboa2020/Por2020/ERDF (CONGENTO LISBOA-01-0145-FEDER-022170). Cell sorting and flow cytometry analysis of mouse-derived cells was performed at IGC Flow Cytometry Unit, and histology was performed at the Histopathology Unit. RNA sequencing was performed at the IGC Genomics Unit.

## Author contributions

**Qian Wu**: Conceptualization; Resources; Data curation; Formal analysis; Funding acquisition; Validation; Investigation; Methodology; Writing—review and editing. **Ana Rita Carlos**: Conceptualization; Resources; Data curation; Formal analysis; Funding acquisition; Validation; Investigation; Methodology; Writing—review and editing. **Faouzi Braza**: Conceptualization; Data curation; Formal analysis; Investigation; Visualization; Methodology. **Marie-Louise Bergman**: Formal analysis; Investigation; Methodology. **Jamil Z Kitoko**: Formal analysis; Methodology. **Patricia Bastos-Amador**: Conceptualization; Formal analysis; Investigation; Methodology. **Eloy Cuadrado**: Formal analysis; Investigation; Methodology. **Rui Martins**: Software; Formal analysis; Investigation; Visualization; Methodology. **Bruna Sabino Oliveira**: Data curation; Investigation; Methodology. **Vera C Martins**: Data curation; Formal analysis; Investigation; Methodology; Writing—review and editing. **Brendon P Scicluna**: Software; Formal analysis; Visualization; Methodology. **Jonathan JM Landry**: Software; Formal analysis; Visualization; Methodology. **Ferris E Jung**: Software; Formal analysis; Investigation. **Temitope W Ademolue**: Data curation; Investigation; Methodology. **Mirko Peitzsch**: Software; Formal analysis; Investigation; Methodology. **Jose Almeida-Santos**: Data curation; Investigation; Methodology. **Jessica Thompson**: Formal analysis; Investigation; Methodology. **Silvia Cardoso**: Resources; Methodology. **Pedro Ventura**: Data curation; Investigation; Methodology. **Manon Slot**: Data curation; Investigation; Methodology. **Stamatia Rontogianni**: Data curation; Investigation; Methodology. **Vanessa Ribeiro**: Data curation; Investigation; Methodology. **Vital Da Silva Domingues**: Data curation; Investigation; Methodology. **Inês A Cabral**: Data curation; Investigation; Methodology. **Sebastian Weis**: Data curation; Software; Investigation; Methodology. **Marco Groth**: Software; Formal analysis; Investigation; Methodology. **Cristina Ameneiro**: Data curation; Investigation; Methodology. **Miguel Fidalgo**: Data curation; Formal analysis; Investigation; Methodology. **Fudi Wang**: Resources. **Jocelyne Demengeot**: Conceptualization; Writing—review and editing. **Derk Amsen**: Data curation; Formal analysis; Investigation; Methodology. **Miguel P Soares**: Conceptualization; Resources; Data curation; Formal analysis; Supervision; Funding acquisition; Validation; Visualization; Writing—original draft; Project administration; Writing—review and editing.

## Disclosure and competing interests statement

MPS is a consultant to the New York Blood Center (NYBC), NYC, USA. The remaining authors declare no competing interests.

# Expanded View Figures

**Figure EV1.   FTH expression in T$_{REG}$ cells alters systemic iron metabolism.**

(**A**) Schematic representation of experimental approach (left panel) and relative quantification of *Fth* mRNA, by qRT-PCR, normalized to *Arbp0* mRNA (right panel), of (CD4$^+$ GFP$^+$) T$_{REG}$ cells sorted from mesenteric lymph nodes (MLN). Data from $N = 3$ mice per genotype. (**B**) Schematic representation of experimental approach used for relative quantification of intracellular Fe$^{2+}$ in (CD4$^+$GFP$^+$) T$_{REG}$ cells in mesenteric lymph nodes (MLN), using the FeRhoNox™-1 probe (left panel). Representative flow cytometry histograms (middle panel) and relative quantification (right panel) of mean fluorescence of intracellular Fe$^{2+}$ intensity (MFI) $N = 3$ mice per genotype. (**C**) Schematic representation of experimental approach (left panel), representative flow cytometry dot plots (middle panel) and corresponding quantification of percentage (%) and cell number (Nbr.) (right panels) of live splenic follicular (CD4$^+$GFP$^+$CXCR5$^+$PD1$^+$) FT$_{REG}$ cells. Data from $N = 4$ mice per genotype, from one experiment. (**D, E**) Representative flow cytometry dot plots (left panels) and number (right panel) of live activated (CD4$^+$CD44$^{high}$CD62L$^{low}$) and (CD8$^+$CD44$^{high}$CD62L$^{low}$) T cells in the spleen (**D**) and MLN (**E**). Data from $N = 6$–8 mice per genotype, pooled from four independent experiments, with similar trend. (**F, G**) Representative flow cytometry dot plots (left panels) and percentage (%) (right panel) of live activated (CD3$^+$CD4$^+$Foxp3$^-$IFN-γ$^+$; T$_H$1) T$_H$1 and (CD3$^+$CD8$^+$Foxp3$^-$IFN-γ$^+$) T$_C$ in the spleen (**F**) and MLN (**G**). Data representative of $N = 5$ mice per genotype, pooled from two independent experiments, with similar trend. Data information: Data in (**A–G**) represented as mean ± SD, circles in (**A, C–G**) correspond to individual mice and red bars to mean values. *P* values in (**A–C**) calculated using unpaired *t* test with Welch's correction, and in (**D–G**) using two-way ANOVA with Sidak's multiple comparison test. *$P < 0.05$, **$P < 0.01$, ***$P < 0.001$, ****$P < 0.0001$. Source data are available online for this figure.

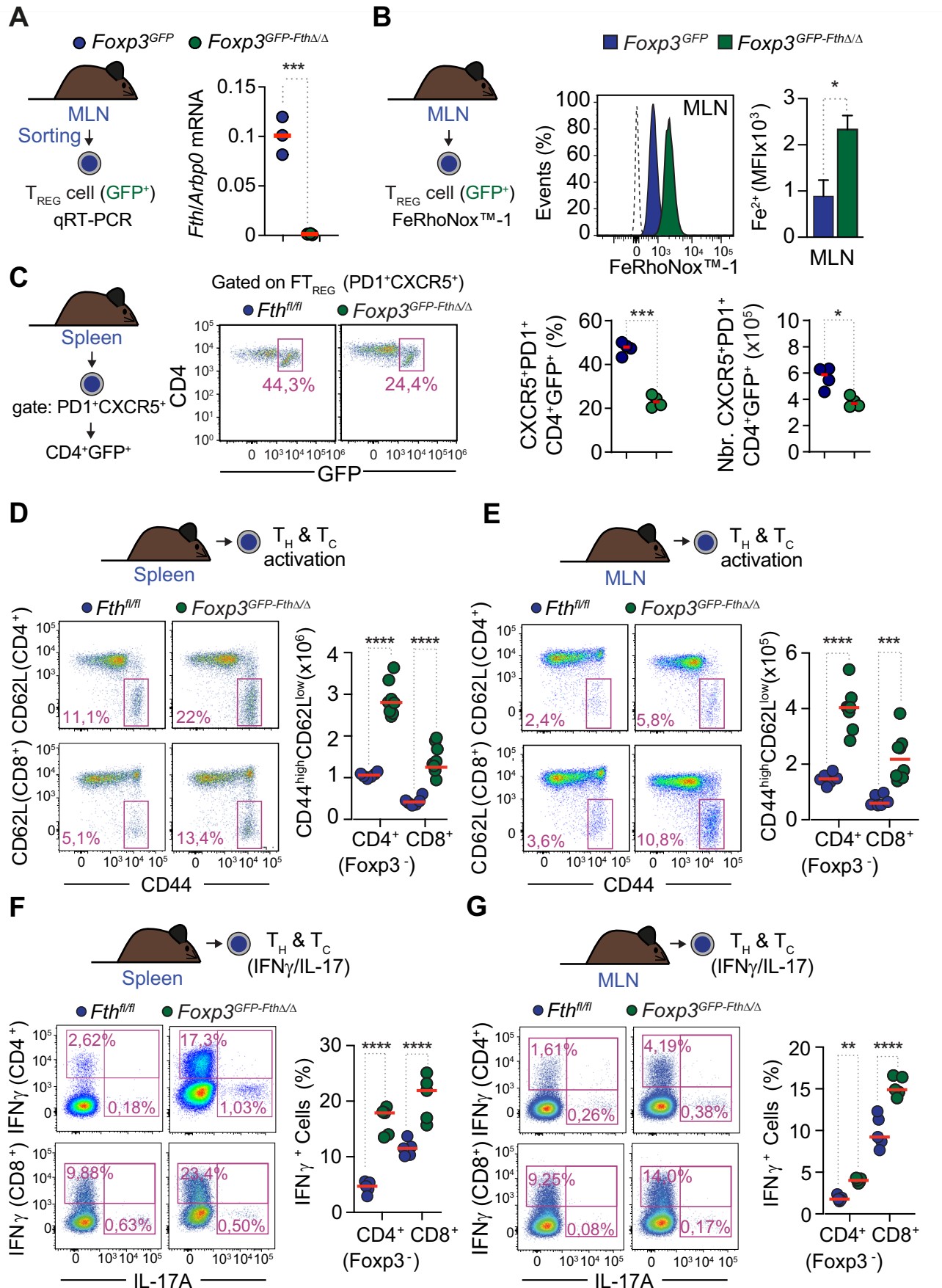

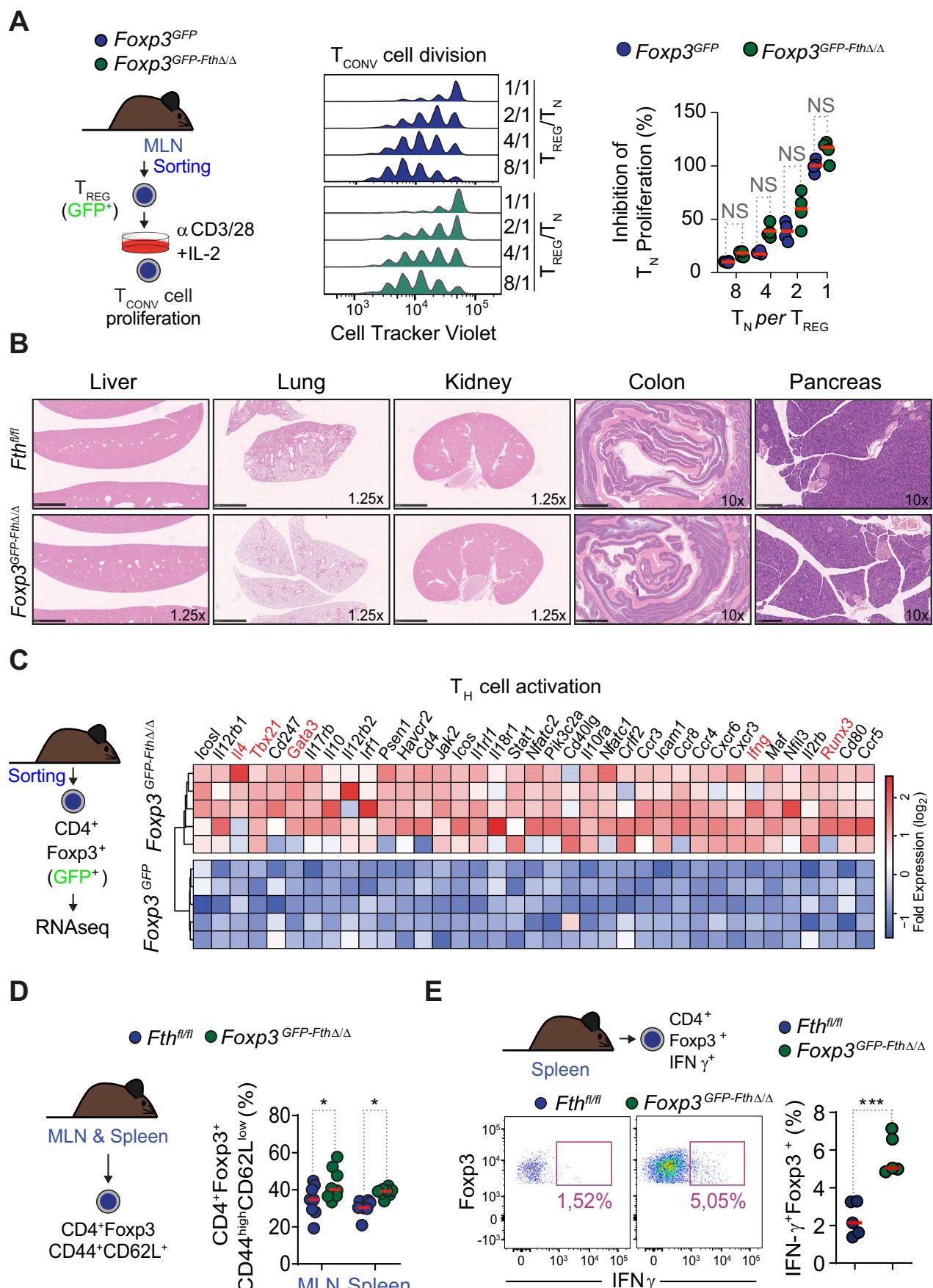

**Figure EV2.  FTH expression in T$_{REG}$ cells prevents systemic cellular inflammation.**

(A) Schematic representation of experimental approach (left panel) for (CD4$^+$CD25$^+$GFP$^+$) T$_{REG}$ cell sorting from the mesenteric lymph nodes (MLN) and coculture with conventional activated T cells (αCD3/28 + IL-2) to evaluate suppressive function of T$_{REG}$ cells. Representative flow cytometry proliferation histograms (Cell Tracer Violet) of in vitro suppression assay of mouse T$_N$ by different ratios of T$_{REG}$ cells (middle panel). Inhibition of T$_N$ cell proliferation quantified as percentage of undivided cells (right panel). Data from 1 out of 3 representative experiments, with similar trend. (B) Representative images of H&E-stained liver, lung, kidney, colon, and pancreas from $N = 3$–4 mice per genotype at 27–31 weeks after birth. (C) Schematic representation of the experimental approach (left panel) used to generate the Heatmap (right panel) of individual genes associated with T$_H$ effector function programs, differentially expressed in (CD4$^+$GFP$^+$) T$_{REG}$ cells sorted from *Foxp3$^{GFP\text{-}Fth\Delta/\Delta}$* vs. *Foxp3$^{GFP}$* mice (same experiment as Fig. 2A,B). (D) Schematic representation of the experimental approach (left panel) to evaluate the percentage (right panel) of (CD4$^+$Foxp3$^+$CD44$^{high}$CD62L$^{low}$) activated T$_{REG}$ cells in the MLN and spleen. Data from $N = 7$–8 mice per genotype, pooled from three independent experiments, with similar trend. (E) Schematic representation of the experimental approach used (top panel), representative flow cytometry dot plots (bottom left panel) and percentage (bottom right panel) of splenic (CD4$^+$Foxp3$^+$) IFNγ-secreting T$_{REG}$. Data from $N = 5$ mice per genotype, pooled from two independent experiments, with similar trend. Data information: Data in (A, D, E) represented as mean ± SD. Circles in (A) represent individual wells, and red bars are mean values. Circles in (D, E) represent individual mice, and red bars are mean values. *P* values in (A, D) calculated using Two-way ANOVA with Sidak's multiple comparison test and in (E) using unpaired *t* test with Welch's correction. NS not significant (*P* > 0.05), *\*P* < 0.05; *\*\*\*P* < 0.001. Source data are available online for this figure.

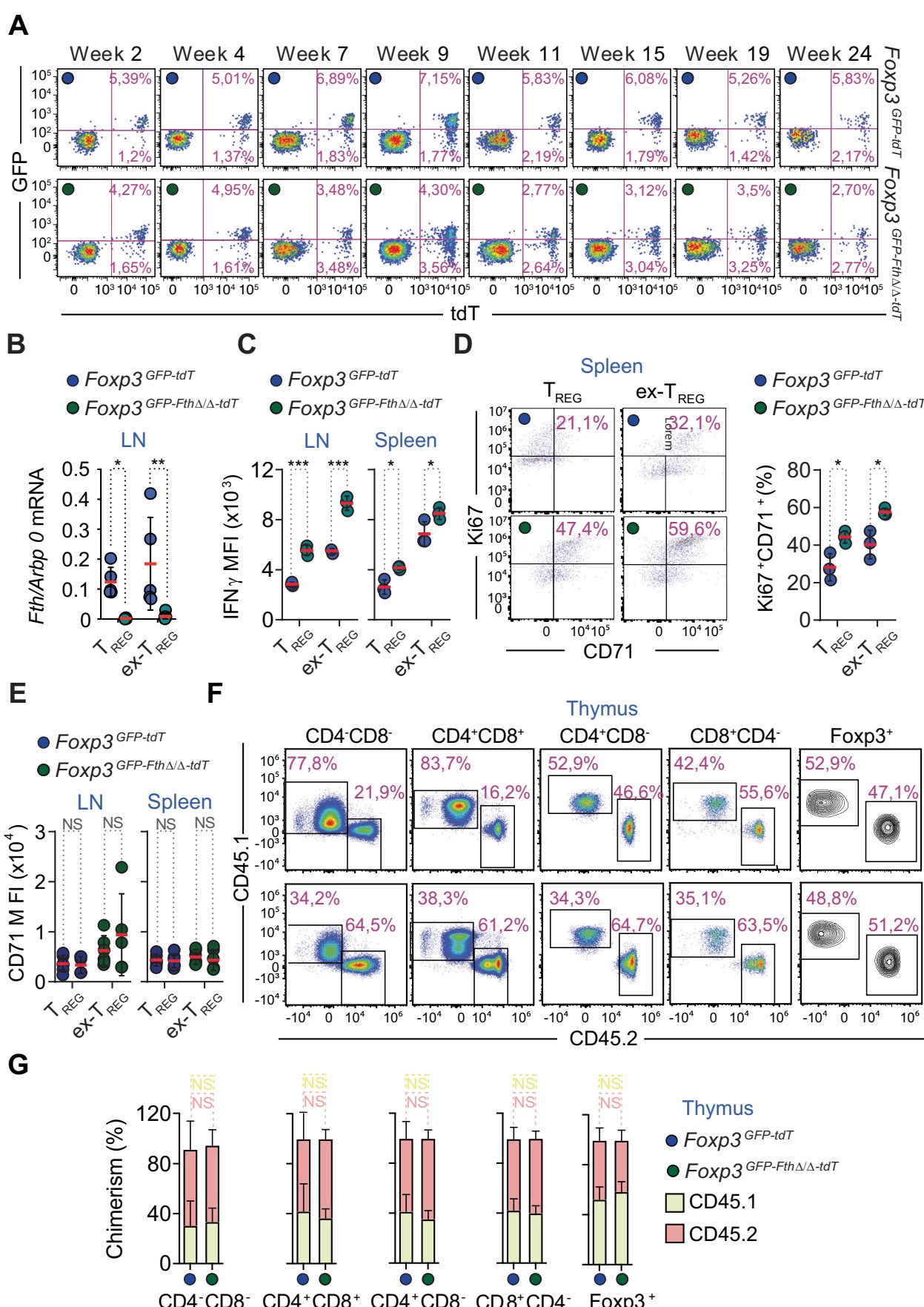

◄ **Figure EV3. FTH expression in T$_{REG}$ cells prevents T$_{REG}$ transdifferentiation into inflammatory T$_{REG}$ cells.**

(A) Representative flow cytometry dot plots of GFP and tdT expression in circulating CD4$^+$ cells (same experiment as Fig. 3A–D). Numbers in quadrants correspond to percentages of positive cells at the indicated weeks after birth. (B) Relative quantification of *Fth* mRNA, by qRT-PCR, normalized to *Arbp0* mRNA, in (CD4$^+$GFP$^+$tdT$^+$) T$_{REG}$ and (CD4$^+$GFP$^-$tdT$^+$) ex-T$_{REG}$ cells sorted from the lymph nodes (LN). Data from $N = 5$–7 mice per genotype, pooled from two experiments with similar trend. (C) Mean fluorescence intensity (MFI) of IFNγ in activated (CD4$^+$GFP$^+$tdT$^+$) T$_{REG}$ cells and (CD4$^+$GFP$^-$tdT$^+$) ex-T$_{REG}$ from the lymph nodes (LN) and spleen in the same experiment as (Fig. 3H). Data from $N = 3$ wells per genotype in one experiment, representative of 2 independent experiments with similar trend. (D) Representative flow cytometry dot plots (left panel) and corresponding percentage of Ki67$^+$CD71$^+$ (right panel) among splenic (CD4$^+$GFP$^+$TdT$^+$) T$_{REG}$ cells and (CD4$^+$GFP$^-$TdT$^+$) ex-T$_{REG}$ cells. Data from $N = 6$ mice per genotype, pooled from two independent experiments, with similar trend. (E) Mean fluorescence intensity (MFI) of CD71 expression in Ki67$^+$ T$_{REG}$ and ex-T$_{REG}$ cells from the lymph nodes (LN) and spleen, from the same experiments as in (D). (F) Representative flow cytometry dot plots and (G) corresponding percentages of CD45.1$^+$ and CD45.2$^+$ double negative (DN), double positive (DP) thymocytes, T$_H$ cells, cytotoxic T cells and (CD4$^+$Foxp3$^+$) T$_{REG}$ cells in the thymus from the same BM chimeric mice illustrated in (Fig. 4A–G). Data from $N = 11$–12 mice *per* genotype, pooled from two independent experiments with similar trend. Data information: Data in (B–E, G) are presented as mean ± SD, circles in (B, D, E) correspond to individual mice or individual wells (C) and red bars are mean values. *P* values in Panel (B–E, G) were calculated using two-way ANOVA with Sidak's multiple comparison test. NS not significant ($P > 0.05$), *$P < 0.05$; **$P < 0.01$; ***$P < 0.001$. Source data are available online for this figure.

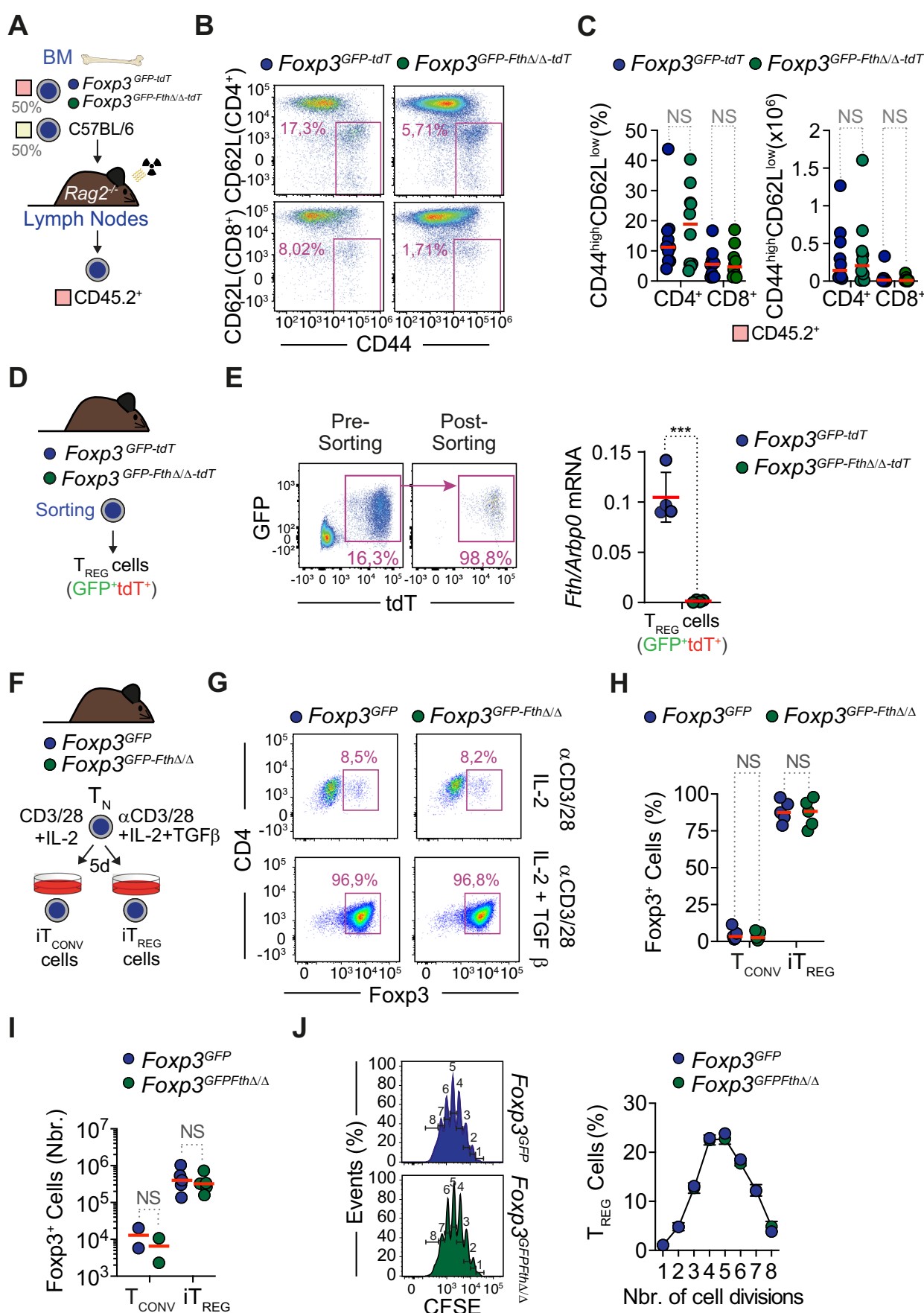

◀

**Figure EV4.  FTH acts in a non-cell-autonomous manner to prevent $T_{REG}$ cells from transdifferentiating into inflammatory $T_{REG}$ cells.**

(A) Schematic representation of the experimental approach used for flow cytometry analysis from the lymph nodes of BM chimeric mice (same experiment as Fig. 4A). (B, C) Representative flow cytometry dot plots (B), quantification of percentage (left panel) and number (right panel) (C) of live activated ($CD45.2^+CD4^+CD44^{high}CD62L^{low}$) and ($CD45.2^+CD8^+CD44^{high}CD62L^{low}$) cells in the lymph nodes of BM chimeric mice from (A). Data in (C) from $n = 11–12$ mice per genotype, pooled from 2 independent experiments, with similar trend. (D) Schematic representation of cell sorting for adoptive transfers in the experiment illustrated in Fig. 5A. (E) Representative flow cytometry dot plots of ($CD4^+GFP^+tdT^+$) $T_{REG}$ cells and relative level of *Fth* mRNA expression in ($CD4^+GFP^+tdT^+$) $T_{REG}$ cells (right panel) used for adoptive transfers in the experiment illustrated in Fig. 5A. (F) Schematic representation of experimental approach used for in vitro generation of induced $T_{REG}$ ($iT_{REG}$) cells and conventional $T_H$ ($T_{CONV}$) cells from sorted naive $T_H$ ($T_N$) cells, activated with anti-CD3 and anti-CD28 mAb plus IL-2 and TGFβ. (G) Representative flow cytometry dot plots of $iT_{REG}$ and $T_{CONV}$ cells, generated in (F). (H) Percentage (%) and (I) Number (Nbr.) of $Foxp3^+$ $T_{CONV}$ and $iT_{REG}$ cells, generated as described in (F). $N = 2–5$ independent experiments with similar trend. Each experiment corresponds to the average of different wells. (J) Representative flow cytometry carboxyfluorescein succinimidyl ester (CFSE) staining (left panel) and quantification of percentage (%) (right panel) of proliferating ($CD4^+Foxp3^+$) $iT_{REG}$ cells, generated as described in (F). Data from 3 to 6 technical replicates in 1 out of 3 independent experiments, with similar trend. Data information: Data in (C, E) are presented as mean ± SD, circles correspond to individual mice and red bars are mean values. Circles in (H–J) correspond to individual wells and red bars are mean values. *P* values in panels (C, H, I) were calculated using two-way ANOVA with Sidak's multiple comparison test. *P* values in (E) were calculated using Mann–Whitney test. NS not significant, ***$P < 0.001$. Source data are available online for this figure.

                                                     

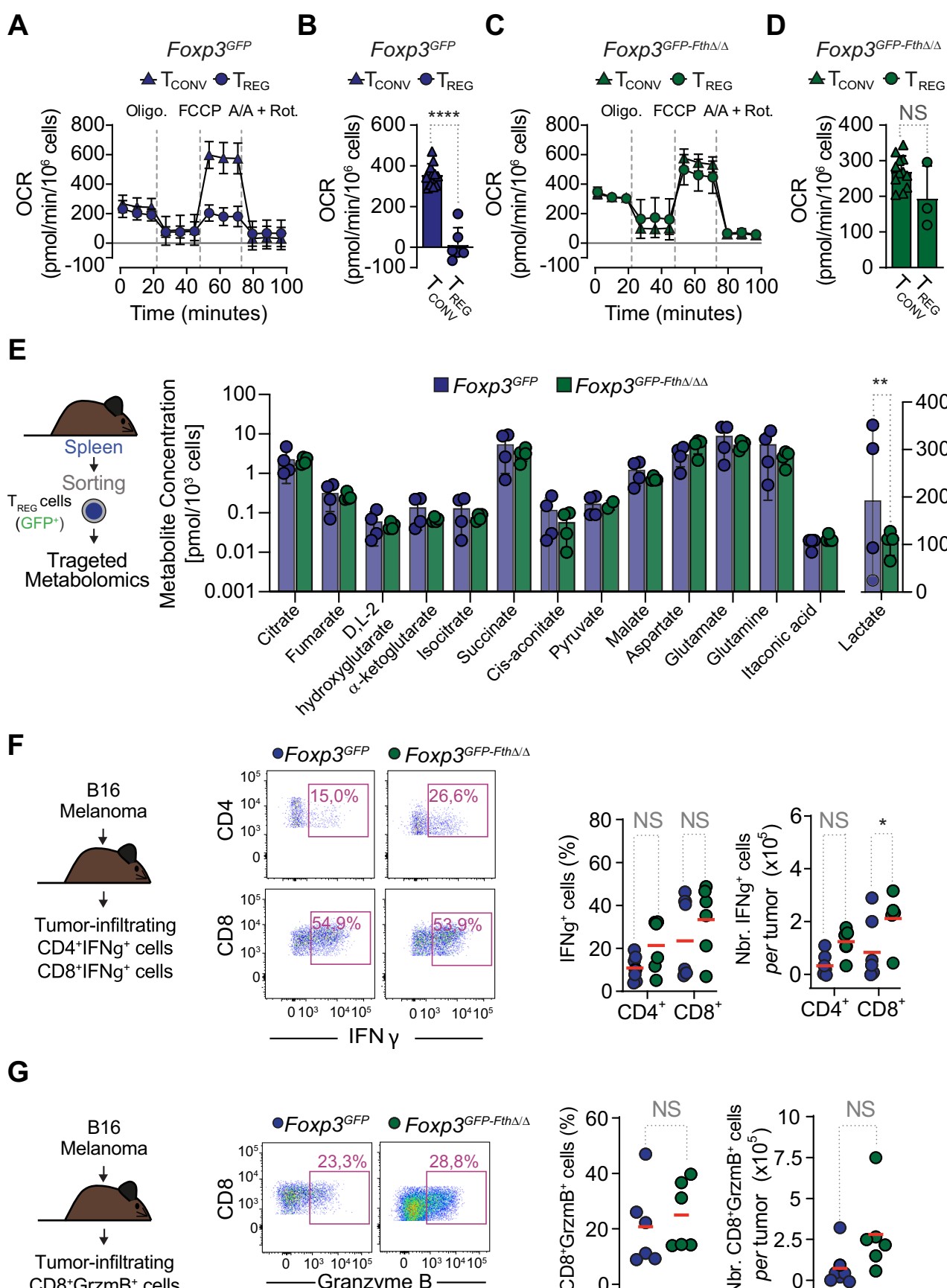

◀  **Figure EV5.  FTH regulates mitochondrial energy metabolism and CPG methylation in T_REG cells.**

(A) Oxygen consumption rate (OCR) in live splenic (CD4$^+$GFP$^+$) T$_{REG}$ cells and (CD4$^+$GFP$^-$) T$_{CONV}$ cells from *Foxp3$^{GFP}$* mice. (B) Quantification of spare respiratory capacity, from data represented in (A). (C) Oxygen consumption rate (OCR) in live splenic (CD4$^+$GFP$^+$) T$_{REG}$ cells and (CD4$^+$GFP$^-$) T$_{CONV}$ cells from *Foxp3$^{GFP\text{-}Fth\Delta/\Delta}$* mice. (D) Quantification of spare respiratory capacity, from data represented in (C). Data in (A–D) pooled from $N = 3$ mice per genotype, represented as mean ± SD. $N = 3$–5 technical replicates in 1 out of 3 independent experiments, with similar trend. Oligomycin (Oligo.), carbonilcyanide p-triflouromethoxyphenylhydrazone (FCCP), Antimycin A/Rotenone (A/A+Rot.). (E) Schematic representation of sorting of splenic (CD4$^+$GFP$^+$) T$_{REG}$ cells used for targeted metabolomics (left panel). Quantification of intermediate metabolites from targeted metabolomics analyzes of splenic T$_{REG}$ cells (right panel). Data from $N = 3$–4 mice per genotype in one experiment representative of 3 independent experiments with similar trend. (F) Schematic representation of the experimental approach used for flow cytometry analysis of tumor-infiltrating cells (left panel), representative flow cytometry dot plots (middle panel) and corresponding percentage and number (right panel) of live tumor-infiltrating (CD4$^+$IFNγ$^+$) T$_H$ cells (CD8$^+$IFNγ$^+$) T$_C$ cells, 3 weeks after tumor inoculation (2 × 10$^5$ B16 cells). Data from $N = 6$ mice per genotype, pooled from two independent experiments, with similar trend. (G) Schematic representation of the experimental approach used for flow cytometry analysis of tumor-infiltrating cells (left panel), representative flow cytometry dot plots (middle panel) and corresponding percentage and number (right panels) of live tumor-infiltrating (CD8$^+$GrzmB$^+$) T cells, 3 weeks after tumor inoculation (2 × 10$^5$ B16 cells). Data from $N = 6$ mice per genotype, pooled from two independent experiments, with similar trend. Data information: Circles and triangles in (A, C) correspond to mean values, circles, and triangles in (B, D) correspond to individual wells, and circles in (E–G) correspond to individual mice, and red bars are mean values. $P$ values in (A, C) calculated using two-way ANOVA with Bonferroni's (A, C) or Sidak's (E, F) multiple comparisons test, in (B, D, G) using unpaired $t$ test with Welch's correction. NS, not significant ($P > 0.05$); *$P < 0.05$; **$P < 0.01$; ****$P < 0.0001$. Source data are available online for this figure.

