## [Peer Review File · The EMBO Journal]

Ferritin heavy chain supports stability and function of the regulatory T cell lineage

Qian Wu, Ana Rita Carlos, Faouzi Brazza, Marie-Louise Bergman, Jamil Kitoko, Patricia Bastos-Amador, Eloy Cuadrado, Rui Martins, Bruna Oliveira, Vera Martins, Brendon Scicluna, Jonathan Landry, Ferris Jung, Temitope Ademolue, Mirko Peitzsch, Jose Almeida-Santos, Jessica Thompson, Silvia Cardoso, Pedro Ventura, Manon Slot, Stamatia Rontogianni, Vanessa Ribeiro, Vital Domingues, Inês Cabral, Sebastian Weis, Marco Groth, Cristina Ameneiro, Miguel Fidalgo, Fudi Wang, Jocelyne Demengeot, Derk Amsen, and Miguel Soares

Corresponding author(s): Miguel Soares (mpsoares@igc.gulbenkian.pt)

Review Timeline:

Submission Date:	26th Apr 23
Editorial Decision:	9th Jun 23
Appeal Received:	19th Jun 23
Editorial Decision:	27th Jun 23
Revision Received:	18th Dec 23
Editorial Decision:	23rd Jan 24
Revision Received:	11th Feb 24
Editorial Decision:	14th Feb 24
Revision Received:	15th Feb 24
Accepted:	20th Feb 24

Editors: Karin Dumstrei / Ioannis Papaioannou

Transaction Report:

Dear Dr. Soares,

Thank you for submitting your manuscript to The EMBO Journal. Your analysis has now been seen by three referees and I am afraid that the overall opinion is not a positive one.

As you can see from the comments below, the referees appreciate the dataset and the questions being addressed. However, they also raise concerns that I am afraid preclude publication here. The referees find that the analysis should be further extended, they question if FTH has a Treg specific function and also the interpretations regarding FTH deficiency in various disease contexts. Given the raised concerns, I am afraid that I can't offer to consider publication here.

I thank you for the opportunity to consider this manuscript. I am sorry that we cannot be more positive on this occasion.

Yours sincerely,

Karin Dumstrei, PhD
Senior Editor
The EMBO Journal

Referee #1:

The paper by Wu and co-workers, investigates the role of iron by ferritin H chain (FTH) in Treg cell immunobiology. Thanks to the use of complex mouse and molecular systems they showed that FTH expression in Treg cells is pivotal in immune regulation in the context of autoimmune neuroinflammation, malaria, and tumor growth. Mechanistically, FTH supports demethylation of the CpG-CNS islands in the Foxp3 gene, to stabilize its expression.

The paper and the results are original and of general relevance. This referee would suggest some further analyses to support the broad relevance in Treg biology:

- 1) Treg cell compartment in humans and mice represents the most proliferative subset at steady state. With this premise, given the role of iron in participating proliferation, is the FTH specific for Treg cells or rather its expression links with T cell proliferation independent by its role in Foxp3 expression? What is the expression level of FTH in TCR-stimulated proliferating Tconv and TCR-stimulated Treg? In other words is proliferation key to induce of FTH expression?
- 2) In the mouse is there any difference FTH levels in tTreg, pTreg and iTreg cells?
- 4) The EAE data are potentially interesting but the EAE-scores of the control mice are too low (barely reach a score of 1!) which is not a good standard for EAE, that generally with MOG should get at least around 2.5-3 in the controls (similar to the scores of the FTH-ko mice). Therefore, this referee would suggest to repeat the experiment and revisit the protocol to make sure in the controls to get a standard level of EAE to compare that of FTH ko mice;
- 5) Another point is the B16 tumor model, once again the tumor growth curve is somehow delayed in the controls, as the tumor appears visible after day 13, which is quite late for B16 standard curves. In this referee opinion is too late and the experiments repeated to have a normalized condition with B16 standard growth;
- 6) The data could be better supported showing also some human data of FTH expression in Treg vs Tconv, in resting and upon TCR stimulated conditions; also, what about frequency of Treg cells in humans (mainly women) with iron-deficiency anemia? Are their Tregs reduced? Do they suppress less? And what about frequency of autoimmune conditions in iron-deficiency anemia?

Referee #2:

Wu et al propose that ferritin H chain (FTH), acting upstream of TET dioxygenases, is necessary to maintain Treg function. They used several biochemical, flow cytometry and transcriptomics approaches along with disease models to show that absence of

FTH in Tregs impairs the maintenance of the Treg lineage and possibly compromises their ability to contribute to tissue homeostasis and propose a mechanism for Treg function through iron metabolism. While much work has been done to address the mechanisms by which Treg exert their function is still very much under debate how Tregs work. The work of Wu et al sheds light on how FTH contributes to the maintenance of Treg lineage and Treg mitochondria function. This manuscript offers an impressive amount of work with well-designed and well performed experiments. They draw important conclusions about the mechanism by which Tregs rely on FTH for their maintenance and consequently their function. However, at times the clarity and flow of the manuscript are compromised making it hard to follow what the authors really wanted to achieve. A few points (see below) need to be addressed and clarified that will allow and facilitate understanding by the readers. It is almost like there are two manuscripts in one. There is a detailed investigation of the molecular mechanisms by which FTH maintains Treg lineage under steady state on one side, and another side that investigates the functional consequences of absence of FTH in Tregs. The two parts are however somewhat disconnected.

Major comments:

1. The authors start by expanding previous observations that human Tregs express high levels of ferritin-regulatory proteins such as FTH and FTL and show that both human and mouse Tregs express higher levels of FTH compared to naïve or memory T cells.

Is Foxp3 expression sufficient to increase the levels of FTH? If the authors were to overexpress Foxp3 on naïve T cells or in vitro differentiate naïve T cells with TGF β and RA into Tregs would this be enough to increase FTH expression in Foxp3-expressing T cells? Could Foxp3 expression control FTH levels?

2. Regarding Fig. 1, does FTH deficiency in Tregs selectively impair thymically-derived natural Treg vs peripherally-induced Treg? This is important as FTH acts via CNS1 that is essential for pTreg development and because there is no difference in the population of thymic Tregs. Is the majority of the remaining Tregs found in the periphery in deleted mice nTregs? Or thymic output is the same despite of FTH deletion in Tregs, but maintenance is impaired as later suggested in figure 3 and 4? The authors could show this by staining for neuropilin-1 to distinguish nTregs from pTregs (PMID: 22966001, PMID: 22966003).

3. It looks like from the FACS plots shown in Fig 1E that the MFI of GFP is lower in FTH-deficient Tregs. The authors only return to this point in the last figure to reinforce the conclusion that FTH regulates Foxp3 by a cell intrinsic mechanism. For clarity, all the data that supports this conclusion could be shown together.

4. It is surprising that FTH deficiency in Tregs does not result in any overt inflammatory disease. What is the age of the mice at the time of analysis for the data shown in Fig. S1G? Were they older than 19 weeks of age (could not find this information in figure legend)? This is relevant as the drastic decrease in Foxp3⁺ occurs at this stage (as shown in figure 4B-G). In addition, it is hard to tell from the low magnification and low resolution images in Fig. S1G but it looks like the colon is a bit inflamed (or maybe more infiltrating cells in lamina propria?), which could indicate a selective defect in pTregs? The drastic effect on mLNs compared to spleen also suggest that since mLNs have more pTregs than spleen.

5. Why mice deficient in FTH in Tregs does not clear the parasite if they have more IFN γ -producing cells under steady state and also after infection (Fig 2G)? In addition, it looks like both groups can clear the parasites after all; only d6 seems to be a bit higher parasite load in RBC of FTH-deficient group. Can the authors explain how FTH-deficient Tregs can contribute to worse survival after plasmodium infection? It is not clear also why some parameters change in number, but not in % and others change in % but not in numbers. This should be explained in the results section.

6. The alternative explanation for improved tumor outcome (Fig 2J) in mice with selective deficiency of FTH in Tregs is that this deficiency impairs Treg accumulation (as shown in Fig 1 and in Fig 2K) and not a direct role of FTH expression in tumor progression as the authors suggest in the text. Is there an increase in tumor-infiltrating cytotoxic CD8 or CD4 T_H1 cells in deleted mice (besides CD25 that could reflect Tregs that lost Foxp3 expression but not CD25)? This could help explain the better tumor outcome in these mice. The authors could also show the numbers of tumor-infiltrating cells as they show in the other panels.

7. The authors should better discuss the functional data shown in Fig 2. They claim that FTH in Tregs maintain homeostasis by limiting neuroinflammation (EAE model) and infectious disease (limiting parasite load but not tissue destruction?) and repress tumor progression (melanoma model). However, the rest of the manuscript deals with the role of FTH in Tregs under steady-state conditions and there is no discussion about the disease models used except in the first paragraph of the discussion. As a suggestion for clarity and flow, the disease models could be shown in the end of the manuscript to illustrate the role of FTH deficiency in Tregs.

8. The RNAseq data and "fate-mapping" strategy in Figs 3 & 4 nicely show that FTH-deficiency in Tregs impairs foxp3-expression making them more inflammatory-like cells (which could help with the transition to the disease models shown in Fig 2). Thus, the conclusions they draw in page 4 saying that FTH is essential to maintain circulating Tregs is not completely accurate and should include the possibility that FTH is affecting the maintenance of the Treg profile, which could then be a good link for the RNAseq data. This is important because it has been shown that deletion of Foxp3 in Tregs can make them pathogenic/inflammatory in different contexts (PMID: 24238343, PMID: 17220874, PMID: 17220892, PMID: 19633673).

9. While the mechanism by which FTH maintains the circulatory and peripheral Treg pool appears to be cell intrinsic, it is not clear why the authors conclude in Fig. 5 it is irrespectively of bystander inflammation since there is no inflammation associated with FTH-deficiency in Tregs under steady state (according to Fig 1 and S1). Is it because the RNAseq data in Fig 3 shows upregulation of pathways related to inflammation such as IFN signaling, Th1 and Th2 and cell cycle? Can the authors clarify what do they mean by development of systemic inflammation in page 9 considering the absence of inflammatory disease under steady state in mice with FTH-deficient Tregs?

10. In Fig 6B-C, it is puzzling that the number of ex-Tregs is not different in the absence of FTH in Tregs. Is the frequency of ex-Tregs different between the two types of Tregs? The FACS plots would suggest so. Are the FTH-deficient Tregs dying more than the FTH-sufficient Tregs? Can the authors address this issue? There are a few occasions in which there appears to be a discrepancy in numbers and frequency of ex-Tregs and Tregs in these mice (figure 4F for example). Can the authors explain these discrepancies? It would help the reader understand what the read-outs mean. Does the high methylation of Foxp3 locus on ex-Tregs the only culprit? How can the authors exclude that FTH deficiency does not lead to increased cell death of Tregs?

Minor comments

1. Can the authors show a better gating for Tfh in Fig 1G? It would be better to show gated from CD4+ and show the whole CXCR5 and PD1 rather than pre-gate in CXCR5. Hard to interpret the results shown in 1G. There is an increase in % of Foxp3+ CXCR5+ PD1+ in deleted mice? But lower numbers? The authors do not discuss this further. Is it because most of the Tregs left in deleted mice are CXCR5+ PD1+? Also, in the text the authors refer to Fig. 1G as a decrease in % and number of CXCR5+ PD1+ Tregs which does not agree with what the panel shows.

2. The plots shown in Fig. 1H suggest that in mice carrying FTH-deficient Tregs there is a decrease in CD62Lhi CD44hi which encompasses central memory T cells. Did the authors quantify that? Is it because there are more Teff cells in these mice? Or is there a selective accumulation of central memory T cells in the tissue?

3. Can the authors also show the quantification of IL17+ IFN γ + DP cells in Fig. 2C as these cells are associated with EAE severity? This is important because 1) by the plots it looks like (at least in that representative plot) that the % of DP is higher in deleted mice; 2) IL17 difference, although significant, is not even a two-fold increase.

4. How the authors explain the frequency of ex-Tregs being similar between the 2 groups in the spleen of 19-30 week-old mice in Fig 4F? Is the data shown in figure 4E-F a pool of different ages (19-30 weeks old)? How many mice in each age group?

5. "irrespectively of bystander activation" appears in the text 3 times in a row: in page 8 and twice in page 9.

6. Can the authors explain why there is no decrease in Foxp3 in Fig 8H (but there is a decrease in GFP shown in Fig 8G) expression in splenic Tregs of mice carrying FTH-deficient Tregs?

Referee #3:

The role of Ferritin H chain protein (FTH) is investigated. In this well performed study it is found to be preferentially expressed in FOXP3 Tregs and plays a role in controlling the activation of the Tregs in the periphery in a cell autonomous way using mice with specific deletion of FTH in Tregs.

Technically the work is largely OK except some few issues.

1) The genetic background of the mice are not detailed in particular since it can be assumed that different backgrounds are mixed due to the breeding of different modified genes. In addition, it is not clarified if the investigated mice are littermates, which is needed in these types of experiments.

2) It is shown that FTH has increased expression in Tregs vs conventional T cells. I think there are two issues missing. Firstly, to determine whether the increased expression is simply due to that activation stage of the cells. Secondly, to quantify the expression in Treg compared to any other cell in the body, i.e. how unique is the overexpression in Tregs. I guess its not, but it should be shown and discussed.

My main issue is the value of the work. Obviously, it can be expected that FTH have profound effects in most cells and Tregs are likely to be affected as well. Maybe Treg have slightly higher expression than conventional T cells as they are maintained by self stimulation in the homeostasis. I wouldn't say that FTH regulate these cells, its just an essential gene as many other genes for the function of cells. The FTH is here artificially knocked out, its not a selected polymorphism. Possibly, it is interesting that the expression of FTH blocks differentiation to exTreg. It could also be of interest to know more on why FTH is not operating on the thymus levels when it operates at the peripheral level, i.e which activation demands is needed to require help from FTH, this is however not addressed. Following this, knocking out a physiological important protein in Tregs it does not come as a surprise that it enhances EAE and other of the disease models used.

The most surprising, and interesting, effect is the lack of effect of FTH in the thymus assuming that FTH play an essential redox regulatory role in all cells during the activation pressure. Thus, one would assume it play a role in Tregs during selection in the

thymus. However, this is not addressed.

** As a service to authors, EMBO Press provides authors with the possibility to transfer a manuscript that one journal cannot offer to publish to another EMBO publication or the open access journal Life Science Alliance launched in partnership between EMBO Press, Rockefeller University Press and Cold Spring Harbor Laboratory Press. The full manuscript and if applicable, reviewers' reports, are automatically sent to the receiving journal to allow for fast handling and a prompt decision on your manuscript. For more details of this service, and to transfer your manuscript please click on Link Not Available. **

Referee #1:

The paper by Wu and co-workers, investigates the role of iron by ferritin H chain (FTH) in Treg cell immunobiology. Thanks to the use of complex mouse and molecular systems they showed that FTH expression in Treg cells is pivotal in immune regulation in the context of autoimmune neuroinflammation, malaria, and tumor growth. Mechanistically, FTH supports demethylation of the CpG-CNS islands in the Foxp3 gene, to stabilize its expression. The paper and the results are original and of general relevance. This referee would suggest some further analyses to support the broad relevance in Treg biology:

Authors reply: Overall the comments from Reviewer 1 are very positive and can be addressed if a resubmission to the EMBO Journal would be considered.

1) Treg cell compartment in humans and mice represents the most proliferative subset at steady state. With this premise, given the role of iron in participating proliferation, is the FTH specific for Treg cells or rather its expression links with T cell proliferation independent by its role in Foxp3 expression? What is the expression level of FTH in TCR-stimulated proliferating Tconv and TCR-stimulated Treg? In other words is proliferation key to induce of FTH expression?

Reply to 1) We can add to the manuscript data showing that, as expected, FTH expression is not specific for T_{REG} cells but rather expressed at higher levels in T_{REG} cells vs. other T_H cell compartments. Please note that in the previous study leading to this manuscript (PMID: 29752063) it was already shown that “*The common Treg cell signature included ...Ferritin heavy and light chains (FTH1, FLT), ... expressed at high abundance in Treg compared to Tconv cells Figures 4A and 4B.*”

We also have data showing that FTH is induced in TCR-stimulated proliferating Tconv and TCR-stimulated T_{REG} cells. While FTH expression is increased upon T_{REG} cell and T_H cell activation/proliferation, FTH plays a rather unique function in T_{REG} cells in that it becomes essential to sustain Foxp3 expression and therefore T_{REG} cell lineage maintenance.

2) In the mouse is there any difference FTH levels in tTreg, pTreg and iTreg cells?

Reply to 2) We have not analyzed this but this can be easily addressed if a resubmission of the manuscript to the EMBO Journal would be considered.

3) The EAE data are potentially interesting but the EAE-scores of the control mice are too low (barely reach a score of 1!) which is not a good standard for EAE, that generally with MOG should get at least around 2.5-3 in the controls (similar to the scores of the FTH-ko mice). Therefore, this referee would suggest to repeat the experiment and revisit the protocol to make sure in the controls to get a standard level of EAE to compare that of FTH ko mice.

Reply to 3) We agree that the EAE-scores are below the usual standard (e.g., our previous work PMID: 17256058). However, milder models of EAE have been extensively used to address phenotypes that increase EAE severity (e.g. PMID: 31507583) as it is the case of our study. In the event that this manuscript would be reconsidered for publication at EMBO journal we could repeat the EAE experiments, if required. Alternatively, these experiments can be removed, in keeping with the overall assessment of the 3 reviewers towards focusing the manuscript.

4) Another point is the B16 tumor model, once again the tumor growth curve is somehow delayed in the controls, as the tumor appears visible after day 13, which is quite late for B16 standard curves. In this referee opinion is too late and the experiments repeated to have a normalized condition with B16 standard growth;

Reply to 4) We agree, in part, with this comment and would note that even though other studies have shown more precocious tumor growth (*i.e.*, starting around day 10 after inoculation; e.g. PMID: 26359984), our manuscript is in line with several other studies whereby tumor growth starts 13 days after inoculation (PMID: 24248371, PMID: 30333312, PMID: 30718660). There are many reasons why the kinetic of tumor development may vary from one study to another including microbiota composition, which we have not addressed in our study.

6) *The data could be better supported showing also some human data of FTH expression in Treg vs Tconv, in resting and upon TCR stimulated conditions; also, what about frequency of Treg cells in humans (mainly women) with iron-deficiency anemia? Are their Tregs reduced? Do they suppress less? And what about frequency of autoimmune conditions in iron-deficiency anemia?*

Reply to 5) A recent study showed that low iron diet increased T_{REG} cell number/proportion in mice, which is associated with an alleviated phenotype in a pristane-induced systemic lupus erythematosus model (PMID: 35296076). However, this study didn't consider regulation of iron metabolism specifically in T_{REG} cells. There are at least two other studies that provide further support to the idea that iron metabolism regulates systemic lupus erythematosus, in mice as well as in humans (PMID: 34880854; ;PMID: 35499082). Nevertheless, these studies do not show that iron metabolism regulates T_{REG} cells. Of note, autoimmune diseases are a major cause of anemia of chronic diseases, disrupting gastrointestinal iron absorption (PMID: 16973955) via the induction of hepcidin. This information can be included in the Discussion of our manuscript as proposed.

Referee #2:

Wu et al propose that ferritin H chain (FTH), acting upstream of TET dioxygenases, is necessary to maintain Treg function. They used several biochemical, flow cytometry and transcriptomics approaches along with disease models to show that absence of FTH in Tregs impairs the maintenance of the Treg lineage and possibly compromises their ability to contribute to tissue homeostasis and propose a mechanism for Treg function through iron metabolism. While much work has been done to address the mechanisms by which Treg exert their function is still very much under debate how Tregs work. The work of Wu et al sheds light on how FTH contributes to the maintenance of Treg lineage and Treg mitochondria function. This manuscript offers an impressive amount of work with well-designed and well performed experiments. They draw important conclusions about the mechanism by which Tregs rely on FTH for their maintenance and consequently their function. However, at times the clarity and flow of the manuscript are compromised making it hard to follow what the authors really wanted to achieve. A few points (see below) need to be addressed and clarified that will allow and facilitate understanding by the readers. It is almost like there are two manuscripts in one. There is a detailed investigation of the molecular mechanisms by which FTH maintains Treg lineage under steady state on one side, and another side that investigates the functional consequences of absence of FTH in Tregs. The two parts are however somewhat disconnected.

Authors reply: This reviewer is also positive about the manuscript stating that it “.. offers an impressive amount of work with well-designed and well performed experiments.” This is not trivial considering the very high technical standard required to provide novel information pertaining to the biology of T_{REG} cells. Having said that, this reviewer's requests are, to some extent, more extensive. Critical comments pertain to the clarity and flow of the manuscript,

which can be easily addressed by changing the order and flow of the manuscript. This would allow the molecular mechanisms by which FTH maintains T_{REG} lineage under steady state to be better linked with the functional consequences of absence of FTH in T_{REG} cells.

Major comments:

1. *The authors start by expanding previous observations that human Tregs express high levels of ferritin-regulatory proteins such as FTH and FTL and show that both human and mouse Tregs express higher levels of FTH compared to naïve or memory T cells. Is Foxp3 expression sufficient to increase the levels of FTH? If the authors were to overexpress Foxp3 on naïve T cells or in vitro differentiate naïve T cells with TGFβ and RA into Tregs would this be enough to increase FTH expression in Foxp3-expressing T cells? Could Foxp3 expression control FTH levels?*

Reply to 1) We found that TGFβ can, by itself, induce FTH expression and this data can be added to the current version of the manuscript. Of note, the *FTH* promoter contains at least one Foxp3 DNA binding site and as such it is possible that Foxp3 would also regulate *FTH* expression. This would argue for a positive feedback loop in which FTH enforces *Foxp3* expression and the later enforces the *FTH* expression. This can also be discussed in our manuscript.

2. *Regarding Fig. 1, does FTH deficiency in Tregs selectively impair thymically-derived natural Treg vs peripherally-induced Treg? This is important as FTH acts via CNS1 that is essential for pTreg development and because there is no difference in the population of thymic Tregs. Is the majority of the remaining Tregs found in the periphery in deleted mice nTregs? Or thymic output is the same despite of FTH deletion in Tregs, but maintenance is impaired as later suggested in figure 3 and 4? The authors could show this by staining for neuropilin-1 to distinguish nTregs from pTregs (PMID: 22966001, PMID: 22966003).*

Reply to 2) These experiments can be performed, considering the possibility of resubmitting to the EMBO Journal would be considered.

3. *It looks like from the FACS plots shown in Fig 1E that the MFI of GFP is lower in FTH-deficient Tregs. The authors only return to this point in the last figure to reinforce the conclusion that FTH regulates Foxp3 by a cell intrinsic mechanism. For clarity, all the data that supports this conclusion could be shown together.*

Reply to 3) We agree to change the flow of the article.

4. *It is surprising that FTH deficiency in Tregs does not result in any overt inflammatory disease. What is the age of the mice at the time of analysis for the data shown in Fig. S1G? Were they older than 19 weeks of age (could not find this information in figure legend)? This is relevant as the drastic decrease in Foxp3+ occurs at this stage (as shown in figure 4B-G). In addition, it is hard to tell from the low magnification and low resolution images in Fig. S1G but it looks like the colon is a bit inflamed (or maybe more infiltrating cells in lamina propria?), which could indicate a selective defect in pTregs? The drastic effect on mLN compared to spleen also suggest that since mLN have more pTregs than spleen.*

Reply to 4) We can better analyze the inflammation of the colon, as proposed.

5. *Why mice deficient in FTH in Tregs does not clear the parasite if they have more IFNγ-producing cells under steady state and also after infection (Fig 2G)? In addition, it looks like both groups can clear the parasites after all; only d6 seems to be a bit higher parasite load in RBC of FTH-deficient group. Can the authors explain how FTH-deficient Tregs can contribute to worse survival after plasmodium infection? It is not clear also why some*

parameters change in number, but not in % and others change in % but not in numbers. This should be explained in the results section.

Reply to 5) We have not elaborated enough on the description of our data and presumably therefore this reviewer has not fully understood the interpretation of our experiments. Namely the experiments show that despite a decrease in the pathogen burden mice succumb to malaria. In line with our previous work these experiments demonstrate that the expression of FTH in T_{REG} cells is essential to establish disease tolerance to malaria (PMID: 30673535).

6. The alternative explanation for improved tumor outcome (Fig 2J) in mice with selective deficiency of FTH in Tregs is that this deficiency impairs Treg accumulation (as shown in Fig 1 and in Fig 2K) and not a direct role of FTH expression in tumor progression as the authors suggest in the text. Is there an increase in tumor-infiltrating cytotoxic CD8 or CD4 Teff cells in deleted mice (besides CD25 that could reflect Tregs that lost Foxp3 expression but not CD25)? This could help explain the better tumor outcome in these mice. The authors could also show the numbers of tumor-infiltrating cells as they show in the other panels.

Reply to 6) These experiments can be performed, as requested, towards resubmission of the manuscript.

7. The authors should better discuss the functional data shown in Fig 2. They claim that FTH in Tregs maintain homeostasis by limiting neuroinflammation (EAE model) and infectious disease (limiting parasite load but not tissue destruction?) and repress tumor progression (melanoma model). However, the rest of the manuscript deals with the role of FTH in Tregs under steady-state conditions and there is no discussion about the disease models used except in the first paragraph of the discussion. As a suggestion for clarity and flow, the disease models could be shown in the end of the manuscript to illustrate the role of FTH deficiency in Tregs.

Reply to 7) We agree to change the flow of the article, as proposed.

8. The RNAseq data and "fate-mapping" strategy in Figs 3 & 4 nicely show that FTH-deficiency in Tregs impairs foxp3-expression making them more inflammatory-like cells (which could help with the transition to the disease models shown in Fig 2). Thus, the conclusions they draw in page 4 saying that FTH is essential to maintain circulating Tregs is not completely accurate and should include the possibility that FTH is affecting the maintenance of the Treg profile, which could then be a good link for the RNAseq data. This is important because it has been shown that deletion of Foxp3 in Tregs can make them pathogenic/inflammatory in different contexts (PMID: 24238343, PMID: 17220874, PMID: 17220892, PMID: 19633673).

Reply to 8) We agree to change the interpretation of the data, as proposed.

9. While the mechanism by which FTH maintains the circulatory and peripheral Treg pool appears to be cell intrinsic, it is not clear why the authors conclude in Fig. 5 it is irrespectively of bystander inflammation since there is no inflammation associated with FTH-deficiency in Tregs under steady state (according to Fig 1 and S1). Is it because the RNAseq data in Fig 3 shows upregulation of pathways related to inflammation such as IFN signaling, Th1 and Th2 and cell cycle? Can the authors clarify what do they mean by development of systemic inflammation in page 9 considering the absence of inflammatory disease under steady state in mice with FTH-deficient Tregs?

Reply to 9) We agree to clarify what we mean by the development of systemic inflammation in page 9.

10. In Fig 6B-C, it is puzzling that the number of ex-Tregs is not different in the absence of FTH in Tregs. Is the frequency of ex-Tregs different between the two types of Tregs? The FACS plots would suggest so. Are the FTH-deficient Tregs dying more than the FTH-sufficient Tregs? Can the authors address this issue? There are a few occasions in which there appears to be a discrepancy in numbers and frequency of ex-Tregs and Tregs in these mice (figure 4F for example). Can the authors explain these discrepancies? It would help the reader understand what the read-outs mean. Does the high methylation of *Foxp3* locus on ex-Tregs the only culprit? How can the authors exclude that FTH deficiency does not lead to increased cell death of Tregs?

Reply to 10) We can certainly explain and interpret the data in a clearer manner. The main conclusion is that while we observe a pronounced decrease in the percentage and number of T_{REG} cells in *Fth*-deficient mice vs. controls, the number and percentage of ex-T_{REG} cells remains unchanged. Most of our analysis was performed around or after 15-weeks of age, at which time the frequency of circulating ex-T_{REG} in *Fth*-deficient mice was similar to that of control mice. There is perhaps one exception at some timepoints, in blood, when the percentage of ex-T_{REG} cells in *Fth*-deficient mice increased, compared to controls.

As to the comment related to T_{REG} cell survival we can provide additional data using annexin V/PI staining to compare T_{REG} and ex-T_{REG} cell death in both genotypes *in vivo*. This data was not included in the initial submission because we didn't observe significant differences in cell death in the T_{REG} and ex-T_{REG} from *Fth*-deficient mice vs. control mice.

We also have data showing that *Fth* deficient iT_{REG} are more vulnerable to oxidative stress, triggered by exogenous Fe²⁺ or by H₂O₂, compared to control iT_{REG} cells. This is in keeping with the well-established cytoprotective effect exerted by FTH (PMID: 15537542 and PMID: 12958189). Taken together, these findings indicate that FTH regulates T_{REG} cell fate through multiple mechanisms, namely, cytoprotection and TET-driven T_{REG} cell lineage maintenance.

Please note that Figure S1F shows that the lack of *FTH* expression in T_{REG} cells is not associated with a proliferation defect *in vitro*, when compared to control T_{REG} cells. This suggests that FTH is not required to support T_{REG} cell proliferation or activated induced cell death.

Minor comments

1. Can the authors show a better gating for *Tfh* in Fig 1G? It would be better to show gated from CD4⁺ and show the whole CXCR5 and PD1 rather than pre-gate in CXCR5. Hard to interpret the results shown in 1G. There is an increase in % of *Foxp3*⁺ CXCR5⁺ PD1⁺ in deleted mice? But lower numbers? The authors do not discuss this further. Is it because most of the Tregs left in deleted mice are CXCR5⁺ PD1⁺? Also, in the text the authors refer to Fig. 1G as a decrease in % and number of CXCR5⁺ PD1⁺ Tregs which does not agree with what the panel shows.

2. The plots shown in Fig. 1H suggest that in mice carrying FTH-deficient Tregs there is a decrease in CD62L^{hi} CD44^{hi} which encompasses central memory T cells. Did the authors quantify that? Is it because there are more Teff cells in these mice? Or is there a selective accumulation of central memory T cells in the tissue?

3. Can the authors also show the quantification of IL17⁺ IFN γ ⁺ DP cells in Fig. 2C as these cells are associated with EAE severity? This is important because 1) by the plots it looks like (at least in that representative plot) that the % of DP is higher in deleted mice; 2) IL17 difference, although significant, is not even a two-fold increase.

4. How the authors explain the frequency of ex-Tregs being similar between the 2 groups in the spleen of 19-30 week-old mice in Fig 4F? Is the data shown in figure 4E-F a pool of different

ages (19-30 weeks old)? How many mice in each age group?

5. "irrespectively of bystander activation" appears in the text 3 times in a row: in page 8 and twice in page 9.

6. Can the authors explain why there is no decrease in *Foxp3* in Fig 8H (but there is a decrease in GFP shown in Fig 8G) expression in splenic Tregs of mice carrying FTH-deficient Tregs?

Reply to Minor comments) These comments can be easily addressed.

Referee #3:

The role of Ferritin H chain protein (FTH) is investigated. In this well performed study it is found to be preferentially expressed in FOXP3 Tregs and plays a role in controlling the activation of the Tregs in the periphery in a cell autonomous way using mice with specific deletion of FTH in Tregs. Technically the work is largely OK except some few issues.

Authors reply: In the assessment of our study this reviewer states that despite technical merits the biological insight provided by the study is questionable. We will argue bellow, why this is not the case.

1) *The genetic background of the mice are not detailed in particular since it can be assumed that different backgrounds are mixed due to the breeding of different modified genes. In addition, it is not clarified if the investigated mice are littermates, which is needed in these types of experiments.*

Reply to 1) The reviewer assumes that the mice used in our study are from different genetic backgrounds. This assumption, however, does not correspond to the reality, as indicated in the Key Resources table provided in the original submission. We can add a more detailed description of the mouse strains. Briefly, all strains used in our study are in C57BL/6J background, including the B6129S-Tg(*Foxp3* EGFP/*icre*)1aJbs/J strain that was backcrossed into C57BL/6/J background for over 10 generations before starting the experiments. We note that the *Rag2*^{-/-} mice used are in C57BL/6NTac background. However, this should bear no impact on the interpretation conclusions reached, given that these mice were used exclusively as recipients of bone marrows precursor cells.

The reviewer also raises concerns as to whether “*mice are littermates, which is needed in these types of experiments.*” We used both *Foxp3*GFP and *Fth*-fl/fl mice as controls of *Foxp3*GFP-*Fth*Δ/Δ mice. The *Fth*-fl/fl mice are littermates of *Foxp3*GFP-*Fth*Δ/Δ mice. The *Foxp3*GFP and *Foxp3*GFP-tdT mice used as controls for *Foxp3*GFP-*Fth*Δ/Δ and *Foxp3*GFP-*Fth*Δ/Δ-tdT mice, respectively, are not littermates. The reason for this being that the breeding strategy required to generate the very large number of mice used in our study would be associated with ethical, logistic and financial constrains that are not acceptable. Instead, *Foxp3*GFP and *Foxp3*GFP-tdT mice were co-housed with *Foxp3*GFP-*Fth*Δ/Δ and *Foxp3*GFP-*Fth*Δ/Δ-tdT mice, respectively, to allow for microbiota homogenization before performing the experiments.

2) *It is shown that FTH has increased expression in Tregs vs conventional T cells. I think there are two issues missing. Firstly, to determine whether the increased expression is simply du to that activation stage of the cells. Secondly, to quantify the expression in Treg compared to any other cell in the body, i.e. how unique is the overexpression in Tregs. I guess its not, but it should be shown and discussed.*

Reply to 2) Related to the issue: “*Firstly, to determine whether the increased expression is simply due to that activation stage of the cells.*” we have conclusive data, not included in the original submission of the manuscript, that addresses this concern. First, we found that FTH expression in iT_{REG} cells is significantly higher when compared to activated T_{CONV} cells, in the same culture. Second, we found that TGFβ can induce FTH in T_{REG} cells and T_{CONV} cells, but the relative level of Fth expression is higher in iT_{REG} cells vs. T_{CONV} cells. These data can be included, towards resubmission of the manuscript to the EMBO Journal.

The reviewer suggests that one should: “... *quantify the expression in Treg compared to any other cell in the body*” inferring that this would answer “*how unique is the overexpression in Tregs. I guess its not, but it should be shown and discussed.*” By accessing BioGPS database (<http://biogps.org/#goto=genereport&id=14319>) the reviewer will confirm that the relative level of Fth expression varies widely from one cell type to another under steady state conditions. The reviewer is kindly invited to confirm further this information by consulting in other data basis (<http://rstats.immgen.org/Skyline/skyline.html?gene=Fth1>) or (<https://tabularis.ds.czbiohub.org/>).

A comparison of the relative level of Fth expression in T_{REG} cells vs. cells in other parts of the body should be addressed through proteomic data analyzes, as performed in the previous study leading to this manuscript (PMID: 29752063). This study, referenced in our manuscript, shows that “*The common Treg cell signature included ...Ferritin heavy and light chains (FTH1, FLT), ... expressed at high abundance in Treg compared to Tconv cells Figures 4A and 4B.*”

As a final note, the reviewer ignores that FTH expression is under the control of the transcription factors NRF2 (e.g., PMID: 12435735; PMID: 30833408) and NF-kB (e.g., PMID: 7797515; PMID: 15537542), implying that FTH is an oxidative-stress responsive gene (*i.e.*, regulated by NRF2) that responds in addition to a variety of pro-inflammatory agonists (*i.e.*, regulated by NF-kB). It is not reasonable therefore to assume that FTH expression is a stable property when stating: “*how unique is the overexpression in Tregs. I guess its not, but it should be shown and discussed.*”

3) My main issue is the value of the work. Obviously, it can be expected that FTH have profound effects in most cells and Tregs are likely to be affected as well. Maybe Treg have slightly higher expression than conventional T cells as they are maintained by self stimulation in the homeostasis. I wouldn't say that FTH regulate these cells, its just an essential gene as many other genes for the function of cells. The FTH is here artificially knocked out, its not a selected polymorphism. Possibly, it is interesting that the expression of FTH blocks differentiation to exTreg. It could also be of interest to know more on why FTH is not operating on the thymus levels when it operates at the peripheral level, *i.e* which activation demands is needed to require help from FTH, this is however not addressed. Following this, knocking out a physiological important protein in Tregs it does not come as a surprise that it enhances EAE and other of the disease models used.

The most surprising, and interesting, effect is the lack of effect of FTH in the thymus assuming that FTH play an essential redox regulatory role in all cells during the activation pressure. Thus, one would assume it play a role in Tregs during selection in the thymus. However, this is not addressed.

Reply to 3) Although regulation of cellular iron metabolism may be critical to several types of cells the reviewer statement: “*I wouldn't say that FTH regulate these cells, its just an essential gene as many other genes for the function of cells.*” is not accurate, based on a large body of literature. Briefly, other laboratories used genetic loss of function approaches, similar to the one used in our manuscript, to shown that FTH controls systemic iron flux in enterocytes (PMID: 20816093) and hepatocytes (PMID: 19492434), sequesters labile iron to prevent oxidative stress/ferroptosis in renal proximal tubule epithelial cells (PMID: 24018561) and cardiomyocytes (PMID: 32349646) while supporting lymphocyte survival (PMID: 24586648). However, deletion of Fth in other cell compartments, including for example in myeloid cells or endothelial cells, is not associated with overt phenotypes, under steady state conditions or

even in response to acute systemic infections such as in malaria (PMID: 36735532). It is not reasonable therefore to assume that FTH is just an “*essential gene as many other genes for the function of cells.*”

The reviewer failed to appreciate that we show in Figure S1F that the lack of FTH expression in T_{REG} cells is not associated with a proliferation defect *in vitro*. This suggests that FTH is not required to support T_{REG} cell proliferation or activated induced cell death associated with T_{REG} cell activation/proliferation.

In our assessment, the major novelty of our study is the finding that regulation of iron metabolism by FTH plays a central role maintaining the developmental program enforcing T_{REG} cell identity. Another novel aspect of our study is the demonstration that regulation of iron metabolism by FTH regulates TET-driven DNA methylation. If, as proposed by the reviewer, this mechanism is also operational in other cell types our study would be enforced, providing a better understanding of how iron controls epigenetic programs defining specific developmental programs.

We agree with the reviewer that exploring further a role for Fth in thymic T_{REG} cells is very interesting and important. This could be addressed experimentally towards resubmission of the manuscript to the EMBO Journal.

Dear Miguel,

Thank you for sending me your point-by-point response to the concerns raised by the referees. I have now had a chance to take a careful look at it.

If you can extend the findings as indicated in your response including looking at the role of Fth in thymic Tregs then I am open to consider a revised version. I should add that I do need strong support from the referees to move forward with the manuscript and that I can't provide any guarantees regarding the outcome of the review process.

However, if you are open to significantly revised the manuscript then I will send it back to the referees. You can use the link below to submit the revised version.

With best wishes

Karin

Karin Dumstrei, PhD
Senior Editor
The EMBO Journal

We realize that it is difficult to revise to a specific deadline. In the interest of protecting the conceptual advance provided by the work, we recommend a revision within 3 months (25th Sep 2023). Please discuss the revision progress ahead of this time with the editor if you require more time to complete the revisions. Use the link below to submit your revision:

Referee #1:

The paper by Wu and co-workers, investigates the role of iron by ferritin H chain (FTH) in Treg cell immunobiology. Thanks to the use of complex mouse and molecular systems they showed that FTH expression in Treg cells is pivotal in immune regulation in the context of autoimmune neuroinflammation, malaria, and tumor growth. Mechanistically, FTH supports demethylation of the CpG-CNS islands in the *Foxp3* gene, to stabilize its expression. The paper and the results are original and of general relevance. This referee would suggest some further analyses to support the broad relevance in Treg biology:

1) Treg cell compartment in humans and mice represents the most proliferative subset at steady state. With this premise, given the role of iron in participating proliferation, is the FTH specific for Treg cells or rather its expression links with T cell proliferation independent by its role in *Foxp3* expression? What is the expression level of FTH in TCR-stimulated proliferating Tconv and TCR-stimulated Treg? In other words is proliferation key to induce of FTH expression?

Reply to 1) We thank the reviewer for the positive appraisal of our findings and manuscript. We cite a manuscript (PMID: 29752063) by two co-authors of the current manuscript, showing that ferritin heavy and light chains (FTH1, FLT) are expressed at higher levels in human T_{REG} cells, compared to T_{CONV} cells (please see Fig. 4A and 4B in PMID: 29752063). In the current manuscript we confirmed this finding and provide a comparison of the relative levels of FTH protein expression in human (Fig. 1A,B) and mouse T_{REG} vs. T_{CONV} cells (Fig. 1C,D). To address specifically the reviewer's request: "What is the expression level of FTH in TCR-stimulated proliferating Tconv and TCR-stimulated Treg?" we now provide additional data showing that FTH expression is induced in mouse iT_{REG} cells, generated upon TCR stimulation of T_N cells (see Fig. 1E-G). The following paragraph was added to the Results section:

"We monitored FTH expression in induced T_{REG} (iT_{REG}) cells, generated from mouse T_N cells activated *in vitro* with anti-CD3/CD28 mAb plus IL-2 and TGFβ (Chen *et al.*, 2003) (Fig. 1E). To this aim we used *Foxp3*^{GFP} T_{REG} cell reporter mice, in which a green fluorescent protein (GFP) humanized Cre-recombinase (GFP-hCre) coding sequence is inserted downstream of the *Foxp3* ATG translational start codon in a bacterial artificial chromosome (BAC) transgene carrying the intact *Foxp3* gene (*Foxp3*-GFP-hCre; referred herein as *Foxp3*^{GFP}) (Chen *et al.*, 2003; Zhou *et al.*, 2008). FTH protein expression was higher under culture conditions containing TGFβ and enriched in CD4⁺GFP⁺ iT_{REG} cells, compared to culture conditions lacking TGFβ and enriched in CD4⁺GFP⁻ T_{CONV} cells (Fig. 1F). A similar trend was observed for *Fth* mRNA, which was upregulated in iT_{REG} cells (Fig. 1G)."

2) In the mouse is there any difference FTH levels in tTreg, pTreg and iTreg cells?

Reply to 2) We analyzed the relative levels of *Fth* mRNA expression in T_{REG} cells, FACS-sorted from the thymus (tT_{REG} cells) or lymph nodes (pT_{REG} cells) of adult C57BL/6 mice as well from iT_{REG} cells generated *in vitro* (see Reply to comment 1). The data obtained is provided to the reviewer. While there is an overall tendency for the level of *Fth* mRNA to be higher in iT_{REG} cells and pT_{REG} cells vs. tT_{REG} cells, this is, however, not statistically significant. We conclude that there are no major differences in *Fth* mRNA expression between tT_{REG} and pT_{REG} cells.

FIGURE LEGEND (Reply 2). *Fth* mRNA expression normalized to *Arbp0* in CD4⁺GFP⁺CD25⁺CD8⁻B220⁻CD11b⁻ Treg cells FACS-sorted from the thymus and lymph nodes as well as in iT_{REG} cells, generated *in vitro* as detailed in Fig. 1E-G from N=3-5 mice from three experiments. P values were calculated with ordinary one-way ANOVA with Tukey's multiple comparisons test and show no significant differences.

3) The EAE data are potentially interesting but the EAE-scores of the control mice are too low (barely reach a score of 1!) which is not a good standard for EAE, that generally with MOG should get at least around 2.5-3 in the controls (similar to the scores of the FTH-ko mice). Therefore, this referee would suggest to repeat the experiment and revisit the protocol to make sure in the controls to get a standard level of EAE to compare that of FTH ko mice.

Reply to 3) We re-analyzed the EAE disease incidence and severity scores and the data illustrated now in Figure 8A-C is essentially identical to the original version of the manuscript with a slight increase in the individual “low” disease scores (*i.e.*, 1) responsible for the increase in EAE incidence in the control group. We agree that the EAE severity scores of control mice appear to be somehow below the 2.5-3 standard (*e.g.*, PMID: 17256058). This “milder” EAE however, allows to identify protective mechanisms against EAE without leading to mortality (PMID: 31507583), as it is the case in our current study. The following paragraph was added to the *Results* section to highlight this point.

“*Fth* deletion in T_{REG} cells led to an increase in EAE susceptible and severity (Fig. 8A-C), induced by an immunization protocol leading to low-grade disease severity in control mice (Fig. 8C). “

4) Another point is the B16 tumor model, once again the tumor growth curve is somehow delayed in the controls, as the tumor appears visible after day 13, which is quite late for B16 standard curves. In this referee opinion is too late and the experiments repeated to have a normalized condition with B16 standard growth;

Reply to 4) Although a more rapid onset of tumor growth (*i.e.*, starting around day 10 after inoculation; PMID: 26359984) is often observed, the kinetics of tumor growth shown in our manuscript are in line with several other studies (PMID: 24248371, PMID: 30333312, PMID: 30718660). Possible reasons for this, include: i) number of tumor cells inoculated, ii) mouse microbiota composition, etc. To address this reviewer concern we performed two independent experiments re-testing whether *Fth* deletion in T_{REG} cells reduces B16 tumor progression. We used *Foxp3*^{GFP-FthΔ/Δ} vs. control littermate *Foxp3*^{GFP} mice, rather than *Fth*^{fl/fl} mice, to ensure that the phenotype observed (*i.e.*, reduced tumor progression in *Foxp3*^{GFP-FthΔ/Δ} vs. *Fth*^{fl/fl} mice) is not attributed to a putative change in the microbiota composition of *Foxp3*^{GFP-FthΔ/Δ} vs. *Fth*^{fl/fl}

mice. The data obtained, provided to the reviewer, is identical in terms of the kinetics of tumor progression (tumor growth from day 10-12 onwards) and a marked reduction of tumor progression in *Foxp3*^{GFP-FthΔΔ} vs. control littermate *Foxp3*^{GFP} mice. The additional data fully supports that *Fth*-deletion in T_{REG} cells hinders tumor progression (see Fig. 8H). We conclude that under the husbandry condition of the Gulbenkian Institute animal house the standard kinetics of B16 tumor growth is similar to the one reported in the original submission of our manuscript, with tumor progression starting 10-13 days after inoculation.

FIGURE LEGEND (Reply 4). Relative tumor size, 10 to 19 days after inoculation of B16-F10-luc2 tumor cells (2×10^5). Data is represented as mean \pm SD from N=8 mice per genotype, pooled from 2 independent experiments, with similar trend. P values were determined using Two-way ANOVA with Sidak's multiple comparisons test. *P < 0.05; ****P < 0.0001.

5) *The data could be better supported showing also some human data of FTH expression in Treg vs Tconv, in resting and upon TCR stimulated conditions; also, what about frequency of Treg cells in humans (mainly women) with iron-deficiency anemia? Are their Tregs reduced? Do they suppress less? And what about frequency of autoimmune conditions in iron-deficiency anemia?*

Reply to 5) To the comment “*The data could be better supported showing also some human data of FTH expression in Treg vs Tconv, in resting and upon TCR stimulated conditions*” we cite previous findings (PMID: 29752063), by two co-authors of our manuscript, showing that ferritin heavy (H) and light (L) chains (FTH, FTL) are expressed at higher levels in human T_{REG} cells, compared to T_{CONV} cells (*please see Fig. 4A and 4B in PMID: 29752063*). We believe that while the comment “*What about frequency of Treg cells in humans (mainly women) with iron-deficiency anemia? Are their Tregs reduced? Do they suppress less? And what about frequency of autoimmune conditions in iron-deficiency anemia?*”, is pertinent, addressing this experimentally falls beyond the scope of the current manuscript. Instead we addressed this comment by the following paragraphs, added to the *Introduction* and *Discussion* sections, strengthening the notion that regulation of Fe metabolism plays a critical role in the control of T_{REG} cell function in the context of different autoimmune conditions *in vivo*.

Introduction:

“Cellular Fe import, via the transferrin receptor 1 (*TFR1/CD71*), supports T_H type 1 (T_H1) cell immunity and its regulation by induced T_{REG} (iT_{REG}) cells (Voss *et al*, 2023) as well as antibody responses to vaccination (Frost *et al*, 2021; Jiang *et al*, 2019) and immunity against infection by pathogens such as *Plasmodium*, the causative agent of malaria (Wideman *et al*, 2023).”

Discussion:

“This is consistent with regulation of Fe metabolism modulating the incidence and severity of autoimmune conditions, as demonstrated for systemic lupus erythematosus (Gao *et al.*, 2022a). Consistent with our findings, this was linked to modulation of T_{REG} (Gao *et al.*, 2022a), T_H17 (Teh *et al.*, 2021) and FT_H (Gao *et al.*, 2022b) cells and was associated with regulation of DNA demethylation (Gao *et al.*, 2022b; Teh *et al.*, 2021). However, whether regulation of Fe metabolism in T_{REG} cells affects systemic lupus erythematosus was, to the best of our knowledge, not established (Gao *et al.*, 2022a).

“In conclusion, regulation of intracellular Fe metabolism by FTH is essential to maintain T_{REG} cell lineage and function *in vivo*, reflecting how intracellular catalytic Fe controls the activity of TET dioxygenases that sustain FOXP3 transcription and support T_{REG} cell lineage identity. Moreover, FTH might regulate other iron-dependent mechanisms supporting T_{REG} function, for example by enforcing the expression of c-Maf in T_{REG} cells (Zhu *et al.*, 2023) that control immunological tolerance to the microbiota (Xu *et al.*, 2018). We propose that targeting Fe metabolism pharmacologically maybe considered when manipulating T_{REG} cells for therapeutic purposes, either to enhance T_{REG} cell function in the context of immune-mediated inflammatory diseases or to dampen T_{REG} cell function as in the context of cancer therapies.”

Referee #2:

Wu et al propose that ferritin H chain (FTH), acting upstream of TET dioxygenases, is necessary to maintain Treg function. They used several biochemical, flow cytometry and transcriptomics approaches along with disease models to show that absence of FTH in Tregs impairs the maintenance of the Treg lineage and possibly compromises their ability to contribute to tissue homeostasis and propose a mechanism for Treg function through iron metabolism. While much work has been done to address the mechanisms by which Treg exert their function is still very much under debate how Tregs work. The work of Wu et al sheds light on how FTH contributes to the maintenance of Treg lineage and Treg mitochondria function. This manuscript offers an impressive amount of work with well-designed and well performed experiments. They draw important conclusions about the mechanism by which Tregs rely on FTH for their maintenance and consequently their function. However, at times the clarity and flow of the manuscript are compromised making it hard to follow what the authors really wanted to achieve. A few points (see below) need to be addressed and clarified that will allow and facilitate understanding by the readers. It is almost like there are two manuscripts in one. There is a detailed investigation of the molecular mechanisms by which FTH maintains Treg lineage under steady state on one side, and another side that investigates the functional consequences of absence of FTH in Tregs. The two parts are however somewhat disconnected.

Reply to the general comments. We thank the reviewer for the positive appraisal of our manuscript. As suggested, we modified the sequence of the data presentation, in an effort to better clarify and facilitate understanding of our findings. Namely, we introduced the effect of Fth on Foxp3 MFI in *Fig. 1* as suggested. We also moved the description of the functional consequences of *Fth* deletion in T_{REG} cells (*Fig. 2* in the original version) to the last section of the manuscript (*Fig. 8*).

Major comments:

1. *The authors start by expanding previous observations that human Tregs express high levels of ferritin-regulatory proteins such as FTH and FTL and show that both human and mouse Tregs express higher levels of FTH compared to naïve or memory T cells. Is Foxp3 expression sufficient to increase the levels of FTH? If the authors were to overexpress Foxp3 on naïve T cells or in vitro differentiate naïve T cells with TGFβ and RA into Tregs would this be enough to increase FTH expression in Foxp3-expressing T cells? Could Foxp3 expression control FTH levels?*

Reply to 1) As requested, we tested whether differentiation of naïve T cells into Foxp3-expressing iT_{REG} cells *in vitro* is associated with an increase in FTH expression. We found that indeed the generation of iT_{REG} cells, via TCR stimulation of T_N cells, is associated with the induction of FTH expression. This data is illustrated in *Fig. 1E-G*. The following paragraph describing this data was added to the *Results* section:

“We monitored FTH expression in induced T_{REG} (iT_{REG}) cells, generated from mouse T_N cells activated *in vitro* with anti-CD3/CD28 mAb plus IL-2 and TGFβ (Chen *et al.*, 2003) (*Fig. 1E*). To this aim we used *Foxp3*^{GFP} T_{REG} cell reporter mice, in which a green fluorescent protein (GFP) humanized Cre-recombinase (GFP-hCre) coding sequence is inserted downstream of the *Foxp3* ATG translational start codon in a bacterial artificial chromosome (BAC) transgene carrying the intact *Foxp3* gene (*Foxp3*-GFP-hCre; referred herein as *Foxp3*^{GFP}) (Chen *et al.*, 2003; Zhou *et al.*, 2008). FTH protein expression was higher under culture conditions containing TGFβ and enriched in CD4⁺GFP⁺ iT_{REG} cells, compared to culture conditions lacking TGFβ and enriched in CD4⁺GFP⁻ T_{Conv} cells (*Fig. 1F*). A similar trend was observed for *Fth* mRNA, which was overexpressed in iT_{Reg} cells (*Fig. 1G*).”

As to the interesting comment: “*Is Foxp3 expression sufficient to increase the levels of FTH?*” we found that the *FTH* promoter contains at least one Foxp3 DNA binding site and as such it is indeed possible that Foxp3 would regulate *FTH* expression. This would argue for a positive feedback loop in which FTH enforces *Foxp3* expression and the later enforces the *FTH* expression. We added the following paragraph to the *Discussion* section of the manuscript to address this point:

“Moreover, the *FTH* promoter contains at least one Foxp3 DNA binding site and as such it is possible that Foxp3 would regulate *FTH* expression directly. This would argue for a positive feedback loop in which FTH enforces *Foxp3* expression, via the regulation of TET dioxygenases, and the later enforces the *FTH* expression transcriptionally. This hypothesis remains however to be tested experimentally.”

2. *Regarding Fig. 1, does FTH deficiency in Tregs selectively impair thymically-derived natural Treg vs peripherally-induced Treg? This is important as FTH acts via CNS1 that is essential for pTreg development and because there is no difference in the population of thymic Tregs. Is the majority of the remaining Tregs found in the periphery in deleted mice nTregs? Or thymic output is the same despite of FTH deletion in Tregs, but maintenance is impaired as later suggested in figure 3 and 4? The authors could show this by staining for neuropilin-1 to distinguish nTregs from pTregs (PMID: 22966001, PMID: 22966003).*

Reply to 2) We addressed this point experimentally and added the data to *Fig. 1L*. The following paragraph was introduced to the *Results* section to describe this data:

“Thymic and peripheral T_{REG} cell development give rise to tT_{REG} and iT_{REG} cells, expressing neuropilin1 (Nrp1) or not, respectively (Weiss *et al*, 2012; Yadav *et al*, 2012). The frequency and number of Nrp1⁺ tT_{REG} cells and Nrp1⁻ iT_{REG} cells was reduced, to the same extent, as assessed in the MLN (Fig. 1L) of *Foxp3*^{GFP-Fth $\Delta\Delta$} vs. control *Foxp3*^{GFP} mice. Similar data was obtained in the spleen (*data not shown*). This suggests that regulation of Fe metabolism by FTH is required for maintenance of thymic-derived and peripherally induced T_{REG} cells.”

The data for splenic tT_{REG} and pT_{REG} cells was not included in the manuscript, due to editorial space constrains, and is referred to as *data not shown*. The data is provided hereby to the reviewer:

FIGURE LEGEND (Reply 2): Schematic representation of the experimental approach (left), and representative flow cytometry dot plots (upper right panels) with corresponding quantification of percentage (%) and number (Nbr.) (bottom right panels) of live (TCR β ⁺CD4⁺GFP⁺) Nrp1⁺ T_{REG} cells and Nrp1⁻ pT_{REG} cells in the spleen. N=8 mice *per* genotype, pooled from 2 independent experiments with similar trend. Data from the same mice as in (Fig. 1L). Circles represent individual mice and red bars are mean values. P values were determined using Two-way ANOVA with Sidak's multiple comparisons test. NS (P>0.05); ***P < 0.001.

3. It looks like from the FACS plots shown in Fig 1E that the MFI of GFP is lower in FTH-deficient Tregs. The authors only return to this point in the last figure to reinforce the conclusion that FTH regulates *Foxp3* by a cell intrinsic mechanism. For clarity, all the data that supports this conclusion could be shown together.

Reply to 3) As suggested by the reviewer the “flow of the article” (see also *reply to the general comments*) was edited to have *Figure 2* from the previous version of the manuscript (*i.e.*, immune mediated inflammatory models) now as *Figure 8*. As proposed by the reviewer, we describe in *Figure 1K* that *Foxp3*-GFP signal (MFI) is reduced in *Fth*-deleted T_{REG} cells, as assessed in the thymus, spleen and MLN. The following paragraph was introduced in the *Results* section to describe this data:

“We noticed that the relative level of GFP expression, reporting on *Foxp3* transcription under the control of an intact *Foxp3* promoter (Chen *et al.*, 2003; Zhou *et al.*, 2008), was reduced in thymic, splenic and LN T_{REG} cells from *Foxp3*^{GFP-Fth $\Delta\Delta$} vs. control *Foxp3*^{GFP} mice (Fig. 1K). These observations suggest that FTH regulates *Foxp3* transcription in the

thymus as well as in circulating T_{REG} cells, which is not sufficient however to interfere with thymic T_{REG} cell output.”

We cannot move the remaining of *Figure 7* (from previous version) to *Figure 1* (in the current version) because the data in *Figure 7E-H* was obtained using experimental approaches introduced only in *Figure 3* and *4*, namely, BM chimeras and/or T_{REG} cells expressing tdT (an additional Rosa26-tandem dimer (td) Tomato-Flox-stop-Flox allele that drives the expression of td Tomato (tdT) by Cre-driven excision of a Flox-stop-Flox cassette, under the control of *Foxp3* regulatory regions), respectively.

We added to *Figure 7C*, data showing that FTH regulates *Foxp3* transcription (*i.e.*, *Foxp3*-driven GFP expression) in tT_{REG} and pT_{REG} cells. The following paragraph was edited in the *Results* section:

“Having established that regulation of Fe metabolism by FTH regulates *Foxp3* transcription in the thymus as well as in circulating T_{REG} cells (*Fig. 1K*) we compared the relative level of *Foxp3*-GFP expression in Nrp1⁺ tT_{REG} and Nrp1⁻ pT_{REG} cells. GFP expression was reduced in both tT_{REG} and pT_{REG} cells, as assessed in the MLN from *Foxp3*^{GFP-Fth $\Delta\Delta$} vs. control *Foxp3*^{GFP} mice (*Fig. 7C*). Similar data was obtained for splenic T_{REG} cells (*data not shown*). Considering that GFP is expressed under the control of a bacterial artificial chromosome (BAC) transgene carrying the intact *Foxp3* gene (Chen *et al.*, 2003; Zhou *et al.*, 2008), these observations suggest that FTH is essential to sustain *Foxp3* transcription in tT_{REG} and pT_{REG} cells.”

Similar data was obtained for the spleen, not included in the manuscript due to editorial space constrains. This data is referred to as *data not shown* and is provided hereby to the reviewer:

FIGURE LEGEND (Reply 3): Schematic representation of the experimental approach (left panel) used to monitor the expression of the GFP transgene in splenic (CD4⁺GFP⁺Nrp1⁺) tT_{REG} cells and (CD4⁺GFP⁺Nrp1⁻) pT_{REG} cells. Representative flow cytometry histogram of GFP (middle panel). Relative quantification of GFP expression (right panel), represented as mean fluorescence intensity (MFI). Data from N=8 mice *per* genotype, pooled from 2 independent experiments with similar trend. Circles represent individual mice and red bars to mean values. P values in were calculated using two-way ANOVA with Sidak’s multiple comparison test. ***P < 0.001.

4. It is surprising that FTH deficiency in Tregs does not result in any overt inflammatory disease. What is the age of the mice at the time of analysis for the data shown in Fig. S1G? Were they older than 19 weeks of age (could not find this information in figure legend)? This is relevant as the drastic decrease in *Foxp3*⁺ occurs at this stage (as shown in figure 4B-G). In addition, it is hard to tell from the low magnification and low resolution images in Fig. S1G but it looks like the colon is a bit inflamed (or maybe more infiltrating cells in lamina propria?), which could indicate a selective defect in pTregs? The drastic effect on mLNs compared to spleen also suggest that since mLN have more pTregs than spleen.

Reply to 4) We added information to EV2 legend, specifying that mice were analyzed at 27 weeks after birth. Further histopathological examination confirmed that there is no difference in the morphological structure of the different organs analyzed among the two genotypes. There are also no apparent differences in the relative number of lymphocytes and plasma cells in the lamina propria of the colon, remaining within what would be expected in a healthy/control mouse. In both genotypes, the lung is the organ more affected by inflammatory lesions with detection of alveolar macrophages filled with crystalline eosinophilic material, consistent with described in several mouse strains (PMID: 26973378).

5. *Why mice deficient in FTH in Tregs does not clear the parasite if they have more IFN γ -producing cells under steady state and also after infection (Fig 2G)? In addition, it looks like both groups can clear the parasites after all; only d6 seems to be a bit higher parasite load in RBC of FTH-deficient group. Can the authors explain how FTH-deficient Tregs can contribute to worse survival after plasmodium infection? It is not clear also why some parameters change in number, but not in % and others change in % but not in numbers. This should be explained in the results section.*

Reply to 5) We did not discuss these observations in detail and presumably therefore the data interpretation was not made sufficiently clear in the original version of our manuscript. Our findings support the hypothesis put forward in the *Results* section: “*Regulation of T_{REG} cell lineage stability by FTH impacts on the severity of infectious diseases (Soares et al, 2017).*”. The following paragraph was added to the *Discussion* section to better contextualize this interpretation :

“In support of our hypothesis, *Foxp3^{GFP-Fth Δ/Δ}* mice succumbed to *Plasmodium chabaudi chabaudi* (*Pcc*) AS infection, as compared to *Foxp3^{GFP}* mice that survived and cleared the parasite (*Fig. 8E*). However, the number of circulating infected red blood cells (*i.e.*, parasite burden) was lower at the peak of infection, in *Foxp3^{GFP-Fth Δ/Δ}* vs. control *Foxp3^{GFP}* mice (*Fig. 8E*). The lethal outcome of *Pcc* infection was associated with a reduction in the frequency (but not the number) of splenic T_{REG} cells (*Fig. 8F*), as well as with an increase in the number (but not the frequency) of IFN γ -expressing T_H cells in the spleen (*Fig. 8G*) and MLN (*data not shown*). This suggests that FTH expression in T_{REG} cells is essential to restrain immune-driven pathology underlying the development of severe presentation of malaria, emphasizing the critical involvement of T_{REG} cells in the control of the pathogenesis of severe malaria (Walther *et al*, 2005). Moreover, these findings illustrate how dysregulation of Fe metabolism promotes the pathogenesis of severe malaria (Ferreira *et al*, 2008; Gozzelino *et al*, 2012; Ramos *et al*, 2019; Seixas *et al*, 2009; Wu *et al*, 2023).”

The data for IFN γ -expressing T_H cells in MLN was removed from the manuscript (see previous version) due to editorial space constrains. This data is referred to as *data not shown*.

6. *The alternative explanation for improved tumor outcome (Fig 2J) in mice with selective deficiency of FTH in Tregs is that this deficiency impairs Treg accumulation (as shown in Fig 1 and in Fig 2K) and not a direct role of FTH expression in tumor progression as the authors suggest in the text. Is there an increase in tumor-infiltrating cytotoxic CD8 or CD4 Teff cells in deleted mice (besides CD25 that could reflect Tregs that lost Foxp3 expression but not CD25)? This could help explain the better tumor outcome in these mice. The authors could also show the numbers of tumor-infiltrating cells as they show in the other panels.*

Reply to 6) Two additional experiments were performed, confirming that *Fth* deletion in T_{REG} cells promotes tumor clearance in *Foxp3*^{GFP-FthΔΔ} vs. control *Foxp3*^{GFP} mice. The data obtained, provided to the reviewer, fully supports the data included in the original submission of manuscript (Fig. 8H).

FIGURE LEGEND (Reply 6): Relative tumor size, 10 to 19 days after inoculation of B16-F10-luc2 tumor cells (2×10^5). Data is represented as mean \pm SD (N=8 mice per genotype), pooled from 2 independent experiments, with similar trend. P value is determined using Two-way ANOVA analysis with Sidak's multiple comparisons test. *P < 0.05; ****P < 0.0001.

As proposed by the reviewer we analyzed tumor-infiltrating effector T cells. While we did not find a consistent increase in the frequency and/or number of tumor-infiltrating CD8 or CD4 T effector cells there was an increase in the number of tumor-infiltrating CD8⁺IFN γ ⁺ cells when *Fth* was deleted in T_{REG} cells. We added this data to *EV5F,G* with the following paragraph in the *Results* section:

“The frequency and number of activated CD4⁺IFN γ ⁺ T_H cells isolated from B16 melanomas was similar in *Foxp3*^{GFP-FthΔΔ} vs. control *Foxp3*^{GFP} mice (*EV5F*). In contrast, the number of activated CD8⁺IFN γ ⁺ T_C cells was higher in B16 melanomas from *Foxp3*^{GFP-FthΔΔ} vs. control *Foxp3*^{GFP} mice (*EV5F*). This tendency was also observed for CD8⁺granzymeB⁺ T_C cells, *albeit* not statistically significant (*EV5F*). This suggests that regulation of Fe metabolism by FTH in T_{REG} cells supports tumor progression, via a mechanism that hinders anti-tumor immunity.”

The following paragraph was also added to the *Discussion* section:

“Dysregulation of Fe metabolism in *Fth*-deleted T_{REG} cells was associated with better control of tumor progression (Fig. 8H), consistent with a relative reduction of tumor-infiltrating T_{REG} cells (Fig. 8I) and a more pronounced activation and/or infiltration of activated T effector cells (Fig. 8J), including CD8⁺IFN γ ⁺ T_C cells (*EV5F*). While the mechanism via which FTH expression in T_{REG} cells promotes tumor progression is not clear, these observations are consistent with regulation of Fe metabolism in the tumor microenvironment impacting on tumor progression (Alaluf *et al*, 2020; Consonni *et al*, 2021).”

7. The authors should better discuss the functional data shown in Fig 2. They claim that FTH in Tregs maintain homeostasis by limiting neuroinflammation (EAE model) and infectious disease (limiting parasite load but not tissue destruction?) and repress tumor progression (melanoma model). However, the rest of the manuscript deals with the role of FTH in Tregs under steady-state conditions and there is no discussion about the disease models used except in the first paragraph of the discussion. As a suggestion for clarity and flow, the disease

models could be shown in the end of the manuscript to illustrate the role of FTH deficiency in Tregs.

Reply to 7) We agree with the Reviewer's proposition to change the flow of the article, placing the disease models at the end of the manuscript. We also added a more detailed discussion of the disease models in the *Discussion* section.

Autoimmunity:

"*Fth* deletion in T_{REG} cells led to an increase in EAE susceptible and severity (Fig. 8A-C), induced by an immunization protocol leading to low-grade disease severity in control mice (Fig. 8C). This was associated with a higher accumulation of activated T_{H1} and T_{H17} cells as well as pathogenic IFN γ ⁺IL-17A⁺ T_H cells (Duhon *et al.*, 2013) in the central nervous system (Fig. 8D), likely originating from auto-reactive T_N cells and/from ex-T_{REG} cells that lost *Foxp3* expression to become inflammatory and presumably pathogenic (Bailey-Bucktrout *et al.*, 2013). This is consistent with regulation of Fe metabolism modulating the incidence and severity of autoimmune conditions, as demonstrated for systemic lupus erythematosus (Gao *et al.*, 2022a). Consistent with our findings, this was linked to modulation of T_{REG} (Gao *et al.*, 2022a), T_{H17} (Teh *et al.*, 2021) and FT_H (Gao *et al.*, 2022b) cells and was associated with regulation of DNA demethylation (Gao *et al.*, 2022b; Teh *et al.*, 2021). However, whether regulation of Fe metabolism in T_{REG} cells affects systemic lupus erythematosus was, to the best of our knowledge, not established (Gao *et al.*, 2022a)."

Malaria:

"*Fth*-deletion in T_{REG} cells increased susceptibility to malaria (Fig. 8E), consistent with dysregulation of Fe metabolism promoting malaria lethality (Ferreira *et al.*, 2008; Ramos *et al.*, 2022; Ramos *et al.*, 2019; Wu *et al.*, 2023). This was associated with an increase in host parasite burden (Fig. 8E), in keeping with T_{REG} cells being essential to counter the pathogenesis of severe presentations of malaria while limiting immune-driven resistance mechanisms driving parasite clearance (Hisaeda *et al.*, 2004; Kurup *et al.*, 2017; Walther *et al.*, 2005). We infer that the protective effect of T_{REG} cells against malaria acts via a mechanism that is not associated with a reduction of the host pathogen burden, a defense strategy known as disease tolerance (Medzhitov *et al.*, 2012; Soares *et al.*, 2017). Presumably the mechanism(s) via which FTH acts in T_{REG} cells to establish disease tolerance to malaria is multifactorial, restraining unfettered immune activation to prevent the pathogenesis of severe forms of malaria."

Cancer

"Dysregulation of Fe metabolism in *Fth*-deleted T_{REG} cells was associated with better control of tumor progression (Fig. 8H), consistent with a relative reduction of tumor-infiltrating T_{REG} cells (Fig. 8I) and a more pronounced activation and/or infiltration of activated T effector cells (Fig. 8J), including CD8⁺IFN γ ⁺ T_C cells (EV5F). While the mechanism via which FTH expression in T_{REG} cells promotes tumor progression is not clear, these observations are consistent with regulation of Fe metabolism in the tumor microenvironment impacting on tumor progression (Alaluf *et al.*, 2020; Consonni *et al.*, 2021)."

8. The RNAseq data and "fate-mapping" strategy in Figs 3 & 4 nicely show that FTH-deficiency in Tregs impairs *foxp3*-expression making them more inflammatory-like cells (which could help with the transition to the disease models shown in Fig 2). Thus, the conclusions they draw in page 4 saying that FTH is essential to maintain circulating Tregs is not completely accurate and should include the possibility that FTH is affecting the maintenance of the Treg profile, which could then be a good link for the RNAseq data. This is important because it has been shown that deletion of *Foxp3* in Tregs can make them pathogenic/inflammatory in different contexts (PMID: 24238343, PMID: 17220874, PMID: 17220892, PMID: 19633673).

Reply to 8) We have taken into consideration that "...deletion of *Foxp3* in Tregs can make them pathogenic/inflammatory in different contexts" and discussed this point, referencing PMID: 24238343, PMID: 17220874 and PMID: 17220892. The following paragraphs were added to the manuscript:

Introduction:

"Sustained *FOXP3* transcription maintains T_{REG} cell lineage stability (Williams & Rudensky, 2007), avoiding transdifferentiation towards pro-inflammatory T helper (T_H) cells (Gavin *et al*, 2007; Morikawa *et al*, 2014)."

Discussion:

"*Fth* deletion in T_{REG} cells led to an increase in EAE susceptible and severity (Fig. 8A-C), induced by an immunization protocol leading to low-grade disease severity in control mice (Fig. 8C). This was associated with a higher accumulation of activated T_H1 and T_H17 cells as well as pathogenic IFN γ ⁺IL-17A⁺ T_H cells (Duhon *et al.*, 2013) in the central nervous system (Fig. 8D), likely originating from auto-reactive T_N cells and/from ex-T_{REG} cells that lost *Foxp3* expression to become inflammatory and presumably pathogenic (Bailey-Bucktrout *et al.*, 2013)."

9. While the mechanism by which FTH maintains the circulatory and peripheral Treg pool appears to be cell intrinsic, it is not clear why the authors conclude in Fig. 5 it is irrespectively of bystander inflammation since there is no inflammation associated with FTH-deficiency in Tregs under steady state (according to Fig 1 and EV1). Is it because the RNAseq data in Fig 3 shows upregulation of pathways related to inflammation such as IFN signaling, Th1 and Th2 and cell cycle? Can the authors clarify what do they mean by development of systemic inflammation in page 9 considering the absence of inflammatory disease under steady state in mice with FTH-deficient Tregs?

Reply to 9) The term "bystander" inflammation was used to refer to the activation of T_H and T_C cells resulting from *Fth* deletion in T_{REG} cells. We edited the manuscript as described below to clarify this point and refer to the development of systemic inflammation to describe the activation of T_H and T_C cells resulting from *Fth* deletion in T_{REG} cells:

"This confirms that FTH sustains the number of circulating T_{REG} cells, via a cell-autonomous mechanism that acts irrespectively of the systemic inflammatory response associated with *Fth* deletion in T_{REG} cells from *Foxp3*^{GFP-*Fth* $\Delta\Delta$} mice (EV1D-G)."

&

"This suggests that FTH maintains peripheral T_{REG} cell lineage stability, via a cell-autonomous mechanism, irrespectively of the systemic inflammatory response associated with *Fth* deletion in T_{REG} cells from *Foxp3*^{GFP-*Fth* $\Delta\Delta$} mice (EV1D-G)."

&

“We then asked whether FTH regulates gene expression in T_{REG} cells, irrespectively of the systemic inflammatory response associated with *Fth* deletion in T_{REG} cells from *Foxp3*^{GFP-Fth $\Delta\Delta$} mice (EV1D-G).”

&

“This suggests that FTH exerts cell-autonomous control of T_{REG} cell redox homeostasis, irrespectively of the systemic inflammatory response associated with *Fth* deletion in T_{REG} cells from *Foxp3*^{GFP-Fth $\Delta\Delta$} mice (EV1D-G).”

10. *In Fig 6B-C, it is puzzling that the number of ex-Tregs is not different in the absence of FTH in Tregs. Is the frequency of ex-Tregs different between the two types of Tregs? The FACS plots would suggest so. Are the FTH-deficient Tregs dying more than the FTH-sufficient Tregs? Can the authors address this issue? There are a few occasions in which there appears to be a discrepancy in numbers and frequency of ex-Tregs and Tregs in these mice (figure 4F for example). Can the authors explain these discrepancies? It would help the reader understand what the read-outs mean. Does the high methylation of Foxp3 locus on ex-Tregs the only culprit? How can the authors exclude that FTH deficiency does not lead to increased cell death of Tregs?*

Reply to 10). *Figure 3A-D suggests that Fth deletion in T_{REG} cells is associated with a progressive increase of the ratio of ex-T_{REG} vs. T_{REG} cells over time. However, this is attributed primarily to a decrease in the number of T_{REG} cells and to a lesser extent to the accumulation of ex-T_{REG} cells. Our interpretation is that FTH is essential not only to restrain T_{REG} cells from transdifferentiating into inflammatory ex-T_{REG} cells but also to support the viability of proliferating T_{REG} and ex-T_{REG} cells. This interpretation is consistent with the lack of autoimmune lesions in *Foxp3*^{GFP-Fth $\Delta\Delta$} mice, lacking Fth expression in T_{REG} cells. The following paragraphs were introduced in the:*

Results:

“Taken together these observations suggest that FTH is essential to restrain T_{REG} cells from transdifferentiating into inflammatory ex-T_{REG} cells.”

Discussion:

“One possible interpretation is that FTH is essential not only to restrain T_{REG} cells from transdifferentiating into inflammatory ex-T_{REG} cells but also to support the viability of highly proliferating T_{REG} and ex-T_{REG} cells. This is consistent with the lack of autoimmune lesions in *Foxp3*^{GFP-Fth $\Delta\Delta$} mice (EV2B). We note however, that *Fth*-deletion in T_{REG} cells does not compromise the T_{REG} cell thymic output (Fig. 1H) nor the generation and proliferation of iT_{REG} cells *in vitro* (EV4F-J).”

“This is consistent with FTH exerting additional effects, beyond the regulation of TET dioxygenases, preventing cellular stress from compromising the viability of the inflammatory ex-T_{REG} cells that would otherwise elicit autoimmunity.”

Minor comments

11. *Can the authors show a better gating for Tfh in Fig 1G? It would be better to show gated from C1D4+ and show the whole CXCR5 and PD1 rather than pre-gate in CXCR5. Hard to interpret the results shown in 1G. There is an increase in % of Foxp3+ CXCR5+ PD1+ in deleted mice? But lower numbers? The authors do not discuss this further. Is it because*

most of the Tregs left in deleted mice are CXCR5+ PD1+? Also, in the text the authors refer to Fig. 1G as a decrease in % and number of CXCR5+ PD1+ Tregs which does not agree with what the panel shows.

Reply to 11) As requested, we gated follicular T_H cells as TCRβ⁺CD4⁺CXCR5⁺PD1⁺ to quantify CD4⁺GFP⁺ FT_{REG} cells. The data showing the percentage and number of FT_{REG} cells is illustrated in EV1C. The following paragraph was added to the *Results* section:

“The frequency and number of CD4⁺GFP⁺CXCR5⁺PD1⁺ follicular T_{REG} (FT_{REG}) cells was also decreased in the spleen of *Foxp3*^{GFP-FthΔΔ} vs. control *Foxp3*^{GFP} mice (EV1C). These observations suggest that regulation of intracellular Fe by FTH is required to sustain the number of circulating T_{REG} and FT_{REG} cells in, without interfering with thymic T_{REG} cell output.”

The gating strategy is provided hereby to the reviewer:

FIGURE LEGEND (Reply 11): Schematic representation of the protocol used (left panel) and representative flow cytometry dot plots of the gating strategy (TCRβ⁺CD4⁺CXCR5⁺PD1⁺) used for analyzes of live splenic follicular CD4⁺GFP⁺ FT_{REG} cells in EV1C.

12. The plots shown in Fig. 1H suggest that in mice carrying FTH-deficient Tregs there is a decrease in CD62L^{hi} CD44^{hi} which encompasses central memory T cells. Did the authors quantify that? Is it because there are more Teff cells in these mice? Or is there a selective accumulation of central memory T cells in the tissue?

Reply to 12) We found no differences in the percentage of CD62L^{hi}CD44^{hi} T_H and T_C cells in the spleen and MLN from *Foxp3*^{GFP-FthΔΔ} vs. control *Fth*^{fl/fl} mice (see figure below). This suggests that the increase Teff cells, observed in *Foxp3*^{GFP-FthΔΔ} vs. control *Fth*^{fl/fl} mice, is not attributed to a selective accumulation of central memory CD62L^{hi}CD44^{hi} T_H and T_C cells.

FIGURE LEGEND (Reply 12): A,B Schematic representation of experimental approach (upper panels), representative flow cytometry dot plots (lower left panels) and corresponding quantification of percentage (%) of live CD44^{high}CD62L^{high} T_H and T_C cells in the spleen and MLN. Data from n=6-8 mice *per* genotype, pooled from 4 independent experiments, with similar trend. Circles correspond to individual mice and red bars to mean values. P values were calculated using two-way ANOVA with Sidak's multiple comparison test. NS P>0.05.

13. *Can the authors also show the quantification of IL17⁺ IFN γ ⁺ DP cells in Fig. 2C as these cells are associated with EAE severity? This is important because 1) by the plots it looks like (at least in that representative plot) that the % of DP is higher in deleted mice; 2) IL17 difference, although significant, is not even a two-fold increase.*

Reply to 13): As requested, we quantified the percentage of IL17⁺IFN γ ⁺ T_H cells in spinal cord and brain after the induction of EAE. As predicted by the reviewer, there is indeed a significant increase in the percentage of IL17⁺IFN γ ⁺ T_H cells in the spinal cord and brain from MOG₃₅₋₅₅-immunized *Foxp3^{GFP-FthΔΔ}* vs. control *Fth^{fl/fl}* mice. The data for spinal cord was included in Figure 8D and the following paragraph was added to the *Results* section:

“The frequency of T_H cells expressing IFN γ , IL-17A and IL17⁺IFN γ ⁺ T_H cells was higher in the spinal cord of MOG₃₅₋₅₅-immunized *Foxp3^{GFP-FthΔΔ}* vs. control *Fth^{fl/fl}* mice (Fig. 8D). Similar results were obtained in the brain (*data not shown*). This suggests that FTH expression in T_{REG} cells limits the activation, proliferation and/or infiltration of self-reactive T_H type 1 (T_H1) and T_H type 17 (T_H17) cells as well as pathogenic IL17⁺IFN γ ⁺ T_H cells (Duhon *et al.*, 2013) into the central nervous system, in response to MOG₃₅₋₅₅ immunization.”

The following paragraph was added to the *Discussion* section:

“This was associated with a higher accumulation of activated T_H1 and T_H17 cells as well as pathogenic IFN γ ⁺IL-17A⁺ T_H cells (Duhon *et al.*, 2013) in the central nervous system (Fig. 8D), likely originating from auto-reactive T_N cells and/from ex-T_{REG} cells that lost Foxp3 expression to become inflammatory and presumably pathogenic (Bailey-Bucktrout *et al.*, 2013).”

The data for the brain, referred to as *data not shown*, is not included in the manuscript due to editorial space constrains and is enclosed hereby.

FIGURE LEGEND (Reply to 13): Experimental approach (right panel), representative flow cytometry dot plots (middle panel) and corresponding quantification of the relative percentage of activated T_H1 (CD3⁺CD4⁺IFN-γ⁺), T_H17 (CD3⁺CD4⁺IL-17A) and IFN-γ⁺IL-17A⁺ T_H cells in the spinal cord, 22 days after MOG₃₅₋₅₅ immunization. Data represented as mean ± SD, circles correspond to individual mice and red bars to mean values. P values determined using unpaired t-test with Welch's correction; *P < 0.05; **P < 0.01.

14. How the authors explain the frequency of ex-T_{REG} cells being similar between the 2 groups in the spleen of 19-30 week-old mice in Fig 4F? Is the data shown in figure 4E-F a pool of different ages (19-30 weeks old)? How many mice in each age group?

Reply to 14) The similar frequency of ex-T_{REG} cells in *Foxp3*^{GFP-FthΔ/Δ} vs. *Foxp3*^{GFP} mice is likely due to the instability of ex-T_{REG} cells in *Fth*-deleted mice. Although we did not observe a survival defect of *Fth*-deleted iT_{REG} cells *in vitro* (EV4F-J), these cells are, more sensitive to the cytotoxic effects of Fe or hydrogen peroxide (H₂O₂) (see Figure enclosed). Briefly, the number of iT_{REG} cells generated *in vitro* from mouse T_N cells activated by anti-CD3/CD28 mAb plus IL-2 and TGFβ, was indistinguishable whether the T_N cells were sorted from *Foxp3*^{GFP-FthΔ/Δ} vs. *Foxp3*^{GFP} mice (see panels A-C). When exposed to exogenous Fe however, FTH became essential to maintain the viability and the relative level of Foxp3 expression in iT_{REG} cells generated from T_N cells sorted from *Foxp3*^{GFP-FthΔ/Δ} vs. *Foxp3*^{GFP} mice (see panels A-C). Moreover, the cytotoxic effect of H₂O₂ was also exacerbated in iT_{REG} cells generated from T_N cells sorted from *Foxp3*^{GFP-FthΔ/Δ} vs. *Foxp3*^{GFP} mice (see panels D,E). Similarly, suppression of FTH expression in human nT_{REG} cells, upon transduction of shRNAs targeting FTH, compromised cell viability when exposed to H₂O₂, compared to control T_{REG} cells transduced with non-targeting shRNA (see panels F). These observations suggest that FTH is essential to sustain T_{REG} cell survival when exposed to Fe and/or reactive oxygen species, consistent with the cytoprotective effects of FTH (Berberat *et al*, 2003; Pham *et al*, 2004). This contributes to explain why one does not observe the accumulation of ex-T_{REG} cells in *Foxp3*^{GFP-FthΔ/Δ} mice. This data was not added to the manuscript due to editorial space constraints.

FIGURE LEGEND (Reply to 14). **A**) Schematic representation of the experimental approach used to probe Fe cytotoxicity in mouse iT_{REG} cells, generated *in vitro* from T_N cells. **B**) Representative flow cytometry dot blots of (CD4⁺Foxp3⁺) iT_{REG} cells, analyzed after 5+7 days of *in vitro* culture, as illustrated in (A). **C**) Number (Nbr) and mean fluorescence intensity (MFI) of (CD4⁺Foxp3⁺) iT_{REG} cells from (B). Data pooled from 5 independent experiments with similar trend. **D**) Schematic representation of the experimental approach used to probe H₂O₂ cytotoxicity in mouse iT_{REG} cells, generated *in vitro* from T_N cells. **E**) Viability of (CD4⁺Foxp3⁺) iT_{REG} cells was analyzed with Cell Counting Kit 8 (CCK-8; λ₄₅₀ nm) after 5+2 days of *in vitro* culture with or without H₂O₂, as illustrated in (D). Data pooled from 2 independent experiments with similar trend. **F**) Viability of human (CD45RA⁺CD25⁺CD127⁻) naive T_{REG} cells, transduced with lentiviral vectors expressing shRNA against *FTH1* or with control non-targeting shRNA before exposure to H₂O₂. Cell viability was determined by Annexin V staining and near infra red (IR). Circles represent mean values of separate donors, from two independent experiments. P values calculated by two-tailed t-test across the range of H₂O₂ concentrations, excluding (900 μM; 100% death).

As to the point: “Is the data show figure 4E-F a pool of different ages (19-30 weeks old)? How many mice in each age group?” The data shown in Fig. 4E-F is a pool of mice listed below:

	N	Gender	Age (weeks)
Foxp3 ^{GFP-tdT}	1	female	28.71428571
	2	female	18.85714286
	3	male	25.42857143
	4	male	25.42857143
Foxp3 ^{GFP-FthΔΔ-tdT}	1	female	29.71428571
	2	female	18.85714286
	3	male	23.85714286
	4	male	23.85714286

15. "irrespectively of bystander activation" appears in the text 3 times in a row: in page 8 and twice in page 9.

Reply to 15) The text was edited to remove this redundancy.

16. Can the authors explain why there is no decrease in *Foxp3* in Fig 8H (but there is a decrease in GFP shown in Fig 8G) expression in splenic Tregs of mice carrying FTH-deficient Tregs?

Reply to 16) The data in Fig. 8H (previous version of the manuscript), now illustrated in Fig. 7H, shows the relative quantification of endogenous *Foxp3* mRNA expression, which reports on transcriptional and post-transcriptional regulation of *Foxp3* mRNA expression, from the endogenous mouse *Foxp3* locus. The relative quantification of *Gfp* mRNA expression (shown on the same Figure), which reports on *Gfp* transcription from an intact *Foxp3* locus inserted in a bacterial artificial chromosome (BAC) transgene (Chen *et al.*, 2003; Zhou *et al.*, 2008). Quantification of *Gfp* mRNA is a surrogate for *Foxp3* transcription, excluding post-transcriptional regulation of *Foxp3* mRNA expression (not exerted over *Gfp* mRNA). The apparent discrepancy between *Gfp* and *Foxp3* mRNA expression upon *Fth* deletion in T_{REG} cells may therefore reflect a putative post-transcriptional regulation of *Foxp3* mRNA expression, not exerted over *Gfp* mRNA expression. This interpretation is in keeping with FTH controlling *Foxp3* transcription via regulation of CpG-rich CNS1 and 2 demethylations in the *Foxp3* locus.

Referee #3:

The role of Ferritin H chain protein (FTH) is investigated. In this well performed study it is found to be preferentially expressed in FOXP3 Tregs and plays a role in controlling the activation of the Tregs in the periphery in a cell autonomous way using mice with specific deletion of FTH in Tregs. Technically the work is largely OK except some few issues.

Authors reply: In the assessment of our study the reviewer states that the biological insight provided by the study is questionable. We will argue why we believe that this is not an accurate evaluation of the findings reported in this manuscript.

1) *The genetic background of the mice are not detailed in particular since it can be assumed that different backgrounds are mixed due to the breeding of different modified genes. In addition, it is not clarified if the investigated mice are littermates, which is needed in these types of experiments.*

Reply to 1) All mice used are in the same genetic background, as indicated in the Key Resources table provided in the original submission. We added a more detailed description of the mouse strains used for clarity.

"The genetic background of the mouse strains used was C57BL/6J, including *Foxp3*^{GFP} mice, backcrossed into C57BL/6/J background for over 10 generations. *Rag2*^{-/-} mice used as recipients of BM precursor cells were in C57BL/6NTac background."

The reviewer raises concerns as to whether "mice are littermates, which is needed in these types of experiments.". We used both *Foxp3*^{GFP} and *Fth*^{fl/fl} mice as controls of *Foxp3*^{GFP-FthΔΔ} mice. *Fth*^{fl/fl} mice are littermates of *Foxp3*^{GFP-FthΔΔ} mice. The *Foxp3*^{GFP} and *Foxp3*^{GFP-tdT} mice used as controls for *Foxp3*^{GFP-FthΔΔ} and *Foxp3*^{GFP-FthΔΔ-tdT} mice, respectively, were not littermates as the breeding strategy required to generate the very large number of mice required in these experiments would be associated with ethical, logistic, and financial constraints that are not acceptable. Instead, *Foxp3*^{GFP} and *Foxp3*^{GFP-tdT} mice were co-housed

with *Foxp3*^{GFP-Fth $\Delta\Delta$} and *Foxp3*^{GFP-Fth $\Delta\Delta$ -tdT} mice, respectively, to allow for microbiota “homogenization” before performing the experiments. We repeated a number of experiments using *Foxp3*^{GFP} mice as controls and without any difference from control *Fth*^{f/f} mice.

2) *It is shown that FTH has increased expression in Tregs vs conventional T cells. I think there are two issues missing. Firstly, to determine whether the increased expression is simply due to that activation stage of the cells. Secondly, to quantify the expression in Treg compared to any other cell in the body, i.e. how unique is the overexpression in Tregs. I guess its not, but it should be shown and discussed.*

Reply to 2) Related to: “*Firstly, to determine whether the increased expression is simply due to that activation stage of the cells.*” As stated by the reviewer a comparison of the relative level of FTH expression in T_{REG} cells vs. T_H cells was performed in a previous study by two co-authors, cited in the current manuscript (PMID: 29752063). This data shows that ferritin heavy and light chains (FTH1, FLT) are expressed at higher abundance in T_{REG} cells, compared to T_{CONV} cells (*please see Fig. 4A and 4B in PMID: 29752063*). In the current manuscript we extended this finding and provide a comparison of the relative levels of FTH protein expression in human (*Fig. 1A,B*) and mouse T_{REG} vs. T_{CONV} cells (*Fig. 1C,D*). This data confirmed that FTH expression is expressed at higher levels in T_{REG} cells, compared to T_{CONV} cells. We also added data showing that FTH expression is highly induced in iT_{REG} cells generated *in vitro* from naive T_N cells (*Fig. 1E-G*) and added the following paragraph to the *Results* section:

“We then monitored whether FTH expression is up-regulated in mouse induced T_{REG} (iT_{REG}) cells, generated from T_N cells activated *in vitro* with anti-CD3/CD28 mAb plus IL-2 and TGF β (Chen *et al.*, 2003) (*Fig. 1E*). To this aim we used *Foxp3*^{GFP} T_{REG} cell reporter mice, in which a green fluorescent protein (GFP) humanized Cre-recombinase (GFP-hCre) coding sequence is inserted downstream of the *Foxp3* ATG translational start codon in a bacterial artificial chromosome (BAC) transgene carrying the intact *Foxp3* gene (*Foxp3*-GFP-hCre; referred herein as *Foxp3*^{GFP}) (Chen *et al.*, 2003; Zhou *et al.*, 2008). FTH protein expression was higher under culture conditions containing TGF β and enriched in CD4⁺GFP⁺ iT_{REG} cells, compared to culture conditions lacking TGF β and enriched in CD4⁺GFP⁻ T_{CONV} cells (*Fig. 1F*). Similar trend was observed for *Fth* mRNA, which was overexpressed in iT_{REG} cells (*Fig. 1G*).”

The reviewer suggests that one should: “... *quantify the expression in Treg compared to any other cell in the body*” inferring that this would answer “*how unique is the overexpression in Tregs. I guess its not, but it should be shown and discussed.*” This information is available in databases such as BioGPS (<http://biogps.org/#goto=genereport&id=14319>), ImmGen (<http://rstats.immgen.org/Skyline/skyline.html?gene=Fth1>) or tabula-muris (<https://tabula-muris.ds.czbiohub.org/>). The Reviewer can confirm that the relative level of Fth expression varies widely from one cell type to another under steady state conditions. We ask the Reviewer to please note that FTH expression is under the control of the transcription factors NRF2 (e.g., PMID: 12435735; PMID: 30833408) and NF- κ B (e.g., PMID: 7797515; PMID: 15537542), implying that FTH is an oxidative-stress responsive gene (i.e., regulated by NRF2) that also responds to a variety of pro-inflammatory agonists (i.e., regulated by NF- κ B). It is not accurate therefore to assume that FTH expression is a stable property of all cell types.

3) *My main issue is the value of the work. Obviously, it can be expected that FTH have profound effects in most cells and Tregs are likely to be affected as well. Maybe Treg have slightly higher expression than conventional T cells as they are maintained by self stimulation in the homeostasis. I wouldn't say that FTH regulate these cells, its just an essential gene as many other genes for the function of cells. The FTH is here artificially knocked out, its not a selected polymorphism. Possibly, it is interesting that the expression of FTH blocks*

differentiation to exTreg. It could also be of interest to know more on why FTH is not operating on the thymus levels when it operates at the peripheral level, i.e which activation demands is needed to require help from FTH, this is however not addressed. Following this, knocking out a physiological important protein in Tregs it does not come as a surprise that it enhances EAE and other of the disease models used.

Reply to 3) Although regulation of cellular iron metabolism may be critical to several cell types the Reviewer statement: “*I wouldn't say that FTH regulate these cells, its just an essential gene as many other genes for the function of cells.*” is not accurate, based on a large body of literature. Briefly, several laboratories have used genetic loss of function approaches, similar to the one in our manuscript, to shown that FTH controls systemic iron flux in enterocytes (PMID: 20816093) and hepatocytes (PMID: 19492434), sequesters labile iron to prevent oxidative stress/ferroptosis in renal proximal tubule epithelial cells (PMID: 24018561) and cardiomyocytes (PMID: 32349646) while supporting lymphocyte survival (PMID: 24586648). However, *Fth*-deletion in other cell compartments, including for example in myeloid cells or endothelial cells, is not associated with overt phenotypes, at steady state conditions or even in response to malaria (PMID: 36735532). It is not reasonable therefore to assume that FTH is just an “*essential gene as many other genes for the function of cells.*”

Please note that the data illustrated in *EV4F-J* shows that *Fth*-deletion is not associated with a proliferation defect of T_{REG} cells *in vitro*. This suggests therefore that FTH is not required to support T_{REG} cell proliferation or activated induced cell death associated with T_{REG} cell activation/proliferation.

In our assessment, the major novelty of our study is the demonstration that regulation of intracellular Fe metabolism, such as afforded by FTH, plays a central role maintaining the developmental program enforcing T_{REG} cell identity. Another novel aspect is that this occurs via the regulation of TET-driven demethylation of CpG-rich CNS1 and 2 in the *Foxp3* locus. This “non-canonical” (*i.e.*, novel) property of FTH is supported by a recent manuscript showing that a similar mechanism regulates adipocyte differentiation (Suzuki *et al*, 2023). The following paragraph was added to the *Discussion* section:

“Consistent with our findings, intracellular Fe mobilization via lysosome-mediated ferritinophagy, was recently shown to regulate TET-driven (de)methylation of the peroxisome proliferator activated receptor γ (PPAR γ) locus, the master regulators of adipocyte development (Suzuki *et al.*, 2023). This suggest that FTH controls Fe-responsive epigenetic programs defining different cellular developmental programs.”

4) *The most surprising, and interesting, effect is the lack of effect of FTH in the thymus assuming that FTH play an essential redox regulatory role in all cells during the activation pressure. Thus, one would assume it play a role in Tregs during selection in the thymus. However, this is not addressed.*

Reply to 4) We agree with that exploring further a role for Fth in thymic T_{REG} cells is very interesting and important. In that sense we addressed experimentally whether *Fth*-deletion controls the number of thymic T_{REG} cells. We found that the number of T_{REG} cells is not affected in *Foxp3*^{GFP-Fth $\Delta\Delta$} vs. control *Foxp3*^{GFP} mice. The following paragraph was edited in the Results section:

“The frequency of thymic CD4⁺GFP⁺ T_{REG} cells was lower in *Foxp3*^{GFP-Fth $\Delta\Delta$} vs. control *Foxp3*^{GFP} mice (*Fig. 1H*). This was not associated however, with a concomitant reduction in the number of CD4⁺GFP⁺ T_{REG} cells (*Fig. 1H*). In contrast, there was a marked reduction of both the frequency and numbers of T_{REG} cells in the spleen (*Fig. 1I*) and in the MLN (*Fig. 1J*) of *Foxp3*^{GFP-Fth $\Delta\Delta$} vs. control *Foxp3*^{GFP} mice. The frequency and number of CD4⁺GFP⁺CXCR5⁺PD1⁺ follicular T_{REG} (FT_{REG}) cells was also decreased

in the spleen of *Foxp3^{GFP-FthΔΔ}* vs. control *Foxp3^{GFP}* mice (EV1C). These observations suggest that regulation of intracellular Fe by FTH is required to sustain the number of circulating T_{REG} and FT_{REG} cells in, without interfering with thymic T_{REG} cell output.”

We found that *Foxp3*-GFP signal (MFI) is reduced in *Fth*-deleted T_{REG} cells in the thymus (Fig. 1K). The following paragraph was introduced to the *Results* section to describe this data:

“We noticed that the relative level of GFP expression, reporting on *Foxp3* transcription under the control of an intact *Foxp3* promoter (Chen *et al.*, 2003; Zhou *et al.*, 2008), was reduced in thymic, splenic and MLN T_{REG} cells from *Foxp3^{GFP-FthΔΔ}* vs. control *Foxp3^{GFP}* mice (Fig. 1K). These observations suggest that FTH regulates *Foxp3* transcription in the thymus as well as in circulating T_{REG} cells, which is not sufficient however, to interfere with thymic T_{REG} cell output.”

We found that *Fth*-deletion is associated with a reduction of thymic-T_{REG} (tT_{REG}) cells as well as peripherally derived T_{REG} (pT_{REG}) cells in the MLN (Fig. 1L) and in the spleen (data not shown, see Figure below). The following paragraph was introduced to the *Results* section to describe this data:

“Thymic and peripheral T_{REG} cell development gives rise to tT_{REG} and iT_{REG} cells, expressing neuropilin1 (*Nrp1*) or not, respectively (Weiss *et al.*, 2012; Yadav *et al.*, 2012). The frequency and number of *Nrp1*⁺ tT_{REG} cells and *Nrp1*⁻ iT_{REG} cells was reduced, to the same extent, as assessed in the MLN (Fig. 1L) of *Foxp3^{GFP-FthΔΔ}* vs. control *Foxp3^{GFP}* mice. Similar data was obtained in the spleen (*data not shown*). This suggests that regulation of Fe metabolism by FTH is required for maintenance of thymic-derived and peripherally induced T_{REG} cells.”

The data for splenic tT_{REG} and pT_{REG} cells was not included in the manuscript due to editorial space constrains and is referred to as *data not shown*. The data is provided hereby to the reviewer:

FIGURE LEGEND (Reply 4a): Schematic representation of the experimental approach (left), and representative flow cytometry dot plots (upper right panels) with corresponding quantification of the percentage (%) and number (Nbr.) (bottom right panels) of live (TCRβ⁺CD4⁺GFP⁺) *Nrp1*⁺ T_{REG} cells and *Nrp1*⁻ pT_{REG} cells in the spleen. Data from N=8 mice *per* genotype, pooled from 2 independent experiments with similar trend. Circles represent individual mice and red

bars are mean values. Data from the same mice as in (Fig. 1L). P values were determined using Two-way ANOVA with Sidak's multiple comparisons test. NS (P>0.05); ***P < 0.001.

We found that *Fth* deletion is associated with a reduction of *Foxp3*-GFP signal (MFI) thymic-derived T_{REG} (tT_{REG}) cells as well as peripherally-derived T_{REG} (pT_{REG}) cells in the MLN (Fig. 7C) and in the spleen (data not shown, see Figure below). The following paragraph was edited in the Results section of the manuscript:

“Having established that regulation of Fe metabolism by FTH is required for maintenance of thymic-derived and peripherally induced T_{REG} cells (Fig. 1L) we compared the relative level of *Foxp3*-GFP expression in tT_{REG} and pT_{REG} cells. The relative level of *Foxp3*-GFP expression was reduced in both Nrp1⁺ tT_{REG} cells and Nrp1⁻ pT_{REG} cells, as assessed in the MLN from *Foxp3*^{GFP-FthΔΔ} vs. control *Foxp3*^{GFP} mice (Fig. 7C). Similar data was obtained for splenic T_{REG} cells (data not shown). Considering that GFP is expressed of under the control of a bacterial artificial chromosome (BAC) transgene carrying the intact *Foxp3* gene (Chen *et al.*, 2003; Zhou *et al.*, 2008), these observations suggest that FTH is essential to sustain *Foxp3* transcription in tT_{REG} and pT_{REG} cells.”

Similar data was obtained for the spleen, not shown in the manuscript, and provided hereby to the reviewer.

FIGURE LEGEND (Reply 4c): Schematic representation of the experimental approach (left panel) used to monitor the expression of the GFP-hCre transgene in splenic (CD4⁺GFP⁺Nrp1⁺) tT_{REG} cells and (CD4⁺GFP⁺Nrp1⁻) pT_{REG} cells. Representative flow cytometry histogram of GFP-hCre (middle panel). Relative quantification of GFP-hCre expression (right panel), represented as mean fluorescence intensity (MFI). Data from N=8 mice *per* genotype, pooled from 2 independent experiments with similar trend. Circles represent individual mice and red bars are mean values. P values were calculated using two-way ANOVA with Sidak's multiple comparison test. ***P < 0.001.

Finally, we analyzed the relative levels of *Fth* mRNA expression in T_{REG} cells, FACS-sorted from the thymus (tT_{REG} cells) or lymph nodes (pT_{REG} cells) of adult C57BL/6 mice as well from iT_{REG} cells generated *in vitro* (see Reply to comment 1). The data obtained is provided to the reviewer. While there is an overall tendency for the level of *Fth* mRNA to be higher in iT_{REG} cells and pT_{REG} cells vs. tT_{REG} cells, this is, however, not statically significant. We conclude that there are no major differences in *Fth* mRNA expression between tT_{REG} and pT_{REG} cells.

FIGURE LEGEND (Reply 2). *Fth* mRNA expression normalized to *Arbp0* in $CD4^+GFP^+CD25^+CD8^-B220^-CD11b^-$ Treg cells FACS-sorted from the thymus and lymph nodes as well as in iT_{REG} cells, generated *in vitro* as detailed in *Fig. 1E-G* from N=3-5 mice from three experiments. P values were calculated with ordinary one-way ANOVA with Tukey's multiple comparisons test and show no significant differences.

References cited

www.r-project.org.

Alaluf E, Vokaer B, Detavernier A, Azouz A, Splittgerber M, Carrette A, Boon L, Libert F, Soares M, Le Moine A *et al* (2020) Heme oxygenase-1 orchestrates the immunosuppressive program of tumor-associated macrophages. *JCI Insight* 5

Bailey-Bucktrout SL, Martinez-Llordella M, Zhou X, Anthony B, Rosenthal W, Luche H, Fehling HJ, Bluestone JA (2013) Self-antigen-driven activation induces instability of regulatory T cells during an inflammatory autoimmune response. *Immunity* 39: 949-962

Berberat PO, Katori M, Kaczmarek E, Anselmo D, Lassman C, Ke B, Shen X, Busuttill RW, Yamashita K, Csizmadia E *et al* (2003) Heavy chain ferritin acts as an antiapoptotic gene that protects livers from ischemia reperfusion injury. *FASEB Journal* 17: 1724-1726

Chen W, Jin W, Hardegen N, Lei KJ, Li L, Marinos N, McGrady G, Wahl SM (2003) Conversion of peripheral CD4+CD25- naive T cells to CD4+CD25+ regulatory T cells by TGF-beta induction of transcription factor Foxp3. *J Exp Med* 198: 1875-1886

Consonni FM, Bleve A, Totaro MG, Storto M, Kunderfranco P, Termanini A, Pasqualini F, Ali C, Pandolfo C, Sgambelluri F *et al* (2021) Heme catabolism by tumor-associated macrophages controls metastasis formation. *Nat Immunol* 22: 595-606

Duhen R, Glatigny S, Arbelaez CA, Blair TC, Oukka M, Bettelli E (2013) Cutting edge: the pathogenicity of IFN-gamma-producing Th17 cells is independent of T-bet. *J Immunol* 190: 4478-4482

Ferreira A, Balla J, Jeney V, Balla G, Soares MP (2008) A central role for free heme in the pathogenesis of severe malaria: the missing link? *J Mol Med* 86: 1097-1111

Frost JN, Tan TK, Abbas M, Wideman SK, Bonadonna M, Stoffel NU, Wray K, Kronsteiner B, Smits G, Campagna DR *et al* (2021) Hcpidin-Mediated Hypoferremia Disrupts Immune Responses to Vaccination and Infection. *Med (N Y)* 2: 164-179 e112

Gao X, Song Y, Lu S, Hu L, Zheng M, Jia S, Zhao M (2022a) Insufficient Iron Improves Pristane-Induced Lupus by Promoting Treg Cell Expansion. *Front Immunol* 13: 799331

Gao X, Song Y, Wu J, Lu S, Min X, Liu L, Hu L, Zheng M, Du P, Yu Y *et al* (2022b) Iron-dependent epigenetic modulation promotes pathogenic T cell differentiation in lupus. *J Clin Invest* 132

Gavin MA, Rasmussen JP, Fontenot JD, Vasta V, Manganiello VC, Beavo JA, Rudensky AY (2007) Foxp3-dependent programme of regulatory T-cell differentiation. *Nature* 445: 771-775

Gozzelino R, Andrade BB, Larsen R, Luz NF, Vanoaica L, Seixas E, Coutinho A, Cardoso S, Rebelo S, Poli M *et al* (2012) Metabolic adaptation to tissue iron overload confers tolerance to malaria. *Cell Host Microbe* 12: 693-704

Hisaeda H, Maekawa Y, Iwakawa D, Okada H, Himeno K, Kishihara K, Tsukumo S, Yasutomo K (2004) Escape of malaria parasites from host immunity requires CD4+ CD25+ regulatory T cells. *Nat Med* 10: 29-30

Jiang Y, Li C, Wu Q, An P, Huang L, Wang J, Chen C, Chen X, Zhang F, Ma L *et al* (2019) Iron-dependent histone 3 lysine 9 demethylation controls B cell proliferation and humoral immune responses. *Nat Commun* 10: 2935

Kurup SP, Obeng-Adjei N, Anthony SM, Traore B, Doumbo OK, Butler NS, Crompton PD, Harty JT (2017) Regulatory T cells impede acute and long-term immunity to blood-stage malaria through CTLA-4. *Nat Med* 23: 1220-1225

Medzhitov R, Schneider D, Soares M (2012) Disease Tolerance as a Defense Strategy. *Science* 335: 936-941

Morikawa H, Ohkura N, Vandenbon A, Itoh M, Nagao-Sato S, Kawaji H, Lassmann T, Carninci P, Hayashizaki Y, Forrest AR *et al* (2014) Differential roles of epigenetic changes and Foxp3 expression in regulatory T cell-specific transcriptional regulation. *Proc Natl Acad Sci U S A* 111: 5289-5294

Pham CG, Bubici C, Zazzeroni F, Papa S, Jones J, Alvarez K, Jayawardena S, De Smaele E, Cong R, Beaumont C *et al* (2004) Ferritin heavy chain upregulation by NF-kappaB inhibits TNFalpha-induced apoptosis by suppressing reactive oxygen species. *Cell* 119: 529-542

Ramos S, Ademolue TW, Jentho E, Wu Q, Guerra J, Martins R, Pires G, Weis S, Carlos AR, Mahú I *et al* (2022) A hypometabolic defense strategy against malaria. *Cell Metab*

Ramos S, Carlos AR, Sundaram B, Jeney V, Ribeiro A, Gozzelino R, Bank C, Gjini E, Braza F, Martins R *et al* (2019) Renal control of disease tolerance to malaria. *Proc Natl Acad Sci U S A* 116: 5681-5686

Seixas E, Gozzelino R, Chora A, Ferreira A, Silva G, Larsen R, Rebelo S, Penido C, Smith NR, Coutinho A *et al* (2009) Heme oxygenase-1 affords protection against noncerebral forms of severe malaria. *Proc Natl Acad Sci U S A* 106: 15837-15842

Soares MP, Teixeira L, Moita LF (2017) Disease tolerance and immunity in host protection against infection. *Nat Rev Immunol* 17: 83-96

Suzuki T, Komatsu T, Shibata H, Tanioka A, Vargas D, Kawabata-Iwakawa R, Miura F, Masuda S, Hayashi M, Tanimura-Inagaki K *et al* (2023) Crucial role of iron in epigenetic rewriting during adipocyte differentiation mediated by JMJD1A and TET2 activity. *Nucleic Acids Res* 51: 6120-6142

Teh MR, Frost JN, Armitage AE, Drakesmith H (2021) Analysis of Iron and Iron-Interacting Protein Dynamics During T-Cell Activation. *Front Immunol* 12: 714613

Voss K, Sewell AE, Krystofiak ES, Gibson-Corley KN, Young AC, Basham JH, Sugiura A, Arner EN, Beavers WN, Kunkle DE *et al* (2023) Elevated transferrin receptor impairs T cell metabolism and function in systemic lupus erythematosus. *Sci Immunol* 8: eabq0178

Walther M, Tongren JE, Andrews L, Korbel D, King E, Fletcher H, Andersen RF, Bejon P, Thompson F, Dunachie SJ *et al* (2005) Upregulation of TGF-beta, FOXP3, and CD4+CD25+ regulatory T cells correlates with more rapid parasite growth in human malaria infection. *Immunity* 23: 287-296

Weiss JM, Bilate AM, Gobert M, Ding Y, Curotto de Lafaille MA, Parkhurst CN, Xiong H, Dolpady J, Frey AB, Ruocco MG *et al* (2012) Neuropilin 1 is expressed on thymus-derived natural regulatory T cells, but not mucosa-generated induced Foxp3+ T reg cells. *J Exp Med* 209: 1723-1742, S1721

Wideman SK, Frost JN, Richter FC, Naylor C, Lopes JM, Viveiros N, Teh MR, Preston AE, White N, Yusuf S *et al* (2023) Cellular iron governs the host response to malaria. *PLoS Pathog* 19: e1011679

Williams LM, Rudensky AY (2007) Maintenance of the Foxp3-dependent developmental program in mature regulatory T cells requires continued expression of Foxp3. *Nat Immunol* 8: 277-284

Wu Q, Sacomboio E, Valente de Souza L, Martins R, Kitoko J, Cardoso S, Ademolue TW, Paixao T, Lehtimaki J, Figueiredo A *et al* (2023) Renal control of life-threatening malarial anemia. *Cell Rep* 42: 112057

Xu M, Pokrovskii M, Ding Y, Yi R, Au C, Harrison OJ, Galan C, Belkaid Y, Bonneau R, Littman DR (2018) c-MAF-dependent regulatory T cells mediate immunological tolerance to a gut pathobiont. *Nature* 554: 373-377

Yadav M, Louvet C, Davini D, Gardner JM, Martinez-Llordella M, Bailey-Bucktrout S, Anthony BA, Sverdrup FM, Head R, Kuster DJ *et al* (2012) Neuropilin-1 distinguishes natural and inducible regulatory T cells among regulatory T cell subsets in vivo. *J Exp Med* 209: 1713-1722, S1711-1719

Zhou X, Jeker LT, Fife BT, Zhu S, Anderson MS, McManus MT, Bluestone JA (2008) Selective miRNA disruption in T reg cells leads to uncontrolled autoimmunity. *J Exp Med* 205: 1983-1991

Zhu L, Li G, Liang Z, Qi T, Deng K, Yu J, Peng Y, Zheng J, Song Y, Chang X (2023) Microbiota-assisted iron uptake promotes immune tolerance in the intestine. *Nat Commun* 14: 2790

Dear Dr. Soares,

Thank you for the submission of your revised manuscript to The EMBO Journal. We have now received the comments of the three referees that were asked to re-assess your study (included below). As you will see, referees #2 and #3 are satisfied with the revision and explain that their previous concerns have been satisfactorily addressed. Please note that referee #2 asks for the inclusion of a suitable reference or clarification regarding their first major comment. Referee #1 also recognizes that the study has been improved but they also point out that three of their concerns have been only incompletely addressed. Given the reviewers' positive comments and recommendations, I would like to invite you to submit another revised version of your manuscript, addressing all remaining concerns, along with a detailed point-by-point response. I should add that acceptance of the manuscript will depend on the completeness of your responses in this revision. Please let me know if you have any questions or comments.

From the editorial side, there are also a number of minor changes and corrections that we need from you before we can proceed with handling of your manuscript:

- Please enter all relevant funding information in our online manuscript handling system (eJP). It should match exactly the information provided in the Acknowledgements section of your manuscript. We noticed that the following are: missing in the manuscript file: 952537; LSBR 1818 is listed three times, Calouste Gulbenkian Foundation listed multiple times in eJP; missing in eJP: FCT (2022.08590.PTDC); FCT (283/BI/15; UID/Multi/04555/2013); SymbNET-Genomics and Metabolomics in a Host-Microbe Symbiosis Network Project (European Union's Horizon 2020); CONGENTO LISBOA-01-0145-FEDER-022170 (FCT, Lisboa2020, Por2020, ERDF).
- Please provide a list of up to 5 keywords after the Abstract in your revised manuscript.
- Please make sure that all new primary (transcriptomics, proteomics, and metabolomics) datasets are deposited in public databases, will be publicly available at the time of publication, and their access information (including specific URLs) is included in your Data Availability section at the end of the Materials and Methods. The reviewer tokens can now be removed from this section.
- Please change the heading of your "Declaration of interests" statement to "Disclosure and competing interests statement".
- The author contributions statement should be removed from the manuscript file. Instead, we use CRediT to specify the contributions of each author in the journal submission system. Please use the free text box to provide more detailed descriptions. See also our guide to authors:
<https://www.embopress.org/page/journal/14602075/authorguide#authorshipguidelines>.
- According to our journal's policy, "data not shown/published" (stated twice on page 5, once on page 13, and twice on page 14 of your manuscript) is not permitted. All data referred to in the paper should be displayed in the main or Expanded View figures, or in the Appendix. Please add these data or change the text accordingly if these data are not central to the study and its conclusions.
- Please update the callouts for EV Figures to "Figure EV#" throughout your manuscript.
- Please clarify which table the callout for "Table 1" refers to.
- Please provide in the first paragraph of your Materials and Methods the reference number for the ethics approval of your experiments involving animals.
- Please note that EMBO press papers are accompanied online by:
 - A) a short (2 sentences) summary of the findings and their significance,
 - B) 2-5 short bullet points highlighting the key results, and
 - C) a synopsis image that is exactly 550 pixels wide and 300-600 pixels high (the height is variable). You can either show a model or key data in the synopsis image. Please note that the text needs to be readable at the final size.Please upload this information along with your revised manuscript (the text for A and B should be provided in a separate Word file).
- Please note that a separate "Data Information" section is required in the legends of Figures 5a-d. You can find more information and an example in our guide:
<https://www.embopress.org/page/journal/14602075/authorguide#figureformat>.
- Please indicate the statistical test used for data analysis in the legends of Figures 2e; 4m; 6c.

- Please note that in Figures 4e-g; 6g-i; 8b-j there is a mismatch between the annotated p values in the Figure legend and the annotated p values in the Figure file that should be corrected.
- Please note that information related to "n" is missing in the legends of Figures 1b; 4c, e-g; 8d.
- Please note that the error bars are not defined in the legends of Figures 3h-i.
- "Summary" should be changed to "Abstract".
- The legends of the main Figures should be moved after the References.
- The EV Figure legends should not be uploaded as an individual file but inserted after the main Figure legends in the manuscript file.
- The Table on page 32 should be renamed "Table 1" (instead of Table S1).

We look forward to seeing a final version of your manuscript as soon as possible. Please use this link to submit your revision: <https://emboj.msubmit.net/cgi-bin/main.plex>

Yours sincerely,

 Referee #1:

The paper has been certainly improved after the first round of revisions but this referee still finds some of the responses incomplete and further needing additional data.
 Specifically:

1) Query The question was "is the FTH specific for Treg cells or rather its expression links with T cell proliferation independent of its role in FoxP3 expression?".

They decided to perform an experiment that only partly answered the question, i.e. comparing the levels in iTreg vs Tconv and did not assess the level of expression of Treg fresh/ex vivo at all. They referred back to two papers they had published in the past in which they saw that two genes FTH1 and FLT were expressed at high levels in Treg when compared to Tconv.

2) Query In the mouse is there any difference FTH levels in tTreg, pTreg and iTreg cells?

The answer they brought showed an experiment they conducted with a very small number of mice justifying an upward trend in iTreg and pTreg vs tTreg that absolutely cannot be taken into account considering the relevance of the question asked by this referee.

Their assumption that there are no substantial differences in Fth mRNA expression between tTreg and pTreg is indeed, an assumption based on a small n but above all while distribution of the 3 mice of tTreg and iTreg is homogeneous, the LN Treg show a large variability. This issue can only be solved with more mice.

3) Query This referee still sticks to the original observation. EAE in the controls not reaching 2.5/3 scores (but only 1) with a standard dose of MOG utilized raises the question on the protocol utilized to induce EAE, as these are not low-doses in MOG utilized (please see the methods). This referee encourages the authors to repeat EAE to a standard level in the controls to reach 2,5-3 scores and address whether Fth deletion really leads to increase in EAE susceptibility, mortality, scores etc.

Referee #2:

The revised version offers significant improvement for the manuscript. The authors have addressed most, if not all, comments raised by all the reviewers. A couple of comments raised by both reviewers 1 and 2 were properly addressed and are now

presented as new panels in figure 1 (panels E-G). As for my specific comments, I am satisfied with all the author's answers. Regarding my first major comment, I'd like to ask the authors to include a reference that shows that FTH promoter contains Foxp3 DNA binding sites or point to where in the manuscript this data is shown (could not find it in the text).

While I partially agree with reviewer #3 about the somewhat expected role of FTH in Tregs given its molecular function, I disagree with the reviewer that the loss of function experiments performed in this manuscript has no value in addressing the role of FTH on Tregs (point #3 of Rev#3). The authors performed well-designed experiments that allowed them to conclude that FTH specifically maintains Treg identity. In addition, the authors gave a body of examples to support their claims.

Referee #3:

My concerns have largely been addressed.

Point by point reply to the Referees:

Referee #1: *The paper has been certainly improved after the first round of revisions but this referee still finds some of the responses incomplete and further needing additional data. Specifically:*

1) *They decided to perform an experiment that only partly answered the question, i.e. comparing the levels in iTreg vs Tconv and did not assess the level of expression of Treg fresh/ex vivo at all. They referred back to two papers they had published in the past in which they saw that two genes FTH1 and FLT were expressed at high levels in Treg when compared to Tconv.*

Reply to 1: We are providing additional data to further illustrate the relative levels of FTH expression in freshly isolated T_{REG} cells. We note however that, in the previous version of the manuscript, we presented data showing that freshly isolated mouse (CD4⁺Foxp3⁺) T_{REG} cells express relatively higher levels of FTH protein, compared to (CD4⁺Foxp3⁻CD44^{low}CD62L^{high}) naïve T_H cells (T_N) or to (CD4⁺Foxp3⁻CD44^{high}CD62L^{low}) memory T_H cells (T_M), as determined by western blot (*please see Fig. 1C,D*). The data enclosed hereby shows that mouse T_{REG} cells express similar levels of *Fth* mRNA, compared to T_N cells. Of note, (CD4⁺Foxp3⁻CD44^{high}CD62L⁻) T_M cells express relatively higher levels of *Fth* mRNA, compared to T_{REG} cells. This suggests that the relatively higher level of FTH protein expression in T_{REG} cells, compared to T_N and T_M cells is enforced post-transcriptionally, similar to described for other cell types (Hentze et al., 1987; Meyron-Holtz et al., 2004; Muckenthaler et al., 2017; Rouault et al., 1988). We are enclosing the new data hereby for the Reviewer and have not included these data in the manuscript as we already show FTH protein expression data (*please see Fig. 1C,D*).

FIGURE LEGEND (Reply 1): Relative quantification of *Fth* mRNA in freshly isolated (FACS-sorted from spleen) CD4⁺Foxp3⁻CD44^{low}CD62L^{high} naïve T cells (T_N), CD4⁺GFP⁺ T_{REG} cells and CD4⁺GFP⁻CD44^{high}CD62L^{low} memory T cells (T_M) by qRT-PCR, normalized to *Arbp0* mRNA. Data from 3 individual mice. Data is represented as mean ± SD. Circles correspond to individual mice and red bars are mean values. P values calculated using Ordinary one-way ANOVA. NS, not significant (P > 0.05), *P < 0.05. Methods for T_{REG} cell sorting and qRT-PCR are detailed in the manuscript.

2) *Query In the mouse is there any difference FTH levels in tTreg, pTreg and iTreg cells? The answer they brought showed an experiment they conducted with a very small number of mice justifying an upward trend in iTreg and pTreg vs tTreg that absolutely cannot be taken into account considering the relevance of the question asked by this referee. Their assumption that there are no substantial differences in Fth mRNA expression between tTreg and pTreg is*

indeed, an assumption based on a small *n* but above all while distribution of the 3 mice of *tTreg* and *iTreg* is homogeneous, the LN *Treg* show a large variability. This issue can only be solved with more mice.

Reply to 2: As per the reviewer request, we added more samples (mice) to this data set to compare the relative levels of *Fth* mRNA expression in T_{REG} cells FACS-sorted from the thymus (tT_{REG} cells) or lymph nodes (pT_{REG} cells) of adult C57BL/6 mice and in iT_{REG} cells generated *in vitro*. We confirmed the data shown in our previous experiments, namely, that the relative level of *Fth* mRNA expression is similar in *tTreg*, *pTreg* and *iTreg* cells. These data is provided to the Reviewer.

FIGURE LEGEND (Reply 2). *Fth* mRNA expression normalized to *Arbp0* in $CD4^+GFP^+CD25^+CD8^-B220^-CD11b^-T_{REG}$ cells, FACS-sorted from the thymus and lymph nodes as well as in iT_{REG} cells, generated *in vitro* as detailed in Fig. 1E-G, from *N*=4-7 mice in four independent experiments. *P* values were calculated with ordinary one-way ANOVA with Tukey's multiple comparisons test and show no significant differences.

3) Query This referee still sticks to the original observation. EAE in the controls not reaching 2.5/3 scores (but only 1) with a standard dose of MOG utilized raises the question on the protocol utilized to induce EAE, as these are not low-doses in MOG utilized (please see the methods). This referee encourages the authors to repeat EAE to a standard level in the controls to reach 2,5-3 scores and address whether *Fth* deletion really leads to increase in EAE susceptibility, mortality, scores etc.

Reply to 3: We have carefully considered this relevant point that suggests a suboptimal immunization dosing in the EAE experiments and addressed it textually by toning down the respective conclusions in the manuscript as needed. The following paragraph was added to the Results section:

“Of note the immunization protocol used was “sub-optimal”, as proposed by the relatively low disease scores in the control immunized *Fth^{f/f}* mice, likely favoring the increase in EAE severity observed in *Foxp3^{GFP-FthΔΔ}* mice.”

Please note that a similar paragraph is included in the Discussion section:

“*Fth* deletion in T_{REG} cells led to an increase in EAE susceptible and severity (Fig. 8A-C), induced by an immunization protocol leading to low-grade disease severity in control mice (Fig. 8C).”

Referee #2: *The revised version offers significant improvement for the manuscript. The authors have addressed most, if not all, comments raised by all the reviewers. A couple of comments raised by both reviewers 1 and 2 were properly addressed and are now presented as new panels in figure 1 (panels E-G). As for my specific comments, I am satisfied with all the author's answers. Regarding my first major comment, I'd like to ask the authors to include a reference that shows that FTH promoter contains Foxp3 DNA binding sites or point to where in the manuscript this data is shown (could not find it in the text).*

While I partially agree with reviewer #3 about the somewhat expected role of FTH in Tregs given its molecular function, I disagree with the reviewer that the loss of function experiments performed in this manuscript has no value in addressing the role of FTH on Tregs (point #3 of Rev#3). The authors performed well-designed experiments that allowed them to conclude that FTH specifically maintains Treg identity. In addition, the authors gave a body of examples to support their claims.

Reply: The suggestion that the *FTH* promoter contains DNA binding sites is supported by *in silico* data, not illustrated in the manuscript due to space constrains. Briefly, the *Fth* promoter region (>FP023586 Fth1_1 :+U EU:NC; range -2000 to 100) was compared with following databases (BLAST analysis):

1) H3K27ac ChIP-seq of Treg replicate 1

1 ION_TORRENT (Ion Torrent Proton) run: 19.6M spots, 2.9G bases, 2.1Gb downloads
Accession: SRX2680370

[https://www.ncbi.nlm.nih.gov/sra/SRX2680370\[accn\]](https://www.ncbi.nlm.nih.gov/sra/SRX2680370[accn])

2) H3K27ac ChIP-seq of Treg replicate 2

1 ION_TORRENT (Ion Torrent Proton) run: 17.7M spots, 2.5G bases, 1.8Gb downloads
Accession: SRX2680369

[https://www.ncbi.nlm.nih.gov/sra/SRX2680369\[accn\]](https://www.ncbi.nlm.nih.gov/sra/SRX2680369[accn])

3) H3K4me3 ChIP-seq of Treg

1 ION_TORRENT (Ion Torrent Proton) run: 13.8M spots, 2G bases, 1.4Gb downloads
Accession: SRX2680336

[https://www.ncbi.nlm.nih.gov/sra/SRX2680336\[accn\]](https://www.ncbi.nlm.nih.gov/sra/SRX2680336[accn])

4) Foxp3 ChIP-seq of Treg

1 ION_TORRENT (Ion Torrent Proton) run: 30.6M spots, 4.5G bases, 3.2Gb downloads
Accession: SRX2680319

[https://www.ncbi.nlm.nih.gov/sra/SRX2680319\[accn\]](https://www.ncbi.nlm.nih.gov/sra/SRX2680319[accn])

FIGURE LEGEND (Reply 2). Blast of -1000 to 100 of *Fth* gene with H3K27ac ChIP-seq of Treg (replicate 1 and 2), H3K4me3 ChIP-seq of Treg and Foxp3 ChIP-seq of Treg

These findings were confirmed in The Eukaryotic Promoter Database (<https://epd.expasy.org/epd>) using the analysis with Transcription factor motifs (Jaspar core 2018 vertebrates). The image below shows the putative FOXP3 binding site in *Fth* promotor:

Referee #3:

My concerns have largely been addressed.

Editorial comments:

From the editorial side, there are also a number of minor changes and corrections that we need from you before we can proceed with handling of your manuscript:

- Please enter all relevant funding information in our online manuscript handling system (eJP). It should match exactly the information provided in the Acknowledgements section of your manuscript. We noticed that the following are: missing in the manuscript file: 952537; LSBR 1818 is listed three times, Calouste Gulbenkian Foundation listed multiple times in eJP; missing in eJP: FCT (2022.08590.PTDC); FCT (283/BI/15; UID/Multi/04555/2013); SymbNET-Genomics and Metabolomics in a Host-Microbe Symbiosis Network Project (European Union's Horizon 2020); CONGENTO LISBOA-01-0145-FEDER-022170 (FCT, Lisboa2020, Por2020, ERDF).

Reply: We have made all the requested edits in the Acknowledgments and eJP sections.

- Please provide a list of up to 5 keywords after the Abstract in your revised manuscript.

Reply: A list of keywords is now provided in the main manuscript, as requested.

- Please make sure that all new primary (transcriptomics, proteomics, and metabolomics) datasets are deposited in public databases, will be publicly available at the time of publication, and their access information (including specific URLs) is included in your Data Availability section at the end of the Materials and Methods. The reviewer tokens can now be removed from this section.

Reply: All datasets are now publicly available.

- Please change the heading of your "Declaration of interests" statement to "Disclosure and competing interests statement".

Reply: This has been corrected.

- The author contributions statement should be removed from the manuscript file. Instead, we use CRediT to specify the contributions of each author in the journal submission system. Please use the free text box to provide more detailed descriptions. See also our guide to authors: <https://www.embopress.org/page/journal/14602075/authorguide#authorshipguidelines>.

Reply: Author contribution section was removed from the manuscript file.

- According to our journal's policy, "data not shown/published" (stated twice on page 5, once on page 13, and twice on page 14 of your manuscript) is not permitted. All data referred to in the paper should be displayed in the main or Expanded View figures, or in the Appendix. Please add these data or change the text accordingly if these data are not central to the study and its conclusions.

Reply: The five statements with "data not shown" were removed. These corresponded to complementary data in the first version of the manuscript, that was not central to the study nor to its conclusions.

- Please update the callouts for EV Figures to "Figure EV#" throughout your manuscript.

Reply: All the callouts for EV Figures were corrected.

- Please clarify which table the callout for "Table 1" refers to.

Reply: There was no callout for "Table 1", there were some for Table EV1, which we have now changed according to your instructions bellow, into the "Table 1" format. There was an erroneous callout for "Expanded View Table EV1" on legend of Figure 4L, which has now been removed.

- Please provide in the first paragraph of your Materials and Methods the reference number for the ethics approval of your experiments involving animals.

Reply: References are now indicated.

- Please note that EMBO press papers are accompanied online by:
A) a short (2 sentences) summary of the findings and their significance,
B) 2-5 short bullet points highlighting the key results, and
C) a synopsis image that is exactly 550 pixels wide and 300-600 pixels high (the height is variable). You can either show a model or key data in the synopsis image. Please note that the text needs to be readable at the final size.
Please upload this information along with your revised manuscript (the text for A and B should be provided in a separate Word file).

Reply: We added: A) a short (2 sentences) summary of the findings and their significance, B) 2-5 short bullet points highlighting the key results, and C) a synopsis image that is exactly 550 pixels wide and 300-600 pixels high.

- Please note that a separate "Data Information" section is required in the legends of Figures 5a-d. You can find more information and an example in our guide:
<https://www.embopress.org/page/journal/14602075/authorguide#figureformat>.

Reply: Figures 5C-E were edited, as requested: "Data is presented as mean \pm SD."

- Please indicate the statistical test used for data analysis in the legends of Figures 2e; 4m; 6c.

Reply: For Figures 2E and 4M, the following sentences were added to:

The Methods section:

"Functional enrichment analysis was performed with gProfiler (Kolberg et al, 2023). Data was analyzed using g:SCS multiple testing correction method with a significance threshold of 0.05."

To the Figure Legend:

"Data was analyzed using g:SCS multiple testing correction method with a significance threshold of 0.05."

For Figure 6C, the following sentences were added to:

The Methods section:

“Sliding Linear Model (SLIM) multiple testing correction (Wang et al, 2011) was used to extract the significant methylated sites.”

To the Figure Legend:

“Statistical analysis for multiple testing correction was performed with Sliding Linear Model (SLIM).”

- Please note that in Figures 4e-g; 6g-i; 8b-j there is a mismatch between the annotated p values in the Figure legend and the annotated p values in the Figure file that should be corrected.

Reply: Corrected, as proposed.

- Please note that information related to "n" is missing in the legends of Figures 1b; 4c, e-g; 8d.

Reply: All “n” values were added, as requested.

- Please note that the error bars are not defined in the legends of Figures 3h-i.

Reply: This information is now included.

- "Summary" should be changed to "Abstract".

Reply: Corrected, as proposed.

- The legends of the main Figures should be moved after the References.

Reply: Corrected, as proposed.

- The EV Figure legends should not be uploaded as an individual file but inserted after the main Figure legends in the manuscript file.

Reply: Corrected, as proposed.

- The Table on page 32 should be renamed "Table 1" (instead of Table S1).

Reply: Corrected, as proposed.

Dear Dr. Soares,

Thank you for the submission of your revised manuscript to The EMBO Journal. I am glad to say that all remaining referees' concerns have now been addressed satisfactorily and to a sufficient extent for publication of your work in The EMBO Journal. Since the additional data you present in your point-by-point response to the referees add value to the work by strengthening your conclusions, I would like to ask you to incorporate them all in your manuscript. For the new Figure panels, you could include them in your existing main and/or EV Figures, if you wish, but alternatively I would recommend you to provide them in an Appendix. This should be a single PDF file with a brief Table of Contents on its first page. The supplementary figures contained in the Appendix should be annotated as "Appendix Figure S1", "Appendix Figure S2" etc. and should also be called out in the main manuscript file.

Please also address the following remaining editorial requests so that we can proceed with acceptance of your work for publication:

- Your "Data availability" section already contains information about the RNA sequencing and DNA methylation datasets generated in your study, but no database identifiers and URLs are provided for the proteomics-metabolomics/mass spectrometry datasets. Please include those as well using the same format.
- We noticed that your Materials and Methods contain a detailed resources table. This fits better our optional "Structured Methods" format that consists of a "Reagents and Tools Table" -to be uploaded separately to the manuscript tracking system as a "Reagent Table" file- followed by a "Methods and Protocols" section, where methods are described (i.e. your existing text). Please find here more information, links to templates and examples, and detailed instructions:
<https://www.embopress.org/page/journal/14602075/authorguide>.
- You are welcome to include one more keyword, if you like (you currently list 4).
- Please replace the website "www.r-project.org" with a proper complete citation of R in your References list.
- Please move Table 1, together with its legend, to the end of the manuscript file (after Figure legends).
- Could you also please upload your Synopsis image as a .png or .jpg file with the proper dimensions (550 pixels wide x 300-600 pixels high)? Please make sure that all text should be legible at these final dimensions.
- Please also note that a separate "Data Information" section is required at the end of the legend of Figure 5. You can find an example in our guide:
<https://www.embopress.org/page/journal/14602075/authorguide#figureformat>.

We look forward to seeing a final version of your manuscript as soon as possible. Please use this link to submit your revision:
<https://emboj.msubmit.net/cgi-bin/main.plex>

Yours sincerely,

Point-by-point reply

Dear Dr. Soares,

Thank you for the submission of your revised manuscript to The EMBO Journal. I am glad to say that all remaining referees' concerns have now been addressed satisfactorily and to a sufficient extent for publication of your work in The EMBO Journal. Since the additional data you present in your point-by-point response to the referees add value to the work by strengthening your conclusions, I would like to ask you to incorporate them all in your manuscript. For the new Figure panels, you could include them in your existing main and/or EV Figures, if you wish, but alternatively I would recommend you to provide them in an Appendix. This should be a single PDF file with a brief Table of Contents on its first page. The supplementary figures contained in the Appendix should be annotated as "Appendix Figure S1", "Appendix Figure S2" etc. and should also be called out in the main manuscript file.

Reply: As suggested, we provide an Appendix as a single PDF file with a brief Table of Contents on its first page. The supplementary figures contained in the Appendix are annotated as "Appendix Figure S1", "Appendix Figure S2" etc and are called out in the main manuscript file. The remaining editorial requests were addressed in a point by point reply, as detailed below.

Please also address the following remaining editorial requests so that we can proceed with acceptance of your work for publication:

- Your "Data availability" section already contains information about the RNA sequencing and DNA methylation datasets generated in your study, but no database identifiers and URLs are provided for the proteomics-metabolomics/mass spectrometry datasets. Please include those as well using the same format.

Reply: The proteomics reference in the methods was wrongly placed as this data is no longer part of the current manuscript. As it related to the metabolomics data, the title was clarified into "Targeted metabolomics", to highlight that only few metabolites were analysed and all the data is present in the Source data files provided.

- We noticed that your Materials and Methods contain a detailed resources table. This fits better our optional "Structured Methods" format that consists of a "Reagents and Tools Table" -to be uploaded separately to the manuscript tracking system as a "Reagent Table" file- followed by a "Methods and Protocols" section, where methods are described (i.e. your existing text). Please find here more information, links to templates and examples, and detailed instructions: <https://www.embopress.org/page/journal/14602075/authorguide>.

Reply: Changed accordingly.

- You are welcome to include one more keyword, if you like (you currently list 4).

Reply: A fifth keyword (FOXP3) was added

- Please replace the website "www.r-project.org" with a proper complete citation of R in your References list.

Reply: Reference was edited and is now properly cited.

- Please move Table 1, together with its legend, to the end of the manuscript file (after Figure legends).

Reply: Figure has been moved

- Could you also please upload your Synopsis image as a .png or .jpg file with the proper dimensions (550 pixels wide x 300-600 pixels high)? Please make sure that all text should be legible at these final dimensions.

Reply: A synopsis image is provided in png format and with the proper dimensions (550 pixels wide x 300-600 pixels high)

*- Please also note that a separate "Data Information" section is required at the end of the legend of Figure 5. You can find an example in our guide:
<https://www.embopress.org/page/journal/14602075/authorguide#figureformat>.*

Reply: We have added "data information" in all the needed legends to create a separate section.

Dear Dr. Soares,

I am pleased to inform you that your manuscript has been accepted for publication in The EMBO Journal.

Yours sincerely,
